# ROUTING CHANNEL-PATCH DEPENDENCIES IN TIME SERIES FORECASTING WITH GRAPH SPECTRAL DECOMPOSITION

**Dongyuan Li**[1]    **Shun Zheng**[2]    **Chang Xu**[2]    **Jiang Bian**[2]    **Renhe Jiang**[1*]

[1]The University of Tokyo    [2]Microsoft Research Asia

{lidy, Jiangrh}@csis.u-tokyo.ac.jp
{shun.zhen, chanx, jiang.bian}@microsoft.com

## ABSTRACT

Time series forecasting has attracted significant attention in the field of AI. Previous works have revealed that the Channel-Independent (CI) strategy improves forecasting performance by modeling each channel individually, but it often suffers from poor generalization and overlooks meaningful inter-channel interactions. Conversely, Channel-Dependent (CD) strategies aggregate all channels, which may introduce irrelevant information and lead to oversmoothing. Despite recent progress, few existing methods offer the flexibility to adaptively balance CI and CD strategies in response to varying channel dependencies. To address this, we propose a generic plugin xCPD, that can adaptively model the channel-patch dependencies from the perspective of graph spectral decomposition. Specifically, xCPD first projects multivariate signals into the frequency domain using a shared graph Fourier basis, and groups patches into low-, mid-, and high-frequency bands based on their spectral energy responses. xCPD then applies a channel-adaptive routing mechanism that dynamically adjusts the degree of inter-channel interaction for each patch, enabling selective activation of frequency-specific experts. This facilitates fine-grained input-aware modeling of smooth trends, local fluctuations, and abrupt transitions. xCPD can be seamlessly integrated on top of existing CI and CD forecasting models, consistently enhancing both accuracy and generalization across benchmarks. The code is available [https://github.com/Clearloveyuan/xCPD].

## 1 INTRODUCTION

Multivariate time series forecasting (MTSF), a fundamental task in time series analysis, has emerged as a key focus in artificial intelligence due to its broad applications in transportation, finance, energy, and environment. Early efforts primarily focused on adapting a variety of model architectures—ranging from linear models (Zeng et al., 2023; Li et al., 2023) and convolutional networks (Liu et al., 2022a; Wu et al., 2023) to Transformers (Nie et al., 2023; Liu et al., 2024a)—for this task, establishing a strong foundation for the field. Whatever the model architecture, the key to success lies in accurate modeling of both temporal dynamics and inter-variable (*a.k.a.*, channel) dependencies. Building on this insight, recent works have been focusing on the channel strategy, giving rise to three distinct time series modeling paradigms: Channel Independence (CI), Channel Dependence (CD), and Channel Partiality (CP). CI methods (Zeng et al., 2023; Zhou et al., 2021) model each channel separately, capturing only intra-channel temporal dynamics while completely ignoring inter-channel interactions. This design enhances robustness and scalability but often overlooks valuable inter-channel patterns. CD methods (Zhang & Yan, 2023; Liu et al., 2024a), on the other hand, jointly model all channels to capture global dependencies, but may suffer from oversmoothing and noise propagation from heterogeneous or weakly correlated variables. CP methods (Qiu et al., 2024; Chen et al., 2024; Hu et al., 2025b) attempt to balance CI and CD by selectively modeling inter-channel relationships, adaptively adjusting the dependency structure based on contextual or learned relevance.

---

*Corresponding Author

Despite their success, existing CP methods fundamentally struggle to capture fine-grained, patch-level, and frequency-decoupled channel dependencies. **First**, current CP approaches operate primarily at the channel level, either by grouping entire channels via clustering (Qiu et al., 2024; Chen et al., 2024) or learning channel-wise attention weights (Hu et al., 2025b; Lee et al., 2025). This reliance on the entire channel embedding as the relational unit introduces a coarse-granularity bottleneck: they fail to model localized patch-level interactions or capture nuanced segment-specific patterns. For example, if Channel-A exhibits a smooth seasonal trend during segment $T_1$ but a sharp anomaly during segment $T_2$, channel-level models generate a single averaged dependency weight for the entire channel, failing to capture the distinct local interactions that trend and anomaly segments should establish with other channels. **Second**, while CD/CP models produce time-varying attention weights, these interactions are computed entirely in the temporal domain, where low-, mid-, and high-frequency components are mixed within a single embedding (Huang et al., 2023; Donghao & Xue, 2024; Zhao & Shen, 2024). As a result, a high attention score between two channels may simultaneously reflect a meaningful low-frequency seasonal dependence and an irrelevant high-frequency noise component, without distinguishing between them. This frequency coupling forces the model to treat heterogeneous spectral behaviors as a single dependency signal, often introducing spurious correlations—especially when high-frequency noise and low-frequency trends coexist. Consequently, temporal-domain dynamism alone is insufficient for capturing fine-grained and frequency-specific dependency evolution, leaving important dynamic cross-channel relationships entangled.

To address these fundamental limitations, we reformulate the basic modeling unit from an entire channel to a channel-patch, a localized temporal segment within a channel, following the widely adopted patching paradigm in modern sequence modeling. We represent each channel-patch as a node in a graph, where edges denote the interactions (dependencies) between channel-patch pairs. This novel graph perspective enables a unified and highly granular treatment of **channel-patch dependencies (CPD)**, encompassing inter-channel patch, intra-channel patch, and patch-level channel interactions. Building on this foundation, we propose xCPD where "x" denotes graph spectral decomposition for Channel-Patch Dependencies. xCPD is a generic and lightweight plugin that enhances existing forecasting models by adaptively routing CPD through the lens of graph spectral decomposition. Unlike prior methods, xCPD operates exclusively in the spectral domain, where it leverages a shared graph Fourier basis to embed channel-patch representations and subsequently groups them by their spectral energy responses. This design enables the model to distinguish between smooth, moderately varying, and highly fluctuating patterns (*i.e.*, low-, mid-, and high-frequency). A dynamic Mixture-of-Experts routing mechanism further adaptively selects frequency-specific filters for each patch, enabling xCPD to capture time-varying dependencies while suppressing irrelevant frequency-coupled correlations. Finally, xCPD is fully model-agnostic and plug-and-play, allowing seamless integration into a wide range of CI and CD backbones, including large foundation forecasters, without retraining. This provides a practical and scalable solution for real-world forecasting pipelines.

## 2 RELATED WORK

**Multivariate Time Series Forecasting (MTSF).** The core challenge of MTSF lies in jointly modeling inter-channel dependencies and temporal dynamics (Li et al., 2023; Shang et al., 2024). Early approaches like ARIMA (Zhang, 2003) and Prophet (Triebe et al., 2021) focus exclusively on temporal patterns within individual series, lacking a mechanism for inter-channel modeling. To address this, deep learning models have been widely explored (Zhou et al., 2022a; Yi et al., 2023b; Zhang et al., 2024). Specifically, **RNNs** (Hewamalage et al., 2021; Lin et al., 2023) capture sequential patterns through recursive hidden states but suffer from vanishing gradients when modeling long-term dependencies. **CNNs** (Wu et al., 2023; Dai et al., 2024) offer efficient extraction of local features through convolutions, but their limited receptive fields constrain global context understanding. To overcome these limitations, **Transformers** (Wu et al., 2021; Zhou et al., 2022b; Liu et al., 2022b; Feng et al., 2024; Liu et al., 2025) use self-attention mechanisms to capture long-range dependencies, but their permutation invariance can compromise the temporal order, which is crucial for forecasting. Recently, **MLPs** (Zeng et al., 2023; Chen et al., 2023; Das et al., 2023; Hu et al., 2025a; Yu et al., 2024) have shown that simple dense connections across timesteps can achieve competitive performance without architectural complexity. Beyond temporal dynamics, **GNNs** (Yi et al., 2023a; Yang et al., 2023; Wang et al., 2024b; Cai et al., 2024) model inter-channel dependencies by treating multivariate series as graphs, capturing structured cross-variable dependencies more effectively.

**Channel Dependency Strategy.** In MTSF, dependency modeling strategies fall into three categories: **CI methods** (Zhou et al., 2021; Jin et al., 2024; Liu et al., 2024b) treat channels independently, capturing only intra-channel temporal patterns. While this design enhances robustness by avoiding inter-channel noise, it fails to use valuable inter-channel relationships. **CD methods** (Liu et al., 2024a; Woo et al., 2024; Gao et al., 2024) jointly model all channels to exploit inter-channel dependencies, but introduce spurious correlations that degrade performance in noisy or heterogeneous settings. **CP methods** (Huang et al., 2023; Donghao & Xue, 2024; Zhao & Shen, 2024; Lee et al., 2025) balance CI and CD strategies by selectively modeling inter-channel interactions based on intrinsic channel relationships. Recent approaches like DUET (Qiu et al., 2024) and CCM (Chen et al., 2024) employ clustering or sparse attention to adaptively identify relevant dependencies. However, although recent methods such as TimeFilter (Hu et al., 2025b) and PCD (Lee et al., 2025) introduce time-varying and even patch-level interactions, they still operate entirely in the temporal domain. As a result, their dependency scores remain frequency-coupled: a single attention mixes low-frequency trends, mid-frequency variations, and high-frequency noise into one unified correlation signal. This prevents the model from distinguishing which frequency component is responsible for the observed interaction, limiting its ability to capture frequency-aware and fine-grained patch-level dependency patterns that evolve over time. In contrast, xCPD performs dependency modeling in the spectral domain at the channel-patch level, enabling fine-grained and interpretable modeling of frequency-specific interactions that evolve over time. Specifically, xCPD differentiates itself from prior CP methods across three critical dimensions: **modeling granularity** (channel-patch vs. channel), **modeling domain** (spectral vs. temporal), and **adaptivity** (frequency-specific routing vs. mixed attention). Furthermore, as a lightweight and fully model-agnostic plugin, xCPD can be seamlessly integrated into a wide range of forecasting architectures without any backbone retraining, substantially enhancing its practical efficiency, deployment flexibility, and scalability in real-world forecasting systems.

## 3 PRELIMINARIES

**Multivariate Time Series Forecasting**. Let $\boldsymbol{X} = [\boldsymbol{x}_1, \ldots, \boldsymbol{x}_C] \in \mathbb{R}^{C \times T}$ denote a multivariate time series, where $C$ represents the number of channels (*i.e.*, variates) and $T$ denotes the lookback horizon. MTSF aims to build a predictive model $f_{\text{model}} \colon \mathbb{R}^{C \times T} \to \mathbb{R}^{C \times T'}$ that takes historical observations $\boldsymbol{X}$ as input and generates future predictions $\widehat{\boldsymbol{X}} = [\hat{\boldsymbol{x}}_1, \ldots, \hat{\boldsymbol{x}}_C] \in \mathbb{R}^{C \times T'}$, where $T'$ is the forecasting horizon. In this study, rather than building $f_{\text{model}}$, we develop a generic plugin model $f_{\text{plugin}} \colon \mathbb{R}^{C \times T'} \to \mathbb{R}^{C \times T'}$ that post-processes the predictions from existing models. Given a base model's output $\widehat{\boldsymbol{X}}^{\text{model}} = f_{\text{model}}(\boldsymbol{X})$, our plugin refines predictions to obtain improved forecasts $\widehat{\boldsymbol{X}} = f_{\text{plugin}}(\widehat{\boldsymbol{X}}^{\text{model}})$.

**Graph Signal Processing**. Given a graph $\mathcal{G} = (\mathcal{V}, \mathcal{E})$ with node set $\mathcal{V}$ and edge set $\mathcal{E}$, let $\boldsymbol{A}$ and $\boldsymbol{D}$ denote the adjacency matrix and the degree matrix, respectively. The normalized Laplacian graph is defined as $\boldsymbol{L} = \boldsymbol{I} - \boldsymbol{D}^{-1/2} \boldsymbol{A} \boldsymbol{D}^{-1/2}$. Its eigendecomposition is $\boldsymbol{L} = \boldsymbol{U} \boldsymbol{\Lambda} \boldsymbol{U}^{\top}$, where $\boldsymbol{U}$ represents the graph Fourier basis, and $\boldsymbol{\Lambda} = \text{diag}(\lambda_1, \ldots, \lambda_n)$ is a diagonal matrix containing the eigenvalues.

## 4 METHODOLOGY

We propose xCPD, a lightweight plugin module designed to enhance multivariate forecasting models. As shown in Figure 1, xCPD comprises three key components: Spectral Channel-Patch Embedding (Section 4.1), Channel-Patch Grouping (Section 4.2), and Channel-Patch Routing (Section 4.3).

### 4.1 SPECTRAL CHANNEL-PATCH EMBEDDING

As shown in Figure 1(A), we first construct a channel-patch graph to model fine-grained patch-level dependencies, where each patch from every channel forms a "channel-patch" node. We then apply graph spectral decomposition to obtain spectral embeddings that capture spectral dependencies.

**Channel-Patch as Node**. We first partition the output $\widehat{\boldsymbol{X}}^{\text{model}} \in \mathbb{R}^{C \times T'}$ from a time series model into contiguous, non-overlapping temporal patches $\boldsymbol{X}^{\text{patch}} \in \mathbb{R}^{C \times (N \times P)}$, where $N = \lceil \frac{T'}{P} \rceil$ is the number of patches, and $P$ is the patch length. Subsequently, we apply $\text{Linear}(\cdot)$ to transform each patch from length $P$ to a hidden dimension $d$, obtaining the embedded tokens $\boldsymbol{X}^{\text{emb}} \in \mathbb{R}^{C \times N \times d}$ as:

$$\boldsymbol{X}^{\text{patch}} = \text{Patching}(\widehat{\boldsymbol{X}}^{\text{model}}), \quad \boldsymbol{X}^{\text{emb}} = \text{Linear}(\boldsymbol{X}^{\text{patch}}) \tag{1}$$

By flattening the dimensions to $n = C \times N$, we obtain channel-patch node embeddings as $\boldsymbol{X}^{\text{emb}} \in \mathbb{R}^{n \times d}$. For the embedding matrix of the $t$-th input $\boldsymbol{X}^{\text{emb},t}$, we construct a weighted similarity

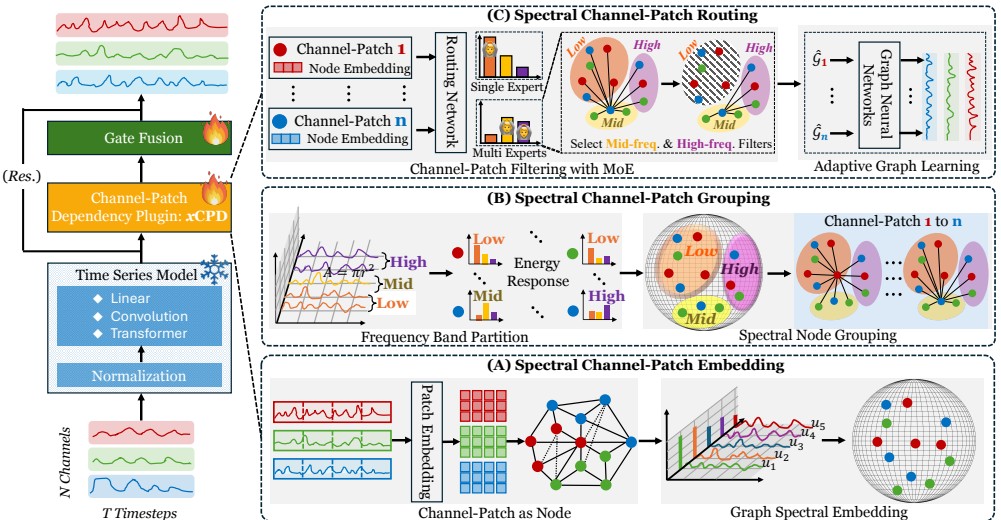

Figure 1: The proposed xCPD plugin consists of three modules: (A) spectral channel-patch embedding, (B) channel-patch grouping, and (C) channel-patch routing with Mixture-of-Experts (MoE).

graph using a *cosine-similarity* formulation: $A_{ij}^t = \cos(\boldsymbol{X}_i^{\text{emb},t}, \boldsymbol{X}_j^{\text{emb},t})$. Cosine similarity yields affinity values bounded within $[-1, 1]$ and automatically introduces self-connections through $A_{ii}^t = 1$. Because it is invariant to scaling, the resulting graph is unaffected by magnitude differences across channels, which is especially important in MTSF where variables operate at different physical scales.

**Graph Spectral Embedding**. Time series channels often exhibit complex dependencies that are more discernible in the spectral domain, where correlated channels tend to share similar spectral patterns (Cao et al., 2020). Motivated by this, we learn spectral embeddings for channel-patch nodes to effectively capture their underlying spectral dependencies. However, batch-wise spectral decomposition yields inconsistent Fourier bases across batches, resulting in incomparable embeddings (Streicher et al., 2023). To overcome this, we introduce a shared graph Fourier basis that ensures a consistent spectral domain across all timesteps. This shared basis is learned to closely approximate the eigenbasis of individual batches, minimizing their differences as bounded in Theorem 4.1.

**Theorem 4.1** (Shared Graph Fourier Basis). *Let $\boldsymbol{L}^t$ be normalized Laplacian graph derived from matrix $\boldsymbol{A}^t$ at the $t$-th timestep where $t \in \{1,\ldots,T'\}$. Define the average Laplacian as $\boldsymbol{L}_{avg} = \frac{1}{T'}\sum_{t=1}^{T'}\boldsymbol{L}^t$, with eigen-decomposition $\boldsymbol{L}_{avg} = \boldsymbol{U}\boldsymbol{\Lambda}\boldsymbol{U}^\top$. $\boldsymbol{U}$ denotes the shared Fourier basis that approximates each individual eigenbasis $\boldsymbol{U}^t$ from the decomposition of $\boldsymbol{L}^t$, satisfying $\|\boldsymbol{U}^t - \boldsymbol{U}\mathbf{R}^t\|_F \leq \mathbf{C}\|\boldsymbol{L}^t - \boldsymbol{L}_{avg}\|_F$, where $\mathbf{C}$ depends on the eigen-gap of $\boldsymbol{L}_{avg}$ and $\mathbf{R}^t$ is an orthogonal rotation matrix.*

Theorem 4.1 establishes the theoretical foundation for our shared Fourier basis, ensuring that all batches are mapped to a consistent and comparable spectral domain. Importantly, because the basis is learned from the normalized Laplacian, whose eigenvalues lie within the bounded interval [0,2], the resulting spectral operator is both scale-invariant and numerically stable across timesteps. Following Theorem 4.1, we construct the shared basis $\boldsymbol{U}$ and compute spectral embeddings as:

$$\boldsymbol{X}^{\text{spc},t} = \mathcal{F}(\boldsymbol{X}^{\text{emb},t}) = \boldsymbol{U}^\top \boldsymbol{X}^{\text{emb},t} \in \mathbb{R}^{n \times d}, \tag{2}$$

where $\mathcal{F}(\cdot)$ denotes the graph Fourier transformation, and $\boldsymbol{X}^{\text{spc},t}$ denotes nodes spectral embeddings. For notational simplicity, we omit the time index $t$ hereafter, e.g., we write $\boldsymbol{X}^{\text{spc}}$ instead of $\boldsymbol{X}^{\text{spc},t}$.

## 4.2 SPECTRAL CHANNEL-PATCH GROUPING

As shown in Figure 1(B), we develop a spectral grouping mechanism that classifies nodes by their dominant frequencies (low/mid/high) through energy analysis. We then construct ego-graphs connecting nodes with similar frequency properties to capture meaningful frequency-aware dependencies.

**Frequency Band Partition**. We employ learnable parameters $\tau_1, \tau_2 \in [1, n]$ with $\tau_1 < \tau_2$ to define adaptive boundaries. To ensure differentiability, we compute soft membership weights as follows:

$$\alpha_j^{\text{low}} = \text{Sigmoid}(m(\tau_1 - j)), \ \alpha_j^{\text{high}} = \text{Sigmoid}(m(j - \tau_2)), \ \alpha_j^{\text{mid}} = 1 - \alpha_j^{\text{low}} - \alpha_j^{\text{high}}, \quad (3)$$

where $j \in \{1, \dots, n\}$ indexes frequencies and temperature $m > 0$ controls sharpness. $\alpha_j \in [0, 1]$ form a normalized convex partition, ensuring stable band membership for each frequency component.

**Spectral Node Grouping**. $\boldsymbol{X}^{\text{spc}}$ captures feature-level frequency responses (Cao et al., 2020), but it does not directly reflect node-specific frequency intensities. We address this by defining a spectral energy function to quantify how strongly each node responds to different frequency components as:

**Theorem 4.2** (Spectral Energy Response). *Given spectral node embeddings $\boldsymbol{X}^{\text{spc}}$ and graph Fourier basis $\boldsymbol{U}$, spectral energy response of node $i$ for frequency $j$ is $\boldsymbol{S}_{i,j} = \|\boldsymbol{U}_{i,j} \cdot \boldsymbol{X}_{j,:}^{\text{spc}}\|_2^2$. The orthonormality of $\boldsymbol{U}$ ensures energy preservation between spatial and spectral domains: $\sum_{j=1}^{n} \boldsymbol{S}_{i,j} = \|\boldsymbol{X}_{i,:}^{\text{emb}}\|_2^2$, confirming spectral energy response retains all information from original channel-patch embeddings.*

Based on Theorem 4.2, we classify the nodes by their maximum spectral energy response as:

$$\text{Group}(i) = \underset{b \in \{0,1,2\}}{\arg\max} \frac{\exp(\sum_{j=1}^{n} \alpha_j^b \cdot \boldsymbol{S}_{i,j})}{\sum_{b' \in \{0,1,2\}} \exp(\sum_{j=1}^{n} \alpha_j^{b'} \cdot \boldsymbol{S}_{i,j})}, \ \{0, 1, 2\} \text{ denotes } \{\text{low}, \text{mid}, \text{high}\}, \quad (4)$$

where $\alpha_j^b$ are the soft membership weights and $\boldsymbol{S}_{i,j}$ is the spectral energy response of Theorem 4.2.

Following Huang et al. (2023), we decompose the graph $\mathcal{G} = (\mathcal{V}, \mathcal{E})$ into $n$ ego-graphs $\{\mathcal{G}_i\}_{i=1}^{n}$ to reduce noise. Each ego-graph $\mathcal{G}_i = (\mathcal{V}, \mathcal{E}_i)$ is centered on node $i$. We construct $i$-th ego-graph adjacency matrices $\boldsymbol{A}^{(i)}$ using the $k$-Nearest Neighbor ($k$-NN) method on similarity matrix as:

$$\boldsymbol{A} = k\text{-NN}\big(\text{Norm}(\boldsymbol{X}^{\text{emb}}(\boldsymbol{X}^{\text{emb}})^\top), \ \alpha\big), \quad \boldsymbol{A}_{pq}^{(i)} = \begin{cases} \boldsymbol{A}_{pq}, & \text{if } p = i \text{ or } q = i, \\ 0, & \text{otherwise}, \end{cases} \quad (5)$$

where $p, q \in \{1, \dots, n\}$, $\alpha$ is a hand-tuned scaling factor that determines the number of neighbors $k = \lfloor \alpha \cdot n \rfloor$, and $\text{Norm}(\cdot)$ applies row-wise normalization. Within each ego-graph $\mathcal{G}_i$, we create frequency-specific subgraphs by grouping neighboring nodes based on their spectral labels, enabling frequency-aware message passing, as illustrated in Figure 1.

## 4.3 SPECTRAL CHANNEL-PATCH ROUTING

As shown in Figure 1(C), we introduce a Mixture of Experts (MoE) routing module that dynamically selects frequency-specific filters for each node to capture diverse spectral dependencies. This adaptive selection enables tailored graph learning of channel-patch relationships.

**Channel-Patch Filtering with MoE**. Different time series exhibit heterogeneous spectral dependencies. To address this, our routing network assigns each ego-graph to one of three specialized frequency filters, which construct adjacency matrices by emphasizing relevant spectral components while suppressing irrelevant dependencies. We design three specialized filters as follows:

- *Low-Frequency Filter* constructs adjacency matrices using low-frequency spectral components, capturing smooth, slowly evolving patterns and long-term regularities in time series, such as seasonal cycles and periodic trends (Yi et al., 2024).
- *Mid-Frequency Filter* constructs adjacency matrices using mid-frequency spectral components, modeling moderate dynamics and local fluctuations in semi-stationary processes, such as shifting traffic patterns and regional weather transitions (Yu et al., 2025).
- *High-Frequency Filter* constructs adjacency matrices using high-frequency spectral components, detecting rapid changes and discontinuities in volatile processes, such as system anomalies, market disruptions, and abrupt state transitions (Galib et al., 2024).

Unlike classical MoE with fixed Top-$K$ expert assignment, our Dynamic MoE (DyMoE) selects a variable number of experts per input. For each ego-graph $\mathcal{G}_i$, we compute routing scores for filters as:

$$\psi(\boldsymbol{x}_i) = \text{Linear}_c(\boldsymbol{x}_i) + \epsilon \cdot \text{Softplus}(\text{Linear}_n(\boldsymbol{x}_i)) \in \mathbb{R}^3, \quad (6)$$

where $\boldsymbol{x}_i$ is the $i$-th row of $\boldsymbol{X}^{\text{emb}}$, $\text{Linear}_c$ and $\text{Linear}_n$ compute deterministic and stochastic components, respectively, $\epsilon$ is a noise scale parameter. DyMoE sorts $\psi(\boldsymbol{x}_i)$ in descending order and selects the minimum number of top-ranked filters whose cumulative probability exceeds the threshold $\tau$:

$$s = \min \left\{ \ell \in \{1, 2, 3\} : \sum_{g=1}^{\ell} \boldsymbol{p}_m \geq \tau \right\}, \quad \boldsymbol{p} = \text{Softmax}(\psi(\boldsymbol{x}_i)), \tag{7}$$

where $\boldsymbol{p}_g$ denotes the $g$-th largest element. For each ego-graph $\mathcal{G}_i$, this produces an adaptive expert set $\Psi(\mathcal{G}_i) = \{\text{Filter}_q \mid 1 \leq q \leq s\}$, where $s \in \{1, 2, 3\}$ denotes the number of selected experts.

**Adaptive Graph Learning**. Using the selected filters $\Psi(\mathcal{G}_i)$ from DyMoE, we construct a sparse adjacency matrix $\boldsymbol{A}^i$ for each ego-graph $\mathcal{G}_i$ by filtering irrelevant dependencies (edges) as:

$$\mathcal{G}_i = \{\mathcal{V}, \mathcal{E}_i = \bigcup_{q=1}^{s} \text{Filter}_q(\mathcal{G}_i)\}, \quad \boldsymbol{A}^i = k\text{-NN}(\mathcal{G}_i), \tag{8}$$

where $\text{Filter}_q(\mathcal{G}_i)$ extracts edges based on specific frequency bands (low, mid, or high), and $k$-NN constructs a sparse adjacency matrix by retaining only the $k$ nearest neighbors for each central node. Then, the graph learning layer adaptively captures the hidden relationships among time series as:

$$\boldsymbol{h}_i^{(\ell)} = \text{COMB}(\boldsymbol{h}_i^{(\ell-1)}, \text{AGGR}\{\boldsymbol{h}_j^{(\ell-1)} : v_j \in \mathcal{N}(v_i)\}), \tag{9}$$

where $\ell \in \{1, \ldots, L\}$ denotes the layer index, $\mathcal{N}(v_i)$ denotes the neighborhood of node $v_i$, $\boldsymbol{h}_i^{(0)}$ corresponds to the $i$-th row of $\boldsymbol{X}^{\text{emb}}$, and $\boldsymbol{h}_i^{(\ell)}$ denotes the embedding of node $v_i$ at the $\ell$-th layer. $\text{AGGR}(\cdot)$ and $\text{COMB}(\cdot)$ are the functions used to aggregate neighborhood information and combine ego- and neighbor-embeddings Xia et al. (2026). After $L$ layers of propagation, we obtain the final node embeddings as:

$$\boldsymbol{H}^{(L)} = [\hat{\boldsymbol{h}}_1^{(L)}, \ldots, \hat{\boldsymbol{h}}_n^{(L)}]^\top \in \mathbb{R}^{C \times T'}, \quad \text{where} \quad \hat{\boldsymbol{h}}_i^{(L)} = \text{Linear}(\boldsymbol{h}_i^{(L)}). \tag{10}$$

## 4.4 PREDICTION AND OPTIMIZATION

**Prediction**. xCPD produces the final prediction via a gated dual-path residual correction:

$$\widehat{\boldsymbol{X}}^{\text{predict}} = \widehat{\boldsymbol{X}}^{\text{model}} + \sigma(\boldsymbol{g}_{\text{GNN}}) \odot \delta_{\text{GNN}} + \sigma(\boldsymbol{g}_{\text{Lin}}) \odot \delta_{\text{Lin}} \tag{11}$$

where $\delta_{\text{GNN}} = W_{\text{proj}} \boldsymbol{H}^{(L)}$ is the GNN-path correction capturing cross-variable spectral dependencies, $\delta_{\text{Lin}} = f_{\text{lin}}(\widehat{\boldsymbol{X}}^{\text{model}})$ is the Linear-path correction preserving CI refinement, and $\boldsymbol{g}_{\text{GNN}}, \boldsymbol{g}_{\text{Lin}} \in \mathbb{R}^C$ are learnable per-variable gate parameters with $\sigma(\cdot)$ denoting the sigmoid. This residual formulation ensures that when gate values approach zero, xCPD degrades to the backbone prediction $\widehat{\boldsymbol{X}}^{\text{model}}$.

**Optimization**. The training objective of xCPD comprises three components. First, we minimize the Mean Squared Error (MSE) loss $\mathcal{L}_{\text{MSE}}$ between predictions $\widehat{\boldsymbol{X}}^{\text{predict}}$ and the ground truth $\widehat{\boldsymbol{X}}$ as:

$$\mathcal{L}_{\text{MSE}} = \frac{1}{T'} \sum_{t=1}^{T'} \|\widehat{\boldsymbol{X}}_{:,t}^{\text{predict}} - \widehat{\boldsymbol{X}}_{:,t}\|_2^2. \tag{12}$$

To encourage diverse expert usage, we add two regularization terms. Let $\boldsymbol{R}_{i,j} = \text{Softmax}(\psi(\boldsymbol{x}_i))_j$ denote the routing probability of the expert $j$ for node $i$. We define the entropy loss $\mathcal{L}_{\text{Entropy}}$ to prevent all experts from having low confidence scores (Huang et al., 2024), and the balance loss $\mathcal{L}_{\text{Balance}}$ to prevent the model from using only a few experts repeatedly (Shazeer et al., 2017):

$$\mathcal{L}_{\text{Entropy}} = -\frac{1}{n} \sum_{i=1}^{n} \sum_{j=1}^{3} (\boldsymbol{R}_{i,j} \log(\boldsymbol{R}_{i,j} + \delta)), \quad \mathcal{L}_{\text{Balance}} = \frac{1}{n} \sum_{i=1}^{n} \frac{\text{Std}(\boldsymbol{R}_{i,:})}{\text{Mean}(\boldsymbol{R}_{i,:}) + \delta}, \tag{13}$$

where $\delta$ is a small constant for numerical stability, $\text{Std}(\cdot)$ and $\text{Mean}(\cdot)$ calculate the standard deviation and mean, respectively. Then, the overall training objective can be formulated as follows:

$$\mathcal{L} = \mathcal{L}_{\text{MSE}} + \mu \mathcal{L}_{\text{Entropy}} + \beta \mathcal{L}_{\text{Balance}}, \tag{14}$$

where hyperparameters $\mu$ and $\beta$ balance the regularization terms.

**Complexity**. xCPD has a time complexity of $\mathcal{O}(nkd + Lnkd)$ and a space complexity of $\mathcal{O}(nd + nk)$, where $n$ is the number of nodes, $k$ is the number of neighbors, $d$ is the embedding size, and $L$ is the GNN layers. These lightweight operations ensure scalability to large multivariate time series.

Table 1: Long-term forecasting results on 9 real-world datasets in terms of MSE and MAE. The lookback horizon is set 96 (512 for TSMixer) and forecasting horizons are {96, 192, 336, 720}. The better performance in each setting is shown in **bold**. The best results for each row are underlined.

| Model | TSMixer | | + xCPD | | DLinear | | + xCPD | | PatchTST | | + xCPD | | TimesNet | | + xCPD | |
|---|---|---|---|---|---|---|---|---|---|---|---|---|---|---|---|---|
| Metric | MSE | MAE | MSE | MAE | MSE | MAE | MSE | MAE | MSE | MAE | MSE | MAE | MSE | MAE | MSE | MAE |
| **ETTh1** 96 | 0.361 | 0.392 | **0.357** | **0.388** | 0.384 | 0.397 | **0.381** | **0.392** | 0.414 | 0.419 | **0.402** | **0.404** | 0.384 | 0.402 | **0.378** | **0.398** |
| 192 | 0.404 | 0.418 | **0.389** | **0.405** | 0.433 | 0.427 | **0.430** | **0.421** | 0.460 | 0.445 | **0.443** | **0.435** | 0.436 | 0.429 | **0.422** | **0.423** |
| 336 | 0.420 | 0.431 | **0.402** | **0.411** | 0.479 | 0.456 | **0.472** | **0.444** | 0.501 | 0.466 | **0.482** | **0.451** | 0.491 | 0.469 | **0.478** | **0.456** |
| 720 | 0.463 | 0.472 | **0.458** | **0.461** | 0.506 | 0.503 | **0.497** | **0.485** | 0.500 | 0.488 | **0.494** | **0.482** | 0.521 | 0.500 | **0.509** | **0.491** |
| **ETTh2** 96 | 0.274 | 0.341 | **0.270** | **0.336** | 0.314 | 0.369 | **0.302** | **0.357** | 0.302 | 0.348 | **0.274** | **0.335** | 0.340 | 0.374 | **0.330** | **0.368** |
| 192 | 0.339 | 0.385 | **0.321** | **0.381** | 0.426 | 0.439 | **0.414** | **0.428** | 0.388 | 0.400 | **0.372** | **0.385** | 0.402 | 0.414 | **0.389** | **0.402** |
| 336 | 0.361 | 0.406 | **0.358** | **0.392** | 0.576 | 0.530 | **0.547** | **0.511** | 0.426 | 0.433 | **0.408** | **0.420** | 0.452 | 0.452 | **0.435** | **0.439** |
| 720 | 0.445 | 0.470 | **0.430** | **0.458** | 0.795 | 0.641 | **0.765** | **0.623** | 0.431 | 0.446 | **0.417** | **0.433** | 0.462 | 0.468 | **0.448** | **0.455** |
| **ETTm1** 96 | 0.285 | 0.339 | **0.280** | **0.334** | 0.345 | 0.373 | **0.329** | **0.362** | 0.329 | 0.367 | **0.318** | **0.358** | 0.338 | 0.375 | **0.329** | **0.367** |
| 192 | 0.327 | 0.365 | **0.321** | **0.360** | 0.381 | 0.391 | **0.371** | **0.381** | 0.367 | 0.385 | **0.352** | **0.374** | 0.374 | 0.387 | **0.368** | **0.376** |
| 336 | 0.356 | 0.382 | **0.354** | **0.378** | 0.415 | 0.415 | **0.401** | **0.403** | 0.399 | 0.410 | **0.380** | **0.394** | 0.410 | 0.411 | **0.408** | **0.410** |
| 720 | 0.419 | 0.414 | **0.417** | **0.413** | 0.473 | 0.450 | **0.457** | **0.436** | 0.454 | 0.439 | **0.437** | **0.428** | 0.478 | 0.450 | **0.467** | **0.441** |
| **ETTm2** 96 | 0.163 | 0.252 | **0.160** | **0.250** | 0.193 | 0.292 | **0.188** | **0.284** | 0.175 | 0.259 | **0.163** | **0.250** | 0.187 | 0.267 | **0.185** | **0.266** |
| 192 | 0.216 | 0.290 | **0.212** | **0.288** | 0.284 | 0.361 | **0.276** | **0.351** | 0.241 | 0.302 | **0.230** | **0.297** | 0.249 | 0.309 | **0.245** | **0.303** |
| 336 | 0.268 | **0.324** | **0.266** | **0.324** | 0.384 | 0.429 | **0.376** | **0.418** | 0.305 | 0.343 | **0.281** | **0.334** | 0.321 | 0.351 | **0.313** | **0.344** |
| 720 | 0.420 | 0.422 | **0.366** | **0.387** | 0.556 | 0.523 | **0.522** | **0.501** | 0.402 | 0.400 | **0.388** | **0.390** | 0.408 | 0.403 | **0.388** | **0.385** |
| **Exchange** 96 | 0.089 | 0.209 | **0.083** | **0.203** | 0.083 | 0.206 | **0.081** | **0.202** | 0.088 | 0.205 | **0.083** | **0.201** | 0.107 | 0.234 | **0.103** | **0.228** |
| 192 | 0.195 | 0.315 | **0.173** | **0.296** | 0.153 | 0.286 | **0.150** | **0.283** | 0.176 | 0.299 | **0.168** | **0.294** | 0.226 | 0.344 | **0.221** | **0.335** |
| 336 | 0.343 | 0.421 | **0.304** | **0.399** | 0.289 | 0.404 | **0.266** | **0.386** | 0.301 | 0.397 | **0.297** | **0.392** | 0.367 | 0.448 | **0.356** | **0.439** |
| 720 | 0.898 | 0.710 | **0.835** | **0.685** | 0.793 | 0.671 | **0.751** | **0.651** | 0.901 | 0.714 | **0.885** | **0.702** | 0.964 | 0.746 | **0.948** | **0.725** |
| **Solar-Eng.** 96 | 0.201 | 0.235 | **0.196** | **0.230** | 0.290 | 0.378 | **0.284** | **0.372** | 0.234 | 0.286 | **0.229** | **0.280** | 0.250 | 0.292 | **0.244** | **0.287** |
| 192 | 0.231 | 0.258 | **0.227** | **0.259** | 0.320 | 0.398 | **0.312** | **0.389** | 0.267 | 0.310 | **0.260** | **0.302** | 0.296 | 0.318 | **0.289** | **0.314** |
| 336 | 0.246 | 0.271 | **0.245** | **0.277** | 0.353 | 0.415 | **0.344** | **0.406** | 0.290 | 0.315 | **0.281** | **0.307** | 0.319 | 0.330 | **0.313** | **0.322** |
| 720 | 0.247 | 0.273 | **0.245** | **0.272** | 0.356 | 0.413 | **0.348** | **0.407** | 0.289 | 0.317 | **0.283** | **0.312** | 0.338 | 0.337 | **0.328** | **0.330** |
| **Weather** 96 | 0.149 | 0.198 | **0.143** | **0.192** | 0.194 | 0.252 | **0.177** | **0.239** | 0.177 | 0.218 | **0.167** | **0.218** | 0.172 | 0.220 | **0.164** | **0.212** |
| 192 | 0.201 | 0.248 | **0.190** | **0.240** | 0.238 | 0.299 | **0.227** | **0.280** | 0.225 | 0.259 | **0.218** | **0.242** | 0.219 | 0.261 | **0.209** | **0.250** |
| 336 | 0.264 | 0.291 | **0.240** | **0.277** | 0.281 | 0.330 | **0.272** | **0.317** | 0.278 | 0.297 | **0.270** | **0.288** | 0.280 | 0.306 | **0.270** | **0.285** |
| 720 | 0.320 | 0.336 | **0.313** | **0.330** | 0.345 | 0.381 | **0.337** | **0.370** | 0.354 | 0.348 | **0.338** | **0.341** | 0.365 | 0.359 | **0.361** | **0.357** |
| **Electricity** 96 | 0.142 | 0.237 | **0.135** | **0.230** | 0.194 | 0.277 | **0.174** | **0.262** | 0.181 | 0.273 | **0.170** | **0.261** | 0.168 | 0.272 | **0.153** | **0.254** |
| 192 | 0.154 | 0.248 | **0.144** | **0.242** | 0.194 | 0.281 | **0.180** | **0.268** | 0.188 | 0.279 | **0.176** | **0.265** | 0.184 | 0.289 | **0.170** | **0.258** |
| 336 | 0.163 | 0.264 | **0.156** | **0.257** | 0.207 | 0.296 | **0.195** | **0.293** | 0.204 | 0.296 | **0.196** | **0.287** | 0.198 | 0.300 | **0.178** | **0.280** |
| 720 | 0.208 | 0.300 | **0.200** | **0.294** | 0.242 | 0.330 | **0.240** | **0.326** | 0.246 | 0.328 | **0.233** | **0.308** | 0.220 | 0.320 | **0.198** | **0.303** |
| **Traffic** 96 | 0.376 | 0.264 | **0.371** | **0.259** | 0.650 | 0.397 | **0.636** | **0.386** | 0.462 | 0.298 | **0.448** | **0.278** | 0.593 | 0.321 | **0.549** | **0.303** |
| 192 | 0.397 | 0.277 | **0.358** | **0.275** | 0.599 | 0.371 | **0.574** | **0.356** | 0.468 | 0.300 | **0.453** | **0.291** | 0.617 | 0.336 | **0.548** | **0.325** |
| 336 | 0.413 | 0.290 | **0.410** | **0.287** | 0.605 | 0.374 | **0.586** | **0.362** | 0.482 | 0.307 | **0.464** | **0.295** | 0.629 | 0.336 | **0.561** | **0.332** |
| 720 | 0.444 | 0.306 | **0.439** | **0.303** | 0.645 | 0.395 | **0.630** | **0.387** | 0.517 | 0.326 | **0.502** | **0.305** | 0.640 | 0.350 | **0.573** | **0.344** |

## 5 EXPERIMENTS

**Benchmarking Datasets**. For long-term forecasting, we experiment with 9 popular benchmarks in various domains (Zhou et al., 2021; Wu et al., 2021; Lai et al., 2018), including weather, electricity, and traffic. For short-term forecasting, we use two univariate datasets: M4 (Makridakis et al., 2018), covering diverse domains and frequencies, and Stock (Chen et al., 2024), containing 1,390 stock price series over 10 years with high variability and non-stationarity. Details are provided in Appendix B.

**Backbones and Baselines**. xCPD is a model-agnostic strategy that can be seamlessly integrated into arbitrary forecasting models to improve their performance. To ensure comprehensive evaluation, we select four recent state-of-the-art forecasting backbones: two Channel-Independent (CI) models—DLinear (Zeng et al., 2023) and PatchTST (Nie et al., 2023)—and two Channel-Dependent (CD) models—TSMixer (Chen et al., 2023) and TimesNet (Wu et al., 2023). These backbones span three representative architectural paradigms: linear, transformer-based, and convolution-based models. In addition, we compare xCPD against four best-performing methods: LIFT (Zhao & Shen, 2024), CCM (Chen et al., 2024), PCD (Lee et al., 2025), and PRReg (Han et al., 2024). CCM and PCD are CP-based baselines. We exclude PCD and PRReg from some experiments, since PCD is transformer-specific and PRReg requires specific optimization. More details refer to Appendix C.2.

**Implementation Details**. For fair evaluation, we adopt Time-Series-Library (Wu et al., 2023) to reproduce all baselines and report their performance with optimal hyperparameters. All experiments are conducted using PyTorch on a single NVIDIA RTX A100 80GB GPU. We use Adam optimizer (Kingma & Ba, 2014) with learning rates selected from {5e-4, 1e-4}. The reported results are averaged over five runs with different random seeds. For dynamic expert allocation, the threshold is selected from {0.0, 0.5}. Experimental setups and hyperparameters are provided in Appendix D.

## 5.1 Long-Term Forecasting Performance

As shown in Table 1, we evaluate long-term time series forecasting performance using mean squared error (MSE) and mean absolute error (MAE) in 9 real-world datasets. The results demonstrate that integrating xCPD consistently improves predictive accuracy over the original backbone models on most datasets and forecasting horizons, encompassing 144 experimental settings in total. The improvements are most pronounced in datasets with complex, high-dimensional channel interactions. Specifically, xCPD achieves great MSE reductions on the electricity dataset (321 variables) and on the traffic dataset (862 variables), averaged over 4 backbones (with additional backbones evaluated in Appendix C.3). These gains stem from xCPD's ability to (i) model fine-grained channel-patch dependencies through ego-graph representations and (ii) suppress irrelevant connections via spectral filtering, thereby improving inter-channel relationship modeling. Since DLinear and PatchTST report results with lookback horizon 336, we provide the same setting in Appendix C.4 to further demonstrate xCPD's effectiveness. We provide more performance analysis in Appendix C.5.

As shown in Table 2, we further compare xCPD with two best-performing baselines: LIFT (Zhao & Shen, 2024) and CCM (Chen et al., 2024). LIFT improves forecasting by identifying leading indicators and leveraging their signals to guide lagging channels. CCM dynamically clusters channels based on similarity, balancing between CI and CD strategies. xCPD consistently outperforms both baselines by modeling fine-grained channel-patch dependencies and explicitly incorporating frequency-domain dependencies. Notably, xCPD maintains its performance advantage even under non-stationary conditions in Appendix C.6, showing the robustness of our shared basis in Theorem 4.1.

Table 2: Averaged long-term forecasting results with horizons {96, 192, 336, 720} and look-back horizon are 512 for TSMixer and 96 for DLinear on 9 datasets in terms of MSE and MAE. The better performance is shown in **bold**.

| Model | TSMixer | | + LIFT | | + CCM | | + xCPD | | DLinear | | + LIFT | | + CCM | | + xCPD | |
|---|---|---|---|---|---|---|---|---|---|---|---|---|---|---|---|---|
| Metric | MSE | MAE | MSE | MAE | MSE | MAE | MSE | MAE | MSE | MAE | MSE | MAE | MSE | MAE | MSE | MAE |
| **ETTh1** | 0.412 | 0.428 | 0.407 | 0.424 | 0.413 | 0.428 | **0.401** | **0.416** | 0.456 | 0.452 | 0.451 | 0.448 | 0.451 | 0.451 | **0.445** | **0.435** |
| **ETTh2** | 0.355 | 0.401 | 0.351 | 0.397 | 0.351 | 0.399 | **0.345** | **0.392** | 0.559 | 0.515 | 0.553 | 0.511 | 0.552 | 0.511 | **0.507** | **0.479** |
| **ETTm1** | 0.347 | 0.375 | 0.342 | 0.372 | 0.351 | 0.380 | **0.343** | **0.371** | 0.403 | 0.407 | 0.399 | 0.404 | 0.397 | 0.403 | **0.389** | **0.395** |
| **ETTm2** | 0.267 | 0.322 | 0.264 | 0.319 | 0.258 | 0.319 | **0.251** | **0.312** | 0.350 | 0.401 | 0.346 | 0.398 | **0.306** | **0.369** | 0.340 | 0.388 |
| **Exchange** | 0.381 | 0.414 | 0.378 | 0.411 | 0.355 | 0.402 | **0.349** | **0.396** | 0.354 | 0.414 | 0.351 | 0.411 | 0.348 | 0.409 | **0.312** | **0.380** |
| **Solar-Eng.** | 0.231 | 0.259 | 0.229 | 0.257 | 0.234 | 0.263 | **0.228** | **0.257** | 0.330 | 0.401 | 0.327 | 0.398 | 0.327 | 0.399 | **0.322** | **0.393** |
| **Weather** | 0.234 | 0.268 | 0.231 | 0.266 | 0.225 | 0.263 | **0.221** | **0.259** | 0.265 | 0.317 | 0.262 | 0.314 | 0.262 | 0.309 | **0.253** | **0.301** |
| **Electricity** | 0.167 | 0.262 | 0.165 | 0.260 | 0.163 | 0.261 | **0.158** | **0.255** | 0.212 | 0.300 | 0.208 | 0.297 | 0.199 | 0.293 | **0.197** | **0.287** |
| **Traffic** | 0.408 | 0.284 | 0.405 | 0.282 | 0.396 | 0.284 | **0.394** | **0.281** | 0.625 | 0.383 | 0.620 | 0.380 | 0.614 | 0.378 | **0.606** | **0.372** |

In Table 3, we evaluate xCPD against baselines in a unified training setting. Following PRReg (Han et al., 2024), we reproduce two backbones (Linear and Transformer) with both CI and CD variants, where baselines are incorporated as weighted regularization terms added to the primary forecasting loss. Table 3 shows xCPD consistently outperforms competing baselines across all settings, demonstrating its superior adaptability to diverse forecasting architectures.

Table 3: Comparison of baselines under a general setting.

| | | CD | CI | +PRReg | +LIFT | +PCD | +CCM | +xCPD |
|---|---|---|---|---|---|---|---|---|
| ETTh1 | Linear | 0.402 | 0.345 | 0.342 | 0.345 | *None* | 0.342 | **0.338** |
| | Transformer | 0.861 | 0.655 | 0.539 | 0.528 | 0.621 | 0.518 | **0.512** |
| ETTm1 | Linear | 0.404 | 0.354 | 0.311 | 0.316 | *None* | 0.310 | **0.305** |
| | Transformer | 0.458 | 0.379 | 0.349 | 0.356 | 0.404 | 0.300 | **0.289** |
| Exchange | Linear | 0.051 | 0.119 | 0.048 | 0.050 | *None* | 0.045 | **0.042** |
| | Transformer | 0.101 | 0.511 | 0.100 | 0.099 | 0.233 | 0.097 | **0.094** |
| Weather | Linear | 0.142 | 0.169 | 0.131 | 0.133 | *None* | 0.130 | **0.128** |
| | Transformer | 0.251 | 0.168 | 0.180 | 0.178 | 0.198 | 0.164 | **0.161** |
| Electricity | Linear | 0.195 | 0.196 | 0.196 | 0.196 | *None* | 0.195 | **0.192** |
| | Transformer | 0.250 | 0.185 | 0.185 | 0.188 | 0.215 | 0.183 | **0.181** |

## 5.2 Short-Term Forecasting Performance

We evaluate xCPD on univariate short-term forecasting using the M4 and Stock datasets. As shown in Table 4, xCPD consistently outperforms baselines through principled frequency-domain modeling via shared Fourier bases, spectral grouping, and adaptive filtering. Performance gains exhibit clear horizon dependency: long-term forecasting benefits substantially from frequency decomposition of seasonal patterns, while short-term predictions show modest improvements due to limited spectral diversity in local temporal dynamics. Despite this inherent constraint, xCPD maintains consistent gains through patch-level denoising, demonstrating its generalizability across temporal scales.

## 5.3 Zero-Shot Forecasting Performance

Most MTSF methods are coupled to specific datasets, limiting generalization to unseen scenarios. In contrast, xCPD leverages learned frequency-aware filters to capture universal spectral structures,

Table 4: Short-term forecasting results on M4 dataset in terms of SMAPE, MASE, and OWA, and Stock dataset in terms of MSE and MAE. The forecasting horizon is $\{7, 24\}$ for the Stock.

| Model | | TSMixer | + LIFT | + CCM | + xCPD | DLinear | + LIFT | + CCM | + xCPD | PatchTST | + LIFT | + CCM | + xCPD | TimesNet | + LIFT | + CCM | + xCPD |
|---|---|---|---|---|---|---|---|---|---|---|---|---|---|---|---|---|---|
| M4 (Yearly) | SMAPE | 14.70 | 14.88 | 14.67 | **14.56** | 16.96 | 15.93 | 14.33 | **14.12** | 13.47 | 13.35 | 13.30 | **13.28** | 15.37 | 15.12 | 14.42 | **14.36** |
| | MASE | 3.343 | 3.341 | 3.370 | **3.221** | 4.283 | 4.038 | 3.144 | **3.121** | 3.019 | 3.006 | 2.997 | **2.989** | 3.554 | 3.512 | 3.448 | **3.434** |
| | OWA | 0.875 | 0.875 | 0.873 | **0.870** | 1.058 | 0.937 | 0.834 | **0.829** | 0.792 | 0.788 | 0.781 | **0.778** | 0.918 | 0.864 | 0.802 | **0.795** |
| M4 (Quarterly) | SMAPE | 11.18 | 11.02 | 10.98 | **10.88** | 12.14 | 11.48 | 10.51 | **10.49** | 10.38 | 10.36 | 10.35 | **10.35** | 10.46 | 10.28 | 10.12 | **10.10** |
| | MASE | 1.346 | 1.338 | 1.332 | **1.326** | 1.520 | 1.482 | 1.243 | **1.224** | 1.233 | 1.228 | 1.224 | **1.218** | 1.227 | 1.221 | 1.183 | **1.176** |
| | OWA | 0.998 | 0.992 | 0.984 | **0.976** | 1.106 | 1.098 | 0.931 | **0.928** | 0.921 | 0.918 | 0.915 | **0.912** | 0.923 | 0.913 | 0.897 | **0.889** |
| M4 (Monthly) | SMAPE | 13.43 | 13.42 | 13.40 | **13.40** | 13.51 | 12.48 | 13.37 | **13.36** | 12.95 | 12.90 | 12.67 | **12.66** | 13.51 | 13.38 | 12.79 | **12.66** |
| | MASE | 1.022 | 1.020 | 1.019 | **1.014** | 1.037 | 1.021 | 1.005 | **1.001** | 0.970 | 0.952 | 0.941 | **0.937** | 1.039 | 1.022 | 0.942 | **0.934** |
| | OWA | 0.946 | 0.945 | 0.944 | **0.935** | 0.956 | 0.948 | 0.936 | **0.931** | 0.905 | 0.900 | 0.895 | **0.892** | 0.957 | 0.931 | 0.891 | **0.883** |
| M4 (Others) | SMAPE | 7.067 | 7.055 | 7.178 | **7.045** | 6.709 | 6.354 | 6.160 | **6.145** | 4.952 | 4.855 | 4.643 | **4.633** | 6.913 | 6.599 | 5.218 | **5.118** |
| | MASE | 5.587 | 5.482 | 5.302 | **5.288** | 4.953 | 4.899 | 4.713 | **4.616** | 3.347 | 3.323 | 3.128 | **3.112** | 4.507 | 4.344 | 3.892 | **3.854** |
| | OWA | 1.642 | 1.580 | 1.536 | **1.522** | 1.487 | 1.443 | 1.389 | **1.378** | 1.049 | 1.022 | 0.997 | **0.988** | 1.438 | 1.359 | 1.217 | **1.208** |
| M4 (Avg.) | SMAPE | 12.86 | 12.83 | 12.80 | **12.79** | 13.63 | 13.42 | 12.54 | **12.43** | 12.05 | 11.89 | 11.85 | **11.81** | 12.88 | 12.56 | 11.91 | **11.73** |
| | MASE | 1.887 | 1.879 | 1.864 | **1.860** | 2.095 | 1.982 | 1.740 | **1.733** | 1.623 | 1.604 | 1.587 | **1.572** | 1.836 | 1.799 | 1.603 | **1.592** |
| | OWA | 0.957 | 0.950 | 0.948 | **0.923** | 1.051 | 1.002 | 0.917 | **0.905** | 0.869 | 0.459 | 0.840 | **0.831** | 0.955 | 0.934 | 0.894 | **0.885** |
| Stock (Horizon 7) | MSE | 0.939 | 0.938 | 0.938 | **0.932** | 0.992 | 0.963 | 0.883 | **0.879** | 0.896 | 0.984 | 0.892 | **0.890** | 0.930 | 0.922 | 0.915 | **0.912** |
| | MAE | 0.807 | 0.807 | 0.806 | **0.800** | 0.831 | 0.821 | 0.774 | **0.768** | 0.771 | 0.773 | 0.771 | **0.770** | 0.802 | 0.805 | 0.793 | **0.789** |
| Stock (Horizon 24) | MSE | 1.007 | 1.005 | 0.991 | **0.986** | 0.996 | 0.968 | 0.917 | **0.912** | 0.930 | 0.912 | 0.880 | **0.878** | 0.998 | 0.958 | 0.937 | **0.933** |
| | MAE | 0.829 | 0.821 | 0.817 | **0.805** | 0.832 | 0.810 | 0.781 | **0.774** | 0.789 | 0.779 | 0.765 | **0.752** | 0.830 | 0.815 | 0.789 | **0.775** |

enabling effective knowledge transfer for zero-shot forecasting without retraining. Following prior work (Jin et al., 2024), we evaluate zero-shot performance on the ETT benchmark comprising four datasets with varying regional and temporal resolutions. As shown in Table 5 and Table 13, xCPD consistently improves zero-shot performance across all 48 evaluation settings. Notably, (i) performance gains increase with horizon length, demonstrating xCPD's ability to transfer frequency knowledge to more challenging long-range dependencies; and (ii) CI-based models benefit more substantially, with 12.0% and 15.2% average improvements for DLinear and PatchTST versus 6.7% and 11.1% for TSMixer and TimesNet. These results indicate promising transferability under frequency-aligned settings. Broader cross-domain transfer is left for future work.

Table 5: Zero-shot forecasting MSE results on ETT datasets. The forecasting horizon is $\{96, 720\}$.

| Model | TSMixer | + CCM | + xCPD | DLinear | + CCM | + xCPD | PatchTST | + CCM | + xCPD | TimesNet | + CCM | + xCPD | IMP(%) |
|---|---|---|---|---|---|---|---|---|---|---|---|---|---|
| h1→h2 | 0.288 | 0.283 | **0.280** | 0.358 | 0.344 | **0.335** | 0.442 | 0.438 | **0.418** | 0.391 | 0.388 | **0.384** | 2.318 |
| | 0.374 | 0.370 | **0.366** | 0.855 | 0.843 | **0.822** | 0.582 | 0.571 | **0.544** | 0.540 | 0.516 | **0.510** | 2.366 |
| h1→m1 | 0.763 | 0.710 | **0.702** | 0.782 | 0.723 | **0.702** | 0.754 | 0.692 | **0.678** | 0.887 | 0.827 | **0.801** | 2.300 |
| | 1.252 | 1.215 | **1.198** | 1.243 | 1.224 | **1.212** | 1.892 | 1.135 | **1.128** | 1.623 | 1.601 | **1.589** | 0.937 |
| h1→m2 | 0.959 | 0.937 | **0.928** | 1.213 | 0.912 | **0.890** | 1.132 | 0.903 | **0.892** | 1.199 | 1.122 | **1.086** | 1.950 |
| | 1.765 | 1.758 | **1.712** | 2.132 | 1.732 | **1.680** | 2.015 | 1.732 | **1.712** | 2.204 | 1.874 | **1.835** | 2.214 |
| h2→h1 | 0.466 | 0.455 | **0.443** | 0.489 | 0.445 | **0.423** | 0.632 | 0.532 | **0.512** | 0.869 | 0.752 | **0.731** | 3.533 |
| | 0.695 | 0.540 | **0.528** | 0.533 | 0.495 | **0.492** | 1.032 | 0.986 | **0.944** | 1.274 | 0.845 | **0.834** | 2.098 |
| h2→m2 | 0.943 | 0.876 | **0.868** | 0.758 | 0.732 | **0.708** | 0.855 | 0.781 | **0.772** | 1.250 | 1.064 | **1.033** | 2.064 |
| | 1.472 | 1.464 | **1.422** | 1.892 | 1.422 | **1.274** | 1.899 | 1.644 | **1.542** | 1.861 | 1.671 | **1.633** | 5.439 |
| h2→m1 | 1.254 | 1.073 | **1.022** | 1.243 | 0.939 | **0.898** | 1.012 | 0.809 | **0.794** | 1.049 | 0.804 | **0.794** | 3.054 |
| | 2.275 | 1.754 | **1.721** | 2.012 | 1.850 | **1.792** | 2.732 | 1.832 | **1.732** | 2.183 | 1.742 | **1.712** | 3.049 |

## 5.4 EFFICIENCY EVALUATION

Figure 2 evaluates the computational efficiency on Traffic under the input-96-predict-96 setting. The overhead of LIFT and CCM is highly model-dependent, increasing substantially with large forecasting models. While CCM achieves strong performance, its quadratic cost with respect to the number of clusters limits scalability. In contrast, xCPD maintains model-agnostic efficiency through linear-complexity spectral operations, achieving comparable performance with significantly

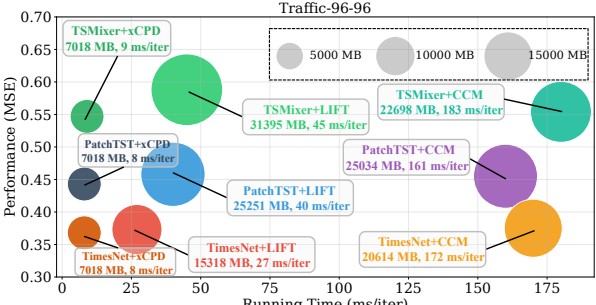

Figure 2: Model efficiency comparison on Traffic.

lower overhead. Table 14 in Appendix C.9 further shows that the time overhead introduced by xCPD accounts for only 9%–11% of the backbone model's training cost. These efficiency advantages make xCPD particularly suitable for large-scale and real-time forecasting applications.

## 5.5 ABLATION STUDY

We conduct ablation studies to evaluate each component's contribution in xCPD, as shown in Table 6. **Component Removal:** (1) w/o Shared Basis removes the shared Fourier basis, using separate bases per batch. (2) w/o Freq. Partition removes learnable frequency boundaries, using fixed equal splits.

Table 6: Ablation study of xCPD. Look-back horizon is 96 and results are averaged over forecasting horizons {96, 192, 336, 720}. Best and second-best results are marked in **bold** and underline.

| Variants | | ETTh1 | | ETTm1 | | Exchange | | Solar-Eng. | | Weather | | Electricity | | Traffic | |
|---|---|---|---|---|---|---|---|---|---|---|---|---|---|---|---|---|
| | | MSE | MAE | MSE | MAE | MSE | MAE | MSE | MAE | MSE | MAE | MSE | MAE | MSE | MAE |
| DLinear+xCPD | | **0.445** | **0.435** | **0.389** | **0.395** | **0.312** | **0.380** | **0.322** | **0.393** | **0.253** | **0.301** | **0.197** | **0.287** | **0.606** | **0.372** |
| Remove | w/o Shared Basis | 0.462 | 0.456 | 0.404 | 0.415 | 0.331 | 0.395 | 0.341 | 0.399 | 0.267 | 0.310 | 0.205 | 0.295 | 0.618 | 0.379 |
| | w/o Freq. Partition | 0.448 | 0.394 | 0.399 | 0.403 | 0.318 | 0.388 | 0.329 | 0.398 | 0.264 | 0.307 | 0.203 | 0.292 | 0.610 | 0.376 |
| | w/o Node Group | 0.461 | 0.455 | 0.406 | 0.415 | 0.329 | 0.394 | 0.344 | 0.402 | 0.266 | 0.310 | 0.204 | 0.293 | 0.616 | 0.377 |
| | w/o Filters | 0.465 | 0.456 | 0.408 | 0.416 | 0.333 | 0.397 | 0.344 | 0.402 | 0.272 | 0.313 | 0.209 | 0.298 | 0.623 | 0.383 |
| | w/o $\mathcal{L}_{dyn}$ & $\mathcal{L}_{imp}$ | 0.460 | 0.453 | 0.401 | 0.412 | 0.325 | 0.390 | 0.338 | 0.396 | 0.264 | 0.302 | 0.201 | 0.291 | 0.609 | 0.376 |
| Replace | Top-$K$ | 0.464 | 0.456 | 0.410 | 0.420 | 0.333 | 0.396 | 0.343 | 0.420 | 0.269 | 0.314 | 0.208 | 0.298 | 0.621 | 0.380 |
| | Random-$K$ | 0.465 | 0.456 | 0.407 | 0.418 | 0.332 | 0.395 | 0.343 | 0.419 | 0.269 | 0.313 | 0.209 | 0.299 | 0.620 | 0.380 |
| | RegionTop-$K$ | 0.459 | 0.453 | 0.405 | 0.415 | 0.331 | 0.392 | 0.335 | 0.398 | 0.265 | 0.308 | 0.206 | 0.295 | 0.617 | 0.378 |
| | TimeFilter | 0.457 | 0.452 | 0.406 | 0.415 | 0.328 | 0.390 | 0.336 | 0.402 | 0.262 | 0.305 | 0.202 | 0.291 | 0.615 | 0.377 |

(3) w/o Node Group replaces energy-based node grouping with random assignment. (4) w/o Filter omits the filtering step and uses full ego-graphs. (5) w/o $\mathcal{L}_{dyn}$&$\mathcal{L}_{imp}$ removes regularization on expert diversity and balance. **Strategy Replacement:** (6) Top-K selects top-$K$ edges by weight in each ego-graph. (7) Random-K randomly samples $K$ edges. (8) RegionTop-K selects top-$K$ edges within each frequency band. (9) TimeFilter applies a spatial-temporal filter (Hu et al., 2025b). Please note that, to ensure a fair mechanism-level comparison within our ablation framework, we adopt only the core filtering module of TimeFilter and integrate it as a plugin into the DLinear backbone (denoted as DLinear+TimeFilter), rather than using the full TimeFilter architecture. All replacement strategies lead to performance drops, highlighting the effectiveness of xCPD's spectral filters. Appendix C.7 provides a more detailed analysis.

## 5.6 QUALITATIVE STUDY

Figure 3 illustrates the correlation between spectral energy and temporal patterns in channel-patch nodes. The red node exhibits smooth temporal trends with gradual variations, concentrating energy in low-frequency bands and primarily aggregating dependencies from other low-frequency nodes. In contrast, the green node displays abrupt fluctuations and rapid state changes, resulting in dominant high-frequency spectral energy that drives its connections to other high-frequency nodes. Importantly, these frequency assignments are not manu-

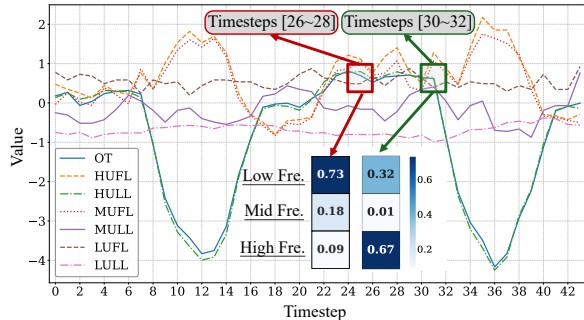

Figure 3: Filters visualization of HULL on ETTh1.

ally predetermined. They emerge naturally from the learned spectral energy response, which allows the model to discover frequency structure inherent in the data rather than relying on heuristic thresholds. This interpretability is difficult to obtain from temporal-domain attention mechanisms, where trend, fluctuation, and noise components remain mixed and cannot be disentangled. Theorem 4.2 further guarantees that spectral energy preserves all information in the original patch embedding. As a result, the alignment between temporal behaviors and spectral energy distributions observed in Figure 3 represents a faithful and principled consequence of spectral decomposition, providing intuitive and robust insight into the learned frequency-based grouping.

## 6 CONCLUSION

In this paper, we propose xCPD, a lightweight plugin module that enhances multivariate time series forecasting by modeling fine-grained, frequency-aware channel-patch dependencies. Extensive experiments demonstrate that xCPD consistently improves forecasting accuracy across diverse datasets and backbones while maintaining linear computational complexity. Its model-agnostic design enables seamless integration into existing forecasting models, establishing spectral decomposition and adaptive routing as powerful tools for capturing complex temporal dependencies in MTSF.

## ETHICS STATEMENT

This work presents a technical contribution to multivariate time series forecasting and raises no ethical concerns. Our research does not involve human subjects, uses only publicly available benchmark datasets with proper citations, and contains no personally identifiable information. The proposed method, xCPD, is a general-purpose forecasting enhancement technique with no inherent potential for harm or misuse beyond standard time series prediction applications. We have no conflicts of interest to declare. The computational requirements of xCPD are modest and environmentally responsible compared to existing models. We confirm that this research complies with the ICLR Code of Ethics.

## REPRODUCIBILITY STATEMENT

To ensure the reproducibility of our work, we provide comprehensive implementation details and experimental protocols throughout the paper and the supplementary materials. The complete architecture of the model and the algorithmic details are presented in Section 4, with mathematical formulations for all key components. Section 5 describes the experimental setup, including the 9 benchmark datasets used, baseline methods, and implementation details such as optimizer settings, learning rates, and random seed protocols. Appendix B provides detailed statistics and pre-processing steps of the dataset. Appendix C contains extended experimental results and additional implementation specifics, including batch sizes, training epochs, and early stopping criteria. Appendix D lists all evaluation metrics and the complete hyperparameter search spaces with selected values for each dataset. All experiments were carried out using PyTorch on NVIDIA RTX A100 GPUs, with average results of 5 random seeds to ensure statistical reliability. The anonymous repository provided contains the full source code, data pre-processing scripts, and trained model checkpoints to facilitate the exact reproduction of all reported results.

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

APPENDIX CONTENTS

## A    THE USE OF LARGE LANGUAGE MODELS (LLMs)

We acknowledge the use of large language models to assist with manuscript preparation. Specifically, LLMs were employed solely for language refinement tasks, including grammatical corrections, vocabulary enhancement, and sentence structure improvements. All scientific content, including research methodology, experimental design, data analysis, theoretical developments, and technical conclusions, represents the original intellectual contribution of the authors. The LLMs did not generate any substantive scientific content, interpretations, or novel insights presented in this work.

## B    DATASETS

### B.1    LONG-TERM TIME SERIES FORECASTING DATASETS

In Table 7, we summarize the detailed statistics of all datasets used in our experiments. We conduct extensive evaluations on 9 widely used multivariate time series datasets for long-term forecasting and two datasets for short-term forecasting. The ETT dataset (Zhou et al., 2021) contains temperature and power load data collected at hourly (ETTh) and 15-minute (ETTm) intervals. Exchange (Lai et al., 2018) compiles daily exchange rate information from 1990 to 2016. Solar-Energy (Lai et al., 2018) provides 10-minute solar power readings from 137 PV plants. The Weather (Wu et al., 2023), Electricity (Wu et al., 2023), and Traffic (Wu et al., 2023) datasets include high-resolution environmental and usage signals, recorded respectively at 10-minute, hourly, and hourly intervals.

### B.2    SHORT-TERM TIME SERIES FORECASTING DATASETS

For short-term univariate forecasting, we use the M4 dataset (Chen et al., 2024), comprising 100,000 time series across multiple domains and six frequency-based subgroups. Additionally, we use a stock forecasting benchmark (Chen et al., 2024) based on 10 years of 5-minute price data from 1,390 stocks. The data span from November 25, 2013 to September 1, 2023, with missing timestamps interpolated to ensure sequence alignment. This stock dataset emphasizes short-term dynamics and market fluctuations, with forecasting horizons set to 7 and 24 time steps.

Table 7: Statistics of all datasets used in our experiments. "Frequency" denotes the sampling interval of time points.

| Dataset | Type | Dim | Lookback Length | Prediction Length | Total Length | Frequency | Domain |
|---|---|---|---|---|---|---|---|
| *Long-term Forecasting Datasets* | | | | | | | |
| ETTh1 | Multivariate | 7 | 96 | {96, 192, 336, 720} | 17,420 | Hourly | Electricity |
| ETTh2 | Multivariate | 7 | 96 | {96, 192, 336, 720} | 17,420 | Hourly | Electricity |
| ETTm1 | Multivariate | 7 | 96 | {96, 192, 336, 720} | 69,680 | 15-min | Electricity |
| ETTm2 | Multivariate | 7 | 96 | {96, 192, 336, 720} | 69,680 | 15-min | Electricity |
| Weather | Multivariate | 21 | 96 | {96, 192, 336, 720} | 52,695 | 10-min | Meteorology |
| Electricity | Multivariate | 321 | 96 | {96, 192, 336, 720} | 26,304 | Hourly | Energy |
| Traffic | Multivariate | 862 | 96 | {96, 192, 336, 720} | 17,544 | Hourly | Transportation |
| Exchange | Multivariate | 8 | 96 | {96, 192, 336, 720} | 7,588 | Daily | Finance |
| Solar-Energy | Multivariate | 137 | 96 | {96, 192, 336, 720} | 52,560 | 10-min | Energy |
| *Short-term Forecasting Datasets* | | | | | | | |
| M4-Yearly | Univariate | 1 | {12, 18, 24, 30, 36, 42} | 6 | Various | Yearly | Mixed |
| M4-Quarterly | Univariate | 1 | {16, 24, 32, 40, 48, 56} | 8 | Various | Quarterly | Mixed |
| M4-Monthly | Univariate | 1 | {36, 54, 72, 90, 108, 126} | 18 | Various | Monthly | Mixed |
| M4-Weekly | Univariate | 1 | {26, 39, 52, 65, 78, 91} | 13 | Various | Weekly | Mixed |
| M4-Daily | Univariate | 1 | {28, 42, 56, 70, 84, 98} | 14 | Various | Daily | Mixed |
| M4-Hourly | Univariate | 1 | {96, 144, 192, 240, 288, 336} | 48 | Various | Hourly | Mixed |
| Stock | Univariate | 1 | {24,48} | {7, 24} | 2,026,990 | 5-min | Finance |

## C    ADDITIONAL EXPERIMENTS

### C.1    ADDITIONAL DETAILS OF ADJACENCY MATRIX CONSTRUCTION

This subsection provides additional details on the construction of the adjacency matrix $A^t$ mentioned in Section 4.1, focusing on three key aspects: (i) non-triviality of dense similarity graphs, (ii)

numerical stability through Laplacian normalization, and (iii) the operational sparsity enforced during routing.

- The adjacency matrix is constructed using cosine similarity $A_{ij}^t = \cos\left(X_i^{\mathrm{emb},t}, X_j^{\mathrm{emb},t}\right)$, which guarantees that all affinity values lie within the bounded interval $[-1, 1]$. This bounded-ness ensures numerical stability and prevents degenerate scaling of affinities. Moreover, cosine similarity preserves angular variation between embeddings, yielding a highly non-uniform structure even when the matrix is dense. The diagonal elements naturally satisfy $A_{ii}^t = 1$, eliminating the need for explicit self-loop regularization. Dense cosine-similarity graphs of this form are standard in spectral graph methods, where global affinity patterns are crucial for identifying meaningful frequency components.

- The shared Fourier basis is derived from the normalized Laplacian $L^t = I - D^{-1/2} A^t D^{-1/2}$, whose eigenvalues are guaranteed to lie within $[0, 2]$. This normalization controls the operator scale, stabilizes the eigendecomposition, and promotes clear separation between low- and high-frequency components regardless of the density of $A^t$. These properties ensure that spectral representations are well-conditioned across timesteps, which is essential for learning the shared Fourier basis in Theorem 4.1.

- Although $A^t$ is dense, downstream routing does not operate on the full dense graph. Instead, xCPD applies a k-NN sparsification procedure to each node, retaining only the most relevant neighbors based on similarity. Furthermore, the frequency-aware MoE router activates only a subset of spectral components for each sample, resulting in selective and structured dependency pathways. Consequently, the effective computation graph is sparse and informative, avoiding the trivial connectivity patterns typically associated with dense graphs.

## C.2 COMPARISON WITH PCD AND PRREG

PCD (Lee et al., 2025) is not included in our main experimental tables due to its architectural constraints. As detailed in their original work, PCD's core mechanisms require self-attention operations, limiting its applicability to Transformer-based models. This restriction prevents fair evaluation across the diverse model families considered in our study, including linear models (DLinear) and CNN-based architectures (TimesNet). To provide comprehensive analysis, we conduct additional experiments comparing xCPD and PCD on PatchTST, a Transformer backbone where both plugins are applicable. As shown in Table 8, xCPD consistently outperforms PCD across all eight benchmark datasets, demonstrating superior performance even within PCD's intended architectural domain.

Table 8: Long-term forecasting results with lookback window 96 and prediction horizons {96, 192, 336, 720}. Results show MSE/MAE averaged over three runs with best in **bold**.

| Method | ETTh1 | ETTh2 | ETTm1 | ETTm2 | Exchange | Weather | Electricity | Traffic |
|---|---|---|---|---|---|---|---|---|
| PatchTST+PCD | 0.468/0.456 | 0.379/0.399 | 0.382/0.393 | 0.272/0.325 | 0.366/0.403 | 0.255/0.281 | 0.198/0.288 | 0.482/0.300 |
| PatchTST+xCPD | **0.455/0.443** | **0.367/0.393** | **0.371/0.388** | **0.265/0.317** | **0.358/0.397** | **0.248/0.272** | **0.191/0.280** | **0.466/0.292** |

PRReg (Han et al., 2024) differs fundamentally from plugin modules in its design philosophy. While we compare against PRReg in Table 3 using their proposed regularization framework, PRReg operates through loss regularization terms requiring joint optimization and careful hyperparameter tuning for each backbone-dataset combination. This contrasts with xCPD's plug-and-play design, which can be applied post-hoc without modifying the training process. Furthermore, PRReg's performance varies significantly across architectures due to its dependence on model-specific gradient dynamics, whereas xCPD maintains consistent improvements through its architecture-agnostic spectral operations. Despite these differences, xCPD outperforms PRReg even when evaluated within PRReg's own optimization framework (Table 3), further validating our approach.

## C.3 PERFORMANCE ON ADDITIONAL BACKBONES

To evaluate xCPD's generalizability beyond our primary baselines, we conduct experiments on two recent forecasting architectures: iTransformer (Liu et al., 2024a) and TimeMixer (Wang et al., 2024a). These models represent state-of-the-art advances in attention-based and MLP-mixing architectures,

respectively. Table 9 presents results across six benchmark datasets. xCPD achieves consistent improvements, with average MSE reductions of 3.7% for iTransformer and 2.8% for TimeMixer. The improvements are particularly pronounced on Solar-Energy (3.0% and 2.8% respectively), where complex multi-scale temporal patterns benefit from our frequency-aware modeling. These results confirm xCPD's compatibility with diverse architectural paradigms beyond the linear, transformer, and CNN-based models evaluated in our main experiments.

Table 9: Performance on iTransformer and TimeMixer. Results averaged over forecasting horizons {96, 192, 336, 720} with lookback window 96. Values show MSE/MAE with best results in **bold**.

| Model | ETTh1 | ETTh2 | ETTm1 | ETTm2 | Solar-Energy | Weather |
|---|---|---|---|---|---|---|
| iTransformer | 0.455/0.447 | 0.383/0.407 | 0.407/0.411 | 0.288/0.332 | 0.233/0.262 | 0.258/0.278 |
| iTransformer+xCPD | **0.438/0.440** | **0.375/0.400** | **0.394/0.401** | **0.277/0.325** | **0.226/0.256** | **0.250/0.269** |
| TimeMixer | 0.447/0.440 | 0.365/0.396 | 0.381/0.395 | 0.275/0.323 | 0.216/0.280 | 0.240/0.271 |
| TimeMixer+xCPD | **0.435/0.432** | **0.358/0.389** | **0.372/0.388** | **0.268/0.317** | **0.210/0.275** | **0.233/0.266** |
| DUET | 0.443/0.437 | 0.373/0.398 | 0.390/0.394 | 0.280/0.325 | 0.238/0.235 | 0.252/0.273 |
| **DUET +xCPD** | **0.436/0.433** | **0.364/0.392** | **0.381/0.390** | **0.275/0.318** | **0.231/0.234** | **0.246/0.270** |

Beyond the CI/CD backbones evaluated in this paper, we were also interested in examining whether xCPD remains effective when paired with a CP-based backbone. To this end, we additionally include DUET Qiu et al. (2024), which performs channel-level dependency modeling through global channel clustering. Following the same experimental setup as in our main experiments, we freeze all backbone parameters and fine-tune only the parameters introduced by our xCPD plugin across six datasets. As shown in Table 9, we observe two interesting findings: (1) xCPD continues to yield performance improvements even on a CP backbone. This is because DUET models channel level dependencies, whereas xCPD captures channel–patch level dependencies that provide finer-grained relational information; (2) The improvement achieved by DUET is less pronounced than that of CI/CD backbones such as TSMixer and iTransformer. We hypothesize that DUET's hard channel clustering partially disrupts the original dependency structure in the raw signal, making subsequent patch-level CPD more difficult to learn.

## C.4 IMPACT OF INPUT LENGTH ON PERFORMANCE

The performance of DLinear (Zeng et al., 2023) on the Traffic dataset differs between our Table 1 and the original paper. This discrepancy arises from different lookback window configurations: DLinear's original experiments use a lookback window of 336, while we adopt the standardized setting of 96 following recent work (Liu et al., 2024a) to ensure fair comparison across all baselines. To investigate the impact of input length and validate xCPD's robustness, we conduct experiments with both configurations. Table 10 and Table 11 present results under lookback horizons of 96 and 336, respectively.

Table 10: Performance on Traffic dataset with look-back horizon 96.

| Method | Forecasting Horizon | | | |
|---|---|---|---|---|
| | 96 MSE/MAE | 192 MSE/MAE | 336 MSE/MAE | 720 MSE/MAE |
| DLinear | 0.650/0.396 | 0.598/0.370 | 0.605/0.373 | 0.645/0.394 |
| DLinear+xCPD | **0.636/0.386** | **0.574/0.356** | **0.586/0.362** | **0.630/0.387** |

The results demonstrate two key findings: (1) DLinear's performance with lookback window 336 aligns with the original paper, confirming our implementation correctness; (2) xCPD consistently improves performance across both configurations, achieving average MSE reductions of 3.2% (lookback=96) and 2.3% (lookback=336). Notably, the longer input window provides better baseline performance (0.435 vs 0.625 average MSE), yet xCPD maintains its effectiveness regardless of input length. This confirms that performance differences stem from experimental settings rather than implementation issues, while validating xCPD's robustness across varying input contexts.

Table 11: Performance on Traffic dataset with look-back horizon 336.

| Method | Prediction Horizon | | | |
|---|---|---|---|---|
| | 96
MSE/MAE | 192
MSE/MAE | 336
MSE/MAE | 720
MSE/MAE |
| DLinear | 0.412/0.282 | 0.423/0.287 | 0.437/0.296 | 0.468/0.316 |
| DLinear+xCPD | **0.406/0.280** | **0.411/0.283** | **0.428/0.295** | **0.457/0.311** |

## C.5 DATASET-DEPENDENT PERFORMANCE IMPROVEMENTS

We observe that the performance gains brought by xCPD are not uniform across datasets, and these differences are closely tied to the spectral and multivariate characteristics of the underlying signals. Since xCPD explicitly models fine-grained, frequency-aware channel–patch dependencies, datasets with richer spectral variability naturally benefit more from our plugin. We summarize the dataset-dependent behavior as follows.

- **Small improvements (1%–3%): ETT and SolarEnergy.** They exhibit highly regular, low-frequency periodicity dominated by daily or hourly seasonal cycles. Their cross-channel relationships are relatively stable and already well captured by CI/CD backbones such as TSMixer and iTransformer. As a result, xCPD provides moderate but consistent improvements by modeling subtle patch-level variations that are not fully exploited by these backbones.

- **Medium improvements (3%–4%): Exchange and Weather.** They contain mixed spectral structures, combining smooth seasonal trends with occasional short-term fluctuations. These datasets feature moderate multivariate interactions that are partially captured by the backbone but still benefit from xCPD's ability to extract localized, frequency-aware channel–patch dependencies. Consequently, xCPD yields steady improvements in this mid-range category.

- **Large improvements (4%–7%): Electricity and Traffic.** They display the richest multi-scale frequency patterns among all datasets, including local bursts, irregular short-term spikes, asynchronous channel shifts, and strong high-frequency components. Such volatile and rapidly changing cross-channel relationships challenge backbones that rely on coarse channel mixing. In these settings, xCPD's fine-grained frequency-aware modeling becomes particularly effective, consistently producing the largest performance gains across all benchmarks.

In summary, the improvement pattern across datasets aligns strongly with their spectral complexity: the richer and more volatile the frequency structure, the greater the benefit from explicit frequency-aware channel–patch dependency modeling. This analysis provides practical guidance on when xCPD yields the most substantial improvements and highlights the importance of spectral diversity in multivariate time-series forecasting.

## C.6 ROBUSTNESS TO NON-STATIONARY DYNAMICS

Real-world multivariate time series often exhibit non-stationary behavior with evolving inter-channel dependencies. To evaluate xCPD's robustness under such conditions, we conduct controlled experiments with synthetically induced non-stationarity.

**Experimental Setup.** We perturb the ETTm1 dataset to simulate temporal misalignment and amplitude variations commonly observed in real-world scenarios. Specifically, we apply two types of perturbations to each channel: (i) random temporal shifts within $\pm 12$ time steps to disrupt cross-channel synchronization, and (ii) random amplitude scaling factors sampled from $\mathcal{U}[0.8, 1.2]$ to introduce magnitude variations. These perturbations directly challenge the stationarity assumption underlying our shared Fourier basis (Theorem 4.1), providing a rigorous test of xCPD's robustness.

**Results and Analysis.** Table 12 presents the performance comparison with and without the shared spectral basis under synthetic non-stationarity. Despite induced perturbations, xCPD maintains strong performance with only marginal degradation compared to the stationary setting (Table 1). Notably, removing the shared basis results in significant performance deterioration (8.3% MSE increase), confirming its critical role in maintaining spectral alignment under non-stationary conditions.

Table 12: Ablation study under synthetic non-stationarity on ETTm1 (lookback window: 96, forecasting horizon: 96). The shared spectral basis maintains performance despite temporal perturbations.

| Model | Configuration | MSE ↓ | MAE ↓ |
|---|---|---|---|
| DLinear+xCPD | w/ Shared Basis | **0.389** | **0.395** |
| | w/o Shared Basis | 0.413 | 0.425 |
| TimesNet+xCPD | w/ Shared Basis | **0.373** | **0.384** |
| | w/o Shared Basis | 0.395 | 0.403 |

This robustness stems from two complementary mechanisms. First, as formalized in Theorem 4.1, the shared basis $U$ approximates time-varying bases $U^{(t)}$ up to orthogonal rotation, preserving spectral relationships despite temporal variations. The bound $\|U^{(t)} - U R^{(t)}\|_F \leq C \|L^{(t)} - L_{\text{avg}}\|_F$ ensures graceful degradation rather than catastrophic failure when stationarity is violated. Second, the dynamic MoE routing provides local adaptability by selecting frequency-specific filters based on instantaneous node characteristics, effectively compensating for global approximation errors introduced by the shared basis. These results, combined with strong performance on inherently non-stationary datasets, validate that xCPD effectively handles time-varying dependencies through the synergy of global spectral alignment and local adaptive routing. This dual mechanism enables robust performance across both controlled perturbations and real-world non-stationary scenarios.

## C.7 ANALYSIS OF ABLATION STUDY

The ablation *w/o Freq. Partition* replaces learnable boundaries with fixed frequency band allocation (33%/34%/33%). Despite limiting adaptation to dataset-specific spectral patterns, this variant shows minimal performance degradation compared to other ablations. This robustness comes from three complementary factors:

- First, the spectral grouping mechanism remains functional. Despite fixed boundaries, nodes compute spectral energy responses across three bands, enabling frequency-aware graph construction that captures coarse but discriminative patterns (e.g., low-frequency dominance for trends).

- Second, the shared Fourier basis $U$ maintains consistent spectral representation across batches, thereby ensuring stable energy computation that remains independent of boundary adaptation.

- Third, the MoE routing module retains full adaptability by dynamically adjusting filter selection based on node-specific energy profiles, which partially compensates for the limitations imposed by fixed partitioning.

These results suggest that while learnable partitioning enhances fine-grained spectral alignment, the three-band decomposition itself provides a strong inductive bias for frequency-aware modeling. In contrast, ablations that disrupt core mechanisms, such as spectral grouping (*w/o Node Group*) or adaptive filtering (*w/o Filters*), yield larger performance drops, confirming their critical roles.

Table 6 compares the proposed xCPD with several alternative dependency modeling mechanisms using a unified backbone (DLinear). We note that the MSE/MAE values of TimeFilter in Table 6 differ from those reported in the original TimeFilter paper (Hu et al., 2025b). We provide below an explanation to clarify this discrepancy and avoid potential misunderstanding. The original TimeFilter is introduced as a complete forecasting architecture built upon temporal graph neural networks, equipped with its own encoder, normalization layers, and additional architectural components. These design choices contribute substantially to its reported performance. In contrast, the purpose of Table 6 is to evaluate dependency mechanisms in a controlled and unified setting. Therefore, we re-implemented only the core spatio-temporal filtering module of TimeFilter and integrated it as a lightweight plugin into the DLinear backbone. This variant is denoted as **DLinear+TimeFilter**. The plugin implementation isolates solely the dependency-filtering logic, without incorporating the architectural advantages of the full TimeFilter model.

Furthermore, Table 6 is designed to answer the following question: *Which dependency mechanism is more effective when integrated into a generic backbone?* Under this controlled setup, **DLinear+xCPD consistently outperforms DLinear+TimeFilter** across all datasets. This demonstrates that the pro-

posed spectral DyMoE mechanism provides a more effective, generalizable, and backbone-agnostic dependency modeling capability than the temporal-domain filtering strategy used in TimeFilter.

## C.8 ZERO-SHOT FORECASTING PERFORMANCE

Our zero-shot evaluation assesses the transferability of plugin modules across domains without retraining. This setting requires plugins to be architecture-agnostic and training-decoupled. Under these criteria, we compare xCPD with CCM, the only existing baseline that satisfies these requirements. LIFT (Zhao & Shen, 2024) is excluded from this evaluation due to fundamental incompatibilities: (i) it modifies backbone architectures through additional layers and loss terms, requiring joint optimization rather than modular integration; (ii) it learns dataset-specific transformations without storing transferable patterns or prototypes; (iii) it lacks decoupled components necessary for cross-domain deployment. In contrast, both CCM and xCPD maintain explicit, reusable representations, i.e., CCM through frequency prototypes and xCPD through spectral bases, enabling zero-shot transfer.

Table 13 presents the zero-shot forecasting results, where the plugins trained on the source datasets are directly applied to the target datasets without adaptation. xCPD demonstrates superior transferability, achieving consistent improvements across diverse domain pairs. This validates our design principle of learning universal spectral representations rather than dataset-specific transformations.

Table 13: Zero-shot forecasting results on ETT datasets. Look-back horizon is 96 and the forecasting horizon is $\{96, 720\}$. Best performance on each setting is in **bold**.

| Model | TSMixer MSE / MAE | + CCM MSE / MAE | + xCPD MSE / MAE | DLinear MSE / MAE | + CCM MSE / MAE | + xCPD MSE / MAE | PatchTST MSE / MAE | + CCM MSE / MAE | + xCPD MSE / MAE | TimesNet MSE / MAE | + CCM MSE / MAE | + xCPD MSE / MAE |
|---|---|---|---|---|---|---|---|---|---|---|---|---|
| h1→h2 | 0.288 / 0.357 | 0.283 / 0.353 | **0.280 / 0.348** | 0.358 / 0.405 | 0.344 / 0.392 | **0.335 /0.382** | 0.442 / 0.458 | 0.438 / 0.449 | **0.418 / 0.426** | 0.391 / 0.412 | 0.388 / 0.410 | **0.384 / 0.404** |
|  | 0.374 / 0.414 | 0.370 / 0.413 | **0.366 / 0.406** | 0.855 / 0.682 | 0.843 / 0.677 | **0.822 / 0.654** | 0.582 / 0.565 | 0.571 / 0.562 | **0.544 / 0.551** | 0.540 / 0.508 | 0.516 / 0.491 | **0.510 / 0.488** |
| h1→m1 | 0.763 / 0.677 | 0.710 / 0.652 | **0.702 / 0.645** | 0.782 / 0.679 | 0.723 / 0.658 | **0.702 / 0.632** | 0.754 / 0.688 | 0.692 / 0.645 | **0.678 / 0.630** | 0.887 / 0.718 | 0.827 / 0.700 | **0.801 / 0.682** |
|  | 1.252 / 0.815 | 1.215 / 0.803 | **1.198 / 0.792** | 1.243 / 0.835 | 1.224 / 0.812 | **1.212 / 0.798** | 1.892 / 0.954 | 1.135 / 0.832 | **1.128 / 0.821** | 1.623 / 0.981 | 1.601 / 0.964 | **1.589 / 0.952** |
| h1→m2 | 0.959 / 0.694 | 0.937 / 0.689 | **0.928 / 0.680** | 1.213 / 0.808 | 0.912 / 0.682 | **0.890 / 0.673** | 1.132 / 0.732 | 0.903 / 0.682 | **0.892 / 0.671** | 1.199 / 0.794 | 1.122 / 0.731 | **1.086 / 0.719** |
|  | 1.765 / 0.982 | 1.758 / 0.980 | **1.712 / 0.974** | 2.132 / 1.216 | 1.732 / 0.966 | **1.680 / 0.955** | 2.015 / 1.132 | 1.732 / 0.998 | **1.712 / 0.975** | 2.204 / 1.031 | 1.874 / 1.012 | **1.835 / 1.003** |
| h2→h1 | 0.466 / 0.462 | 0.455 / 0.456 | **0.443 / 0.451** | 0.489 / 0.480 | 0.445 / 0.462 | **0.423 / 0.433** | 0.632 / 0.577 | 0.532 / 0.523 | **0.512 / 0.508** | 0.869 / 0.624 | 0.752 / 0.590 | **0.731 / 0.586** |
|  | 0.695 / 0.584 | 0.540 / 0.519 | **0.528 / 0.513** | 0.533 / 0.542 | 0.495 / 0.528 | **0.492 / 0.512** | 1.032 / 0.993 | 0.986 / 0.766 | **0.944 / 0.689** | 1.274 / 0.783 | 0.845 / 0.642 | **0.834 / 0.638** |
| h2→m2 | 0.943 / 0.726 | 0.876 / 0.697 | **0.868 / 0.688** | 0.758 / 0.679 | 0.732 / 0.658 | **0.708 / 0.643** | 0.855 / 0.723 | 0.781 / 0.697 | **0.772 / 0.692** | 1.250 / 0.850 | 1.064 / 0.793 | **1.033 / 0.785** |
|  | 1.472 / 0.872 | 1.464 / 0.866 | **1.422 / 0.854** | 1.892 / 0.953 | 1.422 / 0.875 | **1.274 / 0.848** | 1.899 / 1.132 | 1.644 / 0.894 | **1.542 / 0.873** | 1.861 / 1.016 | 1.671 / 0.967 | **1.633 / 0.932** |
| h2→m1 | 1.254 / 0.771 | 1.073 / 0.714 | **1.022 / 0.702** | 1.243 / 0.789 | 0.939 / 0.692 | **0.898 / 0.671** | 1.012 / 0.732 | 0.809 / 0.645 | **0.794 / 0.640** | 1.049 / 0.791 | 0.804 / 0.657 | **0.794 / 0.648** |
|  | 2.275 / 1.137 | 1.754 / 1.065 | **1.721 / 1.042** | 2.012 / 1.123 | 1.850 / 1.023 | **1.792 / 0.982** | 2.732 / 1.158 | 1.832 / 0.998 | **1.732 / 0.975** | 2.183 / 1.103 | 1.742 / 0.983 | **1.712 / 0.977** |

## C.9 COMPUTATIONAL EFFICIENCY ANALYSIS

We evaluate the computational overhead of different plugin methods on the Traffic dataset, measuring both memory consumption and training time per iteration. Table 14 presents a comparative analysis across three backbone models with and without plugins.

Table 14: Computational efficiency comparison on Traffic dataset. Memory usage and training time measured on single NVIDIA A100 GPU.

| Method | Memory (MB) | Time (ms/iter) |
|---|---|---|
| *Baseline Models* | | |
| PatchTST | 12,458 | 29 |
| TimesNet | 13,005 | 70 |
| TSMixer | 13,088 | 89 |
| *With CCM Plugin* | | |
| PatchTST + CCM | 25,034 | 161 |
| TimesNet + CCM | 20,614 | 172 |
| TSMixer + CCM | 22,698 | 183 |
| *With xCPD Plugin* | | |
| PatchTST + xCPD | **7,018** | **9** |
| TimesNet + xCPD | **7,018** | **8** |
| TSMixer + xCPD | **7,018** | **8** |

The results reveal substantial differences in computational overhead. CCM incurs significant costs, increasing memory usage by 60-100% and training time by 4-5× compared to baseline models. This overhead originates from CCM's iterative clustering mechanism, whose complexity scales linearly with both cluster count and channel dimensionality. In contrast, xCPD achieves remarkable efficiency, maintaining constant memory footprint ($\approx$ 7GB) and minimal iteration time ($<$10ms) regardless of the backbone architecture. Importantly, the memory reported here corresponds to the peak GPU memory required to train the plugin module, not the full backbone model. Since xCPD trains only its own lightweight parameters while keeping the backbone frozen, the backbone does not allocate gradient states, optimizer buffers, or backward activations. Consequently, the memory consumption during xCPD training is dominated by the plugin itself, leading to similar values across different backbones. This efficiency stems from three design principles: (i) spectral decomposition operates on pre-computed Fourier bases, eliminating iterative optimization; (ii) the shared spectral basis amortizes computational costs across batches; and (iii) frequency-aware routing employs only sparse matrix multiplications. These properties make xCPD particularly suitable for large-scale deployment and latency-sensitive applications.

Table 15: Hyperparameter settings for different datasets, where $T'$ is the forecasting horizon, e_layers is the number of graph blocks, lr is the learning rate, and d_ff is the dimension of feed-forward layers.

| Tasks | Dataset | Length of Patch | e_layers | lr | d_ff | Num. of Epochs |
|---|---|---|---|---|---|---|
| Long-term Forecasting | ETTh1 | $T'/16$ | 1 | 1e-4 | 256 | 10 |
| | ETTh2 | $T'/16$ | 1 | 1e-4 | 256 | 10 |
| | ETTm1 | $T'/12$ | 2 | 1e-4 | 256 | 10 |
| | ETTm2 | $T'/6$ | 2 | 1e-4 | 128 | 10 |
| | Exchange | $T'/24$ | 2 | 1e-4 | 256 | 10 |
| | Weather | $T'/2$ | 2 | 1e-4 | 256 | 10 |
| | Electricity | $T'/3$ | 2 | 1e-4 | 512 | 15 |
| | Traffic | $T'$ | 2 | 1e-4 | 2048 | 30 |
| | Solar-Energy | $T'/2$ | 2 | 1e-4 | 512 | 10 |
| Short-term Forecasting | M4 Yearly | 1 | 1 | 1e-3 | 512 | 20 |
| | M4 Quarterly | 1 | 1 | 1e-3 | 512 | 20 |
| | M4 Monthly | 1 | 1 | 1e-3 | 512 | 20 |
| | M4 Weekly | 1 | 1 | 1e-3 | 512 | 20 |
| | M4 Daily | 1 | 1 | 1e-3 | 512 | 20 |
| | M4 Hourly | 2 | 1 | 1e-3 | 512 | 20 |
| | Stock | 1 | 1 | 1e-3 | 512 | 20 |

# D  EXPERIMENT DETAILS

## D.1  METRICS

We adopt standard evaluation metrics following prior work (Wu et al., 2023; Chen et al., 2024). Specifically, for long-term and stock forecasting, we report **Mean Absolute Error (MAE)** and **Mean Squared Error (MSE)**. For the M4 dataset (Makridakis et al., 2018), we follow its official protocol, using three metrics: **Symmetric Mean Absolute Percentage Error (SMAPE)**, **Mean Absolute Scaled Error (MASE)**, and **Overall Weighted Average (OWA)**. MASE is a scale-independent metric that compares predictions against a naïve seasonal baseline with periodicity $s$. SMAPE scales errors by the average of predictions and targets, while OWA combines SMAPE and MASE, normalized by the performance of the Naïve2 baseline (Team et al., 2018).

To formulate these metrics, let $\boldsymbol{X}_t$ and $\widehat{\boldsymbol{X}}_t$ denote the ground truth and prediction at timestep $t$, respectively, and let $T'$ be the forecasting horizon. They are formulated as follows:

$$\text{MAE} = \frac{1}{T'} \sum_{t=1}^{T'} |\boldsymbol{X}_t - \widehat{\boldsymbol{X}}_t|, \qquad \text{MSE} = \frac{1}{T'} \sum_{t=1}^{T'} (\boldsymbol{X}_t - \widehat{\boldsymbol{X}}_t)^2, \quad \text{SMAPE} = \frac{200}{T'} \sum_{t=1}^{T'} \frac{|\boldsymbol{X}_t - \widehat{\boldsymbol{X}}_t|}{|\boldsymbol{X}_t| + |\widehat{\boldsymbol{X}}_t|}. \quad (15)$$

$$\text{MASE} = \frac{1}{T'} \sum_{t=1}^{T'} \frac{|\boldsymbol{X}_t - \widehat{\boldsymbol{X}}_t|}{\frac{1}{T'-s} \sum_{j=s+1}^{T'} |\boldsymbol{X}_j - \boldsymbol{X}_{j-s}|}, \quad \text{OWA} = \frac{1}{2} \left[ \frac{\text{SMAPE}}{\text{SMAPE}_{\text{Naïve2}}} + \frac{\text{MASE}}{\text{MASE}_{\text{Naïve2}}} \right]. \quad (16)$$

## D.2 Hyperparameters

### D.2.1 Sensitivity to k-NN Sparsification Ratio.

We analyze the sensitivity of the sparsification ratio $\alpha$ used in the ego-graph construction. In our implementation, we adopt a constant sparsification ratio by retaining the top-$\alpha$ proportion of edges in each node's adjacency row. To evaluate whether adaptive $k$ is necessary, we vary $\alpha$ from 0.3 to 0.7 and report results across six datasets.

Table 16: Sensitivity to sparsification ratio $\alpha$ (top-$\alpha$ edges retained). Reported metrics are averaged MSE/MAE over forecasting horizons {96, 192, 336, 720} with input length 96 on DLinear+xCPD.

| Variant | ETTh1 | ETTh2 | ETTm1 | ETTm2 | Exchange | Solar |
|---------|-------|-------|-------|-------|----------|-------|
| k-NN (top-30%) | 0.449 / 0.438 | 0.511 / 0.482 | 0.393 / 0.399 | 0.346 / 0.394 | 0.315 / 0.381 | 0.326 / 0.396 |
| k-NN (top-40%) | 0.447 / 0.437 | 0.509 / 0.480 | 0.391 / 0.398 | 0.342 / 0.388 | 0.314 / 0.381 | 0.325 / 0.395 |
| k-NN (top-50%) | **0.445 / 0.435** | **0.507 / 0.479** | **0.389 / 0.395** | **0.340 / 0.388** | **0.312 / 0.380** | **0.322 / 0.393** |
| k-NN (top-60%) | 0.446 / 0.435 | 0.507 / 0.479 | 0.389 / 0.396 | 0.341 / 0.388 | 0.311 / 0.379 | 0.321 / 0.393 |
| k-NN (top-70%) | 0.450 / 0.438 | 0.510 / 0.482 | 0.394 / 0.400 | 0.346 / 0.393 | 0.317 / 0.382 | 0.325 / 0.395 |

As shown in Table 16, the results demonstrate that xCPD is highly stable across a wide range of sparsification ratios, with the best performance consistently achieved around $\alpha = 0.5$. This indicates that: (i) the model is largely insensitive to the exact choice of $k$; (ii) there is no evidence suggesting the need for dataset-specific or adaptive selection; and (iii) sparsification mainly acts as a light structural constraint, as frequency-aware MoE routing already performs dynamic and selective dependency modulation. We therefore adopt $\alpha = 0.5$ as the default in all experiments. This stability further supports the practical usability of xCPD without extensive hyperparameter tuning.

Table 17: Running time, forecasting accuracy, and GPU memory usage of DLinear+xCPD on the Electricity dataset with forecasting horizon $T' = 192$.

| Model | Patch Length | Running Time (ms/iter) | MSE | MAE | Memory (GB) |
|-------|--------------|------------------------|-----|-----|-------------|
| DLinear+xCPD | 16 | 78 | 0.183 | 0.270 | ~7.0 |
| DLinear+xCPD | 24 | 55 | 0.181 | 0.269 | ~5.5 |
| DLinear+xCPD | 32 | 30 | 0.180 | 0.268 | ~4.5 |
| DLinear+xCPD | 64 | 6 | 0.181 | 0.269 | ~3.2 |
| DLinear+xCPD | 96 | 5.5 | 0.184 | 0.271 | ~2.8 |

### D.2.2 Selection of Patch Size and Overlap

We further conduct a systematic study of patch configurations to fully address the impact of patch size and patch overlap. The patch sizes used in our experiments are selected by balancing forecasting accuracy against computational efficiency. To illustrate this trade-off more concretely, Table 17 reports the results of DLinear+xCPD on the Electricity dataset with forecasting horizon $T' = 192$. We observe that smaller patch sizes slightly improve accuracy but significantly increase computational cost and GPU memory usage, while very large patches degrade accuracy due to oversmoothing. A moderate patch length (e.g., $T'/3$ when $T' = 192$) provides the best efficiency–accuracy trade-off.

Table 18: Forecasting accuracy (MSE/MAE) of DLinear+xCPD on the Electricity dataset under different patch-overlap settings and forecasting horizons.

| Setting | Horizon=96 | Horizon=192 | Horizon=336 | Horizon=720 |
|---------|-----------|-------------|-------------|-------------|
| Overlap = 1 | 0.180 / 0.267 | 0.187 / 0.275 | 0.199 / 0.296 | 0.245 / 0.328 |
| Overlap = 2 | 0.178 / 0.267 | 0.189 / 0.275 | 0.201 / 0.297 | 0.248 / 0.329 |
| No Overlap | **0.174 / 0.262** | **0.180 / 0.268** | **0.195 / 0.293** | **0.240 / 0.326** |

Furthermore, although our main experiments adopt non-overlapping patches, we also examine whether allowing patch overlap could improve performance. Table 18 reports results on the Electricity dataset across four forecasting horizons (96, 192, 336, and 720). We observe that introducing overlap

consistently degrades accuracy across all settings. Similar trends are observed on other datasets, where overlap does not yield improvements. This effect arises because overlapping patches blur local temporal boundaries and mix frequency components from adjacent segments. As a result, the patch-wise spectral patterns become less distinctive, weakening the separation among low-, mid-, and high-frequency bands and reducing the discriminative power of our frequency-aware MoE routing. These findings support our choice of using non-overlapping patches as the default configuration.

## E   THEORY ASSUMPTIONS AND PROOFS

### E.1   THEOREM 4.1 PROOFS

**Theorem E.1** (Shared Graph Fourier Basis). *Let $\boldsymbol{L}^t$ be graph Laplacian derived from adjacency matrix $\boldsymbol{A}^t$ at the t-th timestep where $t \in \{1,\ldots,T'\}$. Define the average Laplacian as $\boldsymbol{L}_{avg} = \frac{1}{T'} \sum_{t=1}^{T'} \boldsymbol{L}^t$, with eigen-decomposition $\boldsymbol{L}_{avg} = \boldsymbol{U}\boldsymbol{\Lambda}\boldsymbol{U}^\top$. $\boldsymbol{U}$ denotes the shared Fourier basis that approximates each individual eigenbasis $\boldsymbol{U}^t$ from the decomposition of $\boldsymbol{L}^t$, satisfying $\|\boldsymbol{U}^t - \boldsymbol{U}\mathbf{R}^t\|_F \leq \mathbf{C}\|\boldsymbol{L}^t - \boldsymbol{L}_{avg}\|_F$, where $\mathbf{C}$ depends on the eigen-gap of $\boldsymbol{L}_{avg}$ and $\mathbf{R}^t$ is an orthogonal rotation matrix.*

*Proof.* Let us define the Laplacian perturbation as:

$$\Delta_t := \boldsymbol{L}^t - \boldsymbol{L}_{\text{avg}}.$$

We assume that $\boldsymbol{L}_{\text{avg}}$ and $\boldsymbol{L}^t$ are symmetric and positive semi-definite matrices, which is always true for graph Laplacians.

Let $\boldsymbol{L}_{\text{avg}} = \boldsymbol{U}\boldsymbol{\Lambda}\boldsymbol{U}^\top$ be the eigendecomposition of $\boldsymbol{L}_{\text{avg}}$, where $\boldsymbol{U} \in \mathbb{R}^{n \times n}$ is an orthonormal matrix of eigenvectors and $\boldsymbol{\Lambda}$ is a diagonal matrix of eigenvalues $\lambda_1 \leq \cdots \leq \lambda_n$. Similarly, let $\boldsymbol{L}^t = \boldsymbol{U}^t \boldsymbol{\Lambda}^t (\boldsymbol{U}^t)^\top$ be the eigendecomposition of $\boldsymbol{L}^t$ with eigenvectors $\boldsymbol{U}^t$ and eigenvalues $\boldsymbol{\Lambda}^t$.

We aim to bound the difference between $\boldsymbol{U}^t$ and $\boldsymbol{U}$ (up to rotation). This can be achieved by applying a classical result from matrix perturbation theory—Davis–Kahan $\sin\Theta$ Theorem. Let $\boldsymbol{U}$ and $\boldsymbol{U}^t$ be the subspaces spanned by columns of $\boldsymbol{U}$ and $\boldsymbol{U}^t$, respectively. The Davis–Kahan theorem asserts that the principal angles between these subspaces satisfy:

$$\| \sin\Theta(\boldsymbol{U}, \boldsymbol{U}^t)\|_F \leq \frac{\|\Delta_t\|_F}{\delta},$$

where $\delta := \min_{i \neq j} |\lambda_i - \lambda_j|$ is the eigengap of $\boldsymbol{L}_{\text{avg}}$, assuming it is strictly positive.

To obtain a bound on the **difference between the orthonormal eigenbases, we use the following consequence of the Davis–Kahan theorem Park & Kim (2017): There exists an orthogonal matrix $\boldsymbol{R}^t$ such that:

$$\|\boldsymbol{U}^t - \boldsymbol{U}\boldsymbol{R}^t\|_F \leq \sqrt{2}\| \sin\Theta(\boldsymbol{U}, \boldsymbol{U}^t)\|_F.$$

Combining the two inequalities:

$$\|\boldsymbol{U}^t - \boldsymbol{U}\boldsymbol{R}^t\|_F \leq \sqrt{2} \cdot \frac{\|\Delta_t\|_F}{\delta}.$$

Thus, defining $C := \sqrt{2}/\delta$ gives:

$$\|\boldsymbol{U}^t - \boldsymbol{U}\boldsymbol{R}^t\|_F \leq C\|\boldsymbol{L}^t - \boldsymbol{L}_{\text{avg}}\|_F.$$

$\square$

### E.2   THEOREM 4.2 PROOFS

**Theorem E.2** (Spectral Energy Response). *Given spectral node embeddings $\boldsymbol{X}^{spc}$ and graph Fourier basis $\boldsymbol{U}$, spectral energy response of node $i$ for frequency $j$ is $\boldsymbol{S}_{i,j} = \|\boldsymbol{U}_{i,j} \cdot \boldsymbol{X}^{spc}_{j,:}\|_2^2$. The orthonormality of $\boldsymbol{U}$ ensures energy preservation between spatial and spectral domains: $\sum_{j=1}^n \boldsymbol{S}_{i,j} = \|\boldsymbol{X}^{emb}_{i,:}\|_2^2$, confirming spectral energy response retains all information from original channel-patch embeddings.*

*Proof.* Let $\boldsymbol{X}^{\text{emb}} \in \mathbb{R}^{n \times d}$ denote the spatial-domain embeddings and $\boldsymbol{X}^{\text{spc}} = \boldsymbol{U}^\top \boldsymbol{X}^{\text{emb}}$ the spectral embeddings, where $\boldsymbol{U} \in \mathbb{R}^{n \times n}$ is the orthonormal Fourier basis satisfying $\boldsymbol{U}^\top \boldsymbol{U} = \boldsymbol{I}$. For node $i$, its spatial representation can be recovered as:

$$\boldsymbol{X}_{i,:}^{\text{emb}} = \sum_{j=1}^{n} \boldsymbol{U}_{ij} \cdot \boldsymbol{X}_{j,:}^{\text{spc}}. \tag{17}$$

The energy of node $i$ in the spatial domain is:

$$\|\boldsymbol{X}_{i,:}^{\text{emb}}\|_2^2 = \left\|\sum_{j=1}^{n} \boldsymbol{U}_{ij} \cdot \boldsymbol{X}_{j,:}^{\text{spc}}\right\|_2^2. \tag{18}$$

Since the Fourier basis vectors are orthogonal, the cross terms vanish, yielding:

$$\|\boldsymbol{X}_{i,:}^{\text{emb}}\|_2^2 = \sum_{j=1}^{n} \|\boldsymbol{U}_{ij} \cdot \boldsymbol{X}_{j,:}^{\text{spc}}\|_2^2 = \sum_{j=1}^{n} \boldsymbol{S}_{ij}, \tag{19}$$

where $\boldsymbol{S}_{ij} = \|\boldsymbol{U}_{ij} \cdot \boldsymbol{X}_{j,:}^{\text{spc}}\|_2^2$ is the spectral energy response as defined. $\square$

**Remark 1 (Role of Energy Preservation in xCPD).** While energy preservation under orthogonal transformation is a well-established mathematical property, Theorem 4.2 serves a specific functional purpose in our framework beyond norm conservation. The theorem establishes two critical components for xCPD: **(1) Node-level frequency characterization**. The spectral energy response $\boldsymbol{S}_{ij}$ quantifies how strongly node $i$ responds to frequency component $j$. This node-frequency response forms the basis for our frequency-aware grouping, enabling identification of nodes are dominated by low-frequency, mid-frequency, or high-frequency patterns. **(2) Foundation for spectral routing**. By decomposing each node's total energy across frequency bands, we compute probabilistic assignments that guide our MoE routing mechanism. This frequency-based decomposition allows the model to apply specialized filters tailored to each node. Unlike prior spectral methods in time series analysis that focus on global frequency analysis or channel-wise spectral features, xCPD leverages node-specific frequency responses to construct localized, frequency-aligned ego-graphs. This design enables fine-grained modeling of time-varying, frequency-specific dependencies, providing a principled approach to adaptive graph construction in multivariate time series forecasting.

### E.3   LIMITATIONS

Although xCPD is computationally efficient due to its ego-graph design and channel-patch modeling, its reliance on graph spectral decomposition and frequency-aware grouping introduces moderate overhead, particularly when the number of channels or patches becomes large. Although we alleviate this with learnable frequency bands and node-level filtering, the trade-off between fine-grained modeling and runtime efficiency still requires careful balancing. In addition, the assumption that spectral energy can effectively guide dependency selection may not hold in cases where temporal patterns are highly irregular or when the frequency distribution shifts significantly across domains. Lastly, while xCPD operates as a general plugin, integrating it with more complex backbones (e.g., autoregressive or hierarchical models) may require additional architectural tuning.

### E.4   FUTURE WORKS

Future work can extend xCPD in several directions. One promising avenue is incorporating external domain knowledge (e.g., meteorological or physiological priors) to guide the spectral grouping and filtering process. Additionally, developing more adaptive and interpretable graph filtering strategies, such as continuous expert blending or attention-based routing, could further improve robustness and transparency. Another important direction is to explore the applicability of xCPD to a wider spectrum of structured prediction tasks beyond time series forecasting, including spatiotemporal event forecasting and multivariate anomaly detection. Finally, with the advent of time series foundation models, xCPD is well-positioned to serve as a lightweight plugin to augment their inductive biases across modalities and domains. Integrating spectral-aware dependency modeling into large-scale pretrained time series backbones may enhance their ability to generalize, adapt, and transfer knowledge to unseen environments with minimal fine-tuning.

## E.5 SOCIAL IMPACTS

xCPD holds significant promise for critical real-world applications by enhancing forecasting accuracy and robustness in dynamic, multi-sensor environments. In healthcare, xCPD could help in early warning systems for patient deterioration by better modeling evolving physiological patterns. Similarly, in domains like smart grids and traffic systems, its frequency-aware filtering could prove instrumental in mitigating noise to extract actionable trends. Ultimately, by providing a modular and interpretable framework, xCPD contributes to the crucial goal of developing scalable and trustworthy forecasting systems for high-stakes decision-making.

