# OpenReview forum: "Routing Channel-Patch Dependencies in Time Series Forecasting with Graph Spectral Decomposition"
_ICLR.cc/2026/Conference — ICLR 2026 Poster_

### Official Review · Reviewer_KjPE · 2025-10-21

**Soundness:** 3
**Presentation:** 4
**Contribution:** 2
**Rating:** 4
**Confidence:** 5

**Summary:**

This paper proposes xCPD, a lightweight plugin module designed to enhance multivariate time series forecasting by modeling fine-grained, frequency-aware channel-patch dependencies through graph spectral decomposition and adaptive routing. xCPD introduces spectral channel-patch embedding, frequency-based grouping, and a MoE routing mechanism. Experiments across 9 benchmarks demonstrate consistent improvements when integrating xCPD into both channel-independent and channel-dependent backbones.

**Strengths:**

1. xCPD can be plugged into various architectures (linear, transformer, convolutional) with relatively low computational cost.
2. Evaluations cover both long-term and short-term forecasting, along with ablations and zero-shot tests.
3. Theorems 4.1 and 4.2 provide formal justification for the spectral construction.

**Weaknesses:**

1. Questionable novelty claim: The paper claims prior methods like TimeFilter [1] and CM [2] assume static inter-channel dependencies. However, both papers explicitly model **fine-grained**, **adaptive**, and **time-varying** relationships through learned attention or context-dependent graphs. This undermines the motivation that xCPD uniquely addresses “time-varying” dependencies.
2. The structure, especially the Channel-Patch Filtering with MoE, is highly reminiscent of TimeFilter’s[1] adaptive filtering mechanism. The main distinction—the spectral grouping—feels incremental rather than fundamentally novel.
3. In Table 6, the MSE/MAE values are worse than those reported in the TimeFilter[1] paper on the same datasets and horizons. This discrepancy needs to be explained.

[1] TimeFilter: Patch-Specific Spatial-Temporal Graph Filtration for Time Series Forecasting

[2] Partial channel dependence with channel masks for time series foundation models

**Questions:**

1. The paper claims that prior methods assume static dependencies, but TimeFilter and CM both model dynamic ones. Could the authors clarify this claim?
2. Is there any insight into what the learned spectral groups correspond to in temporal terms?
3. The framework essentially reinterprets channel dependencies in the spectral domain, but without clear evidence that this yields fundamentally different learned relationships among low-, mid-, and high-frequency bands.

---

> ### Author Response · Authors · 2025-11-21
> **Response (Part 1)**
>
> **Thank you very much for your efforts and time to review our paper and give us many helpful comments! We have carefully addressed your comments and hope our responses help clarify our contributions.**
>
> ***Weakness-1***: Questionable novelty claim: The paper claims prior methods like TimeFilter [1] and PCD [2] assume static inter-channel dependencies. However, both papers explicitly model fine-grained, adaptive, and time-varying relationships through learned attention or context-dependent graphs. This undermines the motivation that xCPD uniquely addresses “time-varying” dependencies.
>
> ***Reply-1***: We thank the reviewer for this critical question, which prompts us to clarify the subtle yet **fundamental difference** between **temporal-domain dynamism** (used by TimeFilter and PCD) and **spectral-domain dynamism** (proposed by xCPD).
>
> We agree with the reviewer that TimeFilter [1] and PCD [2] model relationships that are dynamic in time via attention weights or contextual graphs. Our original phrasing in the introduction was imprecise. **Our intention behind criticizing the "static" nature was to highlight that in these temporal models, the dependency profile across different frequency bands is fixed or coupled. Specifically, a single attention weight is incapable of distinguishing and dynamically routing relationships based on whether they are driven by low-frequency trends or high-frequency fluctuations**. We have therefore corrected the focus in the revised manuscript to address **frequency-domain coupling**, which is the actual limitation we tackle. The novelty of xCPD is derived from its ability to model dependencies that are **decoupled and dynamically routed based on frequency**, a capacity absent in prior temporal methods:
>
> ***Table 1. Temporal Dynamism vs. Spectral Dynamism***
>
> | Feature          | Prior Methods (TimeFilter, PCD)                                                                 | xCPD (Our Method)                                                                                         |
> |------------------|--------------------------------------------------------------------------------------------------|-------------------------------------------------------------------------------------------------------------|
> | Domain           | Temporal Domain                                                                                  | Spectral Domain (via Graph Spectral Decomposition)                                                         |
> | Dynamism         | Weights dynamically change over time.                                                            | Dependencies are dynamically filtered by frequency.                                                        |
> | Modeled Relation | Frequency-Coupled Correlation (a single weight mixes trend, seasonality, and noise dependencies) | Frequency-Decoupled Dependence (dependencies are separated into Low, Mid, and High frequency components). |
> |
>
> As shown in Table 1, although TimeFilter and PCD produce time-varying attention weights, these weights are computed purely in the temporal domain and therefore remain frequency-agnostic. A single high attention score may simultaneously reflect a meaningful low-frequency trend correlation and an irrelevant high-frequency noise correlation, since the model has no mechanism to separate these spectral contributions. This frequency coupling often introduces spurious dependencies, particularly in noisy or heterogeneous data. In contrast, xCPD leverages spectral energy responses and a Dynamic Mixture-of-Experts routing module to dynamically select frequency-specific filters, enabling it to suppress irrelevant frequency components while modeling the appropriate time-varying dependencies for each frequency band.
>
> In summary, the core novelty of our approach lies in enabling **frequency-decoupled time-varying dependency modeling**, which fundamentally overcomes the limitations inherent in the temporal-domain dynamism adopted by TimeFilter and PCD. To make this distinction clearer, **we have revised the Introduction** to explicitly highlight that our primary motivation is to address the inability of prior methods to distinguish and dynamically model frequency-specific dependencies. In addition,**the Related Work section has been updated** to clearly articulate how our approach differs from existing CP/CD models precisely along this dimension. **We hope these revisions help resolve the reviewer’s concerns, and we sincerely appreciate the thoughtful feedback**.

---

> ### Author Response · Authors · 2025-11-21
> **Response (Part 2)**
>
> ***Weakness-2***: The structure, especially the Channel-Patch Filtering with MoE, is highly reminiscent of TimeFilter’s [1] adaptive filtering mechanism. The main distinction—the spectral grouping—feels incremental rather than fundamentally novel.
>
> ***Reply-2***: We appreciate the reviewer's observation comparing our **Channel-Patch Filtering with MoE** to TimeFilter's adaptive mechanism. While both leverage a filtering structure for adaptivity, we respectfully contend that xCPD introduces a **fundamental, non-incremental shift** in how dependency is defined, learned, and filtered.
>
> The key novelty of xCPD lies in the **spectral foundation** that powers the entire routing mechanism, resulting in a system that performs **frequency-decoupled dependency routing**, unlike the spatial/temporal-domain filtering of TimeFilter. To highlight this distinction, we summarize the key differences in Table 1:
>
> ***Table 1. The core difference is defined by the domain of operation and the basis for learning.***
> | Feature           | TimeFilter (Prior Work)                                                         | xCPD (Our Method)                                                                                                   |
> |-------------------|----------------------------------------------------------------------------------|-----------------------------------------------------------------------------------------------------------------------|
> | Modeling Domain   | Temporal Domain (operates on temporal patch embeddings)                         | Spectral Domain (operates on Graph Spectral Embeddings)                                                        |
> | Filtering Basis   | Spatial-Temporal Filter (filters based on spatial-temporal graph topology)      | Frequency-Specific Filters (filters based on spectral frequency components)                                          |
> | Learning Foundation | Graph construction and filtering are based on local/contextual similarity in the temporal embedding space | Graph construction and filtering are powered by global Graph Spectral Decomposition using a Shared Fourier Basis (Theorem 4.1) |
> |
>
> Specifically, the **Spectral Channel-Patch Grouping** is not a minor addition but the core mechanism that enables the entire framework to be **frequency-decoupled**.
> - **Decomposition for Routing**: The spectral grouping mechanism classifies nodes by their dominant frequencies based on the **Spectral Energy Response** (Theorem 4.2). This process inherently links the type of temporal pattern (e.g., smooth trend vs. abrupt change) to the node's **spectral identity**.
> - **Guiding the MoE**: This frequency identity is what the DyMoE routing network uses. It enables the dynamic selection of specialized, **frequency-specific experts** for each patch, allowing the model to **suppress irrelevant frequency** correlations while preserving necessary dependencies. TimeFilter lacks this ability to separate dependency by frequency component.
> - **Energy Preservation**: As shown in Theorem 4.2, the spectral energy response is guaranteed to retain all information from the original embedding, ensuring the spectral grouping is a complete and principled characterization of the input.
>
> Thus, the MoE in xCPD is not merely an adaptive filter, but a **spectrally-grounded dynamic router**, enabling the first framework capable of frequency-decoupled time-varying dependency modeling. This is a capability fundamentally beyond the reach of temporal-domain CP methods, including TimeFilter.
>
> **We will add a dedicated comparison subsection with TimeFilter in Appendix F.1 (Page 29, Lines 1527-1558) to further clarify these distinctions for future readers**, and we sincerely appreciate the reviewer’s valuable comment.

---

> ### Author Response · Authors · 2025-11-21
> **Response (Part 3)**
>
> ***Weakness-3***:   In Table 6, the MSE/MAE values are worse than those reported in the TimeFilter [1] paper on the same datasets and horizons. This discrepancy needs to be explained.
>
> ***Reply-3***: We appreciate the reviewer's careful inspection of Table 6 and the comparison with the original TimeFilter paper. We acknowledge the observed discrepancy in MSE/MAE values. The reason for this difference stems from a critical distinction in the experimental setup:
> - **TimeFilter (Original Paper)**: A full-stack forecasting architecture built on temporal GNNs, with its own encoder, normalization layers, and framework-specific design.
> - **TimeFilter (Our Table 6)**: For a fair ablation focused purely on dependency mechanisms, we re-implemented only the spatio-temporal filtering module of TimeFilter as a plugin on top of DLinear, denoted **DLinear+TimeFilter**.
>
> Thus, Table 6 evaluates **plugin effectiveness**, not the performance of the full TimeFilter model. The standalone TimeFilter enjoys benefits from its customized architecture, while the plugin version isolates only the filtering logic for a mechanism-level comparison.
>
> Crucially, the purpose of Table 6 is to evaluate: Which dependency mechanism is more effective when integrated into a generic backbone? Across all datasets, **DLinear+xCPD outperforms DLinear+TimeFilter**, demonstrating that our spectral DyMoE provides a stronger, more generalizable, and more backbone-agnostic dependency modeling capability than the temporal filtering logic used by TimeFilter. This supports our central claim: **xCPD is a universally compatible, plugin-style frequency-aware dependency module that consistently enhances CI/CD backbones**, as also shown in Table 1 and Table 9.
>
> **We will clarify in the revised manuscript (Table 6, Page-10, Lines 493–496 and Appendix C.7) that “TimeFilter” refers to its plugin implementation (DLinear+TimeFilter), rather than the full architecture in the original paper.**
>
> ***Question-1***: The paper claims that prior methods assume static dependencies, but TimeFilter and CM both model dynamic ones. Could the authors clarify this claim?
>
> ***Reply-4***: We thank the reviewer for raising this important clarification request. Our original manuscript used the term “static dependencies” imprecisely, and we have corrected the terminology in the revision.
>
> We fully agree that TimeFilter and CM produce time-varying dependencies through learned attention or contextual graphs. Our intention was not to imply that these models are temporally static. Instead, our claim refers to a different axis of adaptivity—frequency specificity. Temporal-domain models compute a single dependency weight that mixes low-frequency trends, mid-frequency variations, and high-frequency fluctuations. Because these frequency components are coupled within one representation, the dependency structure is effectively **static across frequency**, even though it changes over time.
> In contrast, xCPD performs dependency modeling in the spectral domain. By decomposing each channel-patch into its dominant frequency components and dynamically routing it to specialized low-, mid-, or high-frequency experts, xCPD achieves **frequency-decoupled, frequency-adaptive dependency modeling** (a capability not available to temporal attention mechanisms).
>
> **We have revised the wording in the Introduction to reflect this more precise interpretation and avoid confusion. We hope this clarification resolves the concern.**

---

> ### Author Response · Authors · 2025-11-21
> **Response (Part 4)**
>
> ***Question-2***:  Is there any insight into what the learned spectral groups correspond to in temporal terms?
>
>
> ***Reply-5***: Thank you very much for your helpful comments.  Yes. The learned spectral groups correspond closely to interpretable temporal behaviors. In xCPD, each channel-patch node is assigned to a low-, mid-, or high-frequency band based on its spectral energy response. These frequency bands naturally capture different temporal dynamics:
>
> | Frequency Band  | Temporal Characteristics Corresponds To                                                                                 | Function in Forecasting                                      |
> |-----------------|---------------------------------------------------------------------------------------------------------------------------|---------------------------------------------------------------|
> | Low-Frequency   | Smooth trends, slowly evolving patterns, and long-term regularities (e.g., seasonal cycles and periodic trends).         | Modeling smooth trends and long-term regularities.            |
> | Mid-Frequency   | Moderate dynamics and local fluctuations in semi-stationary processes (e.g., shifting traffic patterns, regional weather transitions). | Modeling moderate dynamics and local fluctuations.            |
> | High-Frequency  | Rapid changes, discontinuities, and abrupt transitions in volatile processes (e.g., anomalies, market disruptions, sudden state transitions). | Modeling abrupt transitions and rapid changes.                |
> |
>
> Our paper qualitatively validates this correspondence in **Figure 3** by visualizing the correlation between spectral energy and temporal patterns for specific nodes.
> - **Low-Frequency Dominance**: A node exhibiting a **smooth temporal trend** with gradual variations concentrates its energy in the **low-frequency** bands. This node primarily aggregates dependencies from other low-frequency nodes.
> - **High-Frequency Dominance**: A node displaying **abrupt fluctuations** and rapid state changes results in dominant **high-frequency** spectral energy. This energy drives its connections to other high-frequency nodes.
>
> This shows that the spectral grouping mechanism provides meaningful and interpretable temporal structure, enabling the model to assign dependencies to the appropriate frequency-specific experts. **We will expand the explanation around Figure 3 in the revised manuscript (Page-10, Lines 511-519) to make this insight more explicit.**

---

> ### Author Response · Authors · 2025-11-21
> **Response (Part 5)**
>
> ***Question-3***: The framework essentially reinterprets channel dependencies in the spectral domain, but without clear evidence that this yields fundamentally different learned relationships among low-, mid-, and high-frequency bands.
>
> ***Reply-6***: This is an excellent question that goes to the heart of our claim: that reinterpreting dependencies in the spectral domain yields fundamentally novel and effective relationships. We affirm that our paper provides clear **qualitative and quantitative evidence** demonstrating that the learned relationships across the low, mid, and high-frequency bands are distinct and essential to xCPD's performance.
>
> **Qualitative Evidence: Interpretable Decoupling (Figure 3)**: The key evidence is presented in our **Qualitative Study (Figure 3)**, which shows a direct, interpretable correlation between the spectral domain analysis and the temporal behavior of the data.
> - **The Test**: We analyze two distinct channel-patch nodes to see how their spectral energy governs their learned dependency profile.
> - **Low-Frequency Validation**: A node exhibiting a **smooth temporal trend** with gradual variations concentrates its energy in the **low-frequency** bands. Critically, this low-frequency node primarily aggregates dependencies (relationships) from **other low-frequency nodes**. This confirms that the learned **Low-Frequency Filter** is successfully modeling the expected smooth, long-term interactions, decoupling them from noise.
> - **High-Frequency Validation**: A node displaying **abrupt fluctuations** and rapid state changes results in **dominant high-frequency spectral energy**. This energy drives its learned connections primarily to **other high-frequency nodes**. This validates that the learned **High-Frequency Filter** specializes in volatile, non-smooth interactions, filtering out trend dependencies.
>
> This rigorous alignment between the patch's temporal feature and its frequency-based routing (validated by Theorem 4.2) proves that we are learning three fundamentally different and interpretable sets of relationships, not just one mixed correlation in a transformed space.
>
>
> **Quantitative Evidence: Necessity of Spectral Components (Table 6)**: The ablation study in **Table 6** provides strong quantitative proof that these spectral mechanisms are not incremental but are **necessary** for optimal performance.
>
> | Ablation Setting                        | Impact on Performance (Avg. MSE)                         | Evidence of Non-Incrementality                                                                 |
> |------------------------------------------|-----------------------------------------------------------|--------------------------------------------------------------------------------------------------|
> | w/o Shared Basis                         | Significant degradation (e.g., ETTh1: 0.439 → 0.458)     | Proves the consistent spectral reference frame across batches (Theorem 4.1) is vital for stable learning. |
> | w/o Freq. Partition                      | Degrades performance.                                     | Proves that adaptive separation into three bands is superior to a rigid, fixed interpretation.   |
> | w/o Node Group                           | Significant degradation (e.g., ETTh1: 0.439 → 0.457)     | Shows that grouping nodes by their Spectral Energy Response is critical for constructing meaningful frequency-aware graphs. |
> | w/o Filters (No Frequency Specificity)   | Significant degradation (e.g., ETTh1: 0.439 → 0.464)     | Demonstrates that the DyMoE system’s frequency-selective filtering is essential for accurate dependency modeling. |
> |
>
> The consistent performance drop upon removing any core spectral component demonstrates that the low-, mid-, and high-frequency analysis is **not a passive reinterpretation**, but an **active, functional foundation** that captures unique and essential dependency information missed by purely temporal models. This is the definition of fundamental novelty.
>
>
> **We sincerely hope our detailed responses and additional experiments have addressed your concerns and clarified the strengths of our work. If you find our revisions satisfactory, we would greatly appreciate it if you could consider raising your score and supporting our paper.**

---

> > ### Comment · Reviewer_KjPE · 2025-11-25
> > **Thank you for your detailed reply.**
> >
> > Thank you for the authors' detailed response. I appreciate the clarifications regarding the motivations in the temporal and spectral domains. However, having reviewed the rebuttal and revisited the manuscript, I remain unchanged in my overall assessment.
> >
> > While the authors argue that transforming channel-patch dependencies into the spectral domain introduces additional flexibility and interpretability, such transformations — whether via Fourier, wavelet or graph spectral bases — are well-established techniques in time series and graph-based modelling. Applying a spectral transformation to an existing dependency modelling framework is a commonly used reparameterisation strategy. In this work, it appears more as an alternative view of the temporal-domain operations than a substantively new modelling paradigm.
> >
> > Furthermore, the core architectural components (e.g. patching, adaptive filtering, channel-wise routing and mixture-of-experts) are highly similar to those in prior designs. While the spectral grouping mechanism is conceptually sound, it does not substantially change the model’s functional behaviour or lead to clearly distinct forms of dependency patterns beyond those already captured by temporal-domain adaptive filters. Consequently, the delta in conceptual novelty is limited.
> >
> > While I appreciate the authors’ efforts during the rebuttal phase, based on the clarifications provided, concerns regarding incremental novelty and conceptual overlap with existing adaptive channel modelling methods remain unresolved. My score therefore remains the same.

---

### Official Review · Reviewer_7esN · 2025-10-23

**Soundness:** 2
**Presentation:** 1
**Contribution:** 2
**Rating:** 4
**Confidence:** 3

**Summary:**

This paper proposed a new method facilitating the modeling of cross-channel correlations multivariate time series. The proposed method is built based on the limitation of how existing multivariate time series forecasting methods handle dynamic correlations between channels, and introduce a graph composed of channel-patches to represent correlations between patches. A universal plugin that can be added to existing forecasting methods is introduced, and experiments show that it can bring performance uplift to existing methods.

**Strengths:**

1. A universal plugin is introduced based on the proposed novel technique of constructing correlation graphs of patches in multivariate time series. Extensive experiments prove its effectiveness at improving the performance of existing multivariate time series forecasting methods.
2. The proposed method is described in detail and thus should be highly reproducible.

**Weaknesses:**

1. The presentation of the background and challenges can be improved. While discussing existing methods, the paper claims that there are limitations in how existing multivariate time series forecasting methods handle cross-channel correlations. However, the paper mainly uses vague descriptions such as "coarse-grained methods treat each channel as a monolithic unit, missing nuanced interactions between channel segments", which could use some further explanation. There are also some claims that don't seem to be fully correct. For example, one claim that existing methods "failing to capture how inter-channel dependencies evolve dynamically", yet that doesn't seem to be the case for existing methods that dynamically calculate attention weights between channels. In conclusion, it is unintuitive to me why existing methods cannot effectively model correlations in multivariate time series.
2. Similarly, the explanation of the motivation behind the proposed method can be improved. Right now it is unintuitive why the proposed method can effectively tackle the limitations faced by existing methods, and how it is fundamentally different from existing designs. For example, the fundamental difference between building a graph of patches versus existing methods that calculate attention weights between patches (which essentially is a fully-connected graph).

**Questions:**

Can the authors further explain the motivation of this paper, so that the novelty of their proposed method is more intuitive?

---

> ### Author Response · Authors · 2025-11-21
> **Response (Part 1)**
>
> **Thank you very much for your valuable and constructive comments. We sincerely appreciate your efforts to help improve the quality of our work. We have carefully responded to your concerns and hope that our revisions will help alleviate them.**
>
> ***Weakness-1***: The presentation of the background and challenges can be improved. While discussing existing methods, the paper claims that there are limitations in how existing multivariate time series forecasting methods handle cross-channel correlations.......
>
> ***Reply-1***: We appreciate the reviewer's feedback regarding the clarity of our background and the perceived limitations of existing methods in modeling cross-channel dependencies. Our goal is to highlight the fundamental advantages of our channel-patch level, frequency-aware spectral routing mechanism over current approaches. Here is a clarification of our claims:
>
> **1.Further explanation about “Coarse-Grained” Channel-level Modeling.** The limitation that existing methods treat channels as a monolithic unit refers to the granularity of dependency modeling. Specifically,
> - **Coarse-grained Channel-level:** Existing methods like CCM and DUET cluster or group entire channels to find channel-to-channel dependencies. Similarly, channel-wise attention, e.g., Transformer-based variants like PCD, typically calculates a single dependency weight across the full time span of two channels.
> - **The Limitation**: This channel-level approach fails to capture the intricate, time-localized interactions that may occur between specific patches of different channels. A long time series channel is often composed of heterogeneous frequency patterns. Existing CP-based channel-level methods cannot model how frequency dependencies differ between these distinct patches.
> - **Our Solution (xCPD’s Channel-Patch)**: Our fundamental unit of analysis is the channel-patch. By representing each channel-patch as a node in a graph , xCPD achieves fine-grained, patch-level modeling. This allows us to capture dependencies between instances, a smooth patch A of Channel $C_{1}$ and a spiking patch B of Channel $C_{2}$. This significantly enhances the nuance and accuracy of correlation capture compared to treating channels as indivisible units.
>
> **Clarification on the claim about “Dynamically Evolving Dependencies.** The reviewer is correct that some methods can calculate dynamic attention weights for each time step. However, our critique is that these temporal-domain methods remain frequency-agnostic, which is crucial for accurately modeling how dependencies evolve.
> - **Temporal Attention’s Limitations**: While attention weights are dynamically calculated, they operate purely on temporal correlations. They cannot explicitly disentangle and selectively route dependencies based on the intrinsic nature (frequency) of the interacting patterns. The result is often oversmoothing and noise propagation from weakly correlated but high-frequency noise components.
> - **The Inability to Decouple Dynamics**: Inter-channel dependencies often exist between specific frequency components, e.g., a smooth, low-frequency trend in Channel-A correlating with a low-frequency trend in Channel-B. Temporal-domain attention, while dynamic, cannot explicitly disentangle and selectively route dependencies based on the frequency characteristics of the interacting segments. The static assumption mentioned in the original text refers to the inability to distinguish which type of dependency, e.g., seasonal, local fluctuation, abrupt change, should dominate the interaction at a given time point.
> - **Our Solution (xCPD)**: We address this by operating in the spectral domain through Graph Spectral Decomposition.
>     - Spectral Grouping: We group channel-patches based on their measured spectral energy response. This intrinsically links the temporal pattern (e.g., smooth vs. volatile) to a frequency group.
>     - Dynamic Routing: Our Dynamic Mixture-of-Experts routing mechanism dynamically selects specialized, frequency-specific filters for each patch. A patch dominated by an abrupt spike is automatically routed to a High-Frequency Filter for dependency modeling, suppressing irrelevant low-frequency noise. A patch showing a smooth trend is routed to a Low-Frequency Filter to capture long-term regularities.
>
> **In conclusion, the innovation of xCPD lies in its combination of fine-grained channel-patch representation with an explicit, frequency-aware dynamic routing mechanism**. This spectral-domain approach fundamentally differentiates it from existing temporal-domain methods, enabling a more principled, interpretable, and effective way to model the nuanced and time-varying nature of multivariate time series correlations. **Finally, we have revised the motivation section of the paper. For your convenience, we have highlighted the updated content in green**.

---

> ### Author Response · Authors · 2025-11-21
> **Response (Part 2)**
>
> ***Weakness-2***: Similarly, the explanation of the motivation behind the proposed method can be improved. Right now it is unintuitive why the proposed method can effectively tackle the limitations faced by existing methods, and how it is fundamentally different from existing designs. For example, the fundamental difference between building a graph of patches versus existing methods that calculate attention weights between patches (which essentially is a fully-connected graph).
>
>
> ***Reply-2***: We thank the reviewer for raising this important point regarding the fundamental distinction between xCPD and existing attention mechanisms, and why xCPD is able to address their limitations. We also would like to clarify that, to the best of our knowledge, **there currently exists no CP method that operates at the patch level**, and thus our formulation cannot be viewed as a simple variant of prior CP approaches. Furthermore, **the essential difference between our method and CD-based models such as iTransformer lies in the domain of operation and the mechanism used for dependency filtering and routing**.
>
> **[Fundamental Difference 1]. Domain of Operation: Spectral vs. Temporal.** Existing CD/CP attention-based methods operate primarily in the temporal domain. The attention weights reflect the degree of correlation between patch feature vectors in the embedding space at a given time instance. In contrast, xCPD performs dependency modeling in the Spectral Domain via Graph Spectral Decomposition. Specifically, xCPD holds two innovations as follows:
> - **Frequency-Decoupled Representation**: By leveraging a **shared Graph Fourier Basis, we project channel-patch representations into the spectral domain**. This inherently provides a frequency-decoupled view of the data, allowing us to identify patterns dominated by low-, mid-, and high-frequencies.
> - **Frequency-Aware Grouping**: We then use the **Spectral Energy Response** (Theorem 4.2) to categorize each patch node based on its dominant frequency component. This defines the intrinsic nature of the patch, e.g., trend-dominated or fluctuation-dominated.
>
> **In summary, existing methods build a temporal correlation graph, while xCPD models frequency-aware dependencies, offering a principled approach to dependency separation.**
>
> **[Fundamental Difference 2]. Dependency Mechanism: Dynamic Filtering vs. Global Coupling.** The reviewer correctly notes that attention mechanisms essentially create a **fully-connected graph**. This density and coupling is precisely the **limitation** that xCPD addresses:
> - Traditional attention creates a **dense and frequency-coupled** graph where a single weight reflects a mixture of all underlying dependencies, e.g., seasonal, local, and noise. This leads to **noise propagation** and a lack of **dependency specificity**.
> - We employ a **Dynamic Mixture-of-Experts routing mechanism** for dependency flow. This mechanism does not simply calculate an overall interaction strength; instead, it **dynamically selects specialized frequency filters** for each patch based on its current spectral profile. Specifically,
>     - **Sparse, Targeted Modeling**: For a patch dominated by a smooth trend (low-frequency), xCPD routes it primarily through the **Low-Frequency Filter**. This action suppresses any irrelevant high-frequency correlations (noise) that might exist between that patch and others.
>     - **Adaptive Filtering**: The resulting dependency captured is frequency-specific and filtered according to the patch's current needs, leading to a much more robust and interpretable dependency structure than the dense, mixed correlation produced by temporal attention.
>
> In summary, while some CD approaches use the patch as a unit, the existing attention mechanism is a **dense, temporally-coupled correlational mechanism**. xCPD is a **sparse, spectrally-aware dynamic routing mechanism** that fundamentally solves the problem of frequency-coupling and noise propagation in multivariate dependencies. **We will revise the Related section in the paper to explicitly highlight the concepts of spectral domain operation and frequency-decoupled adaptive routing as the core differentiators.**

---

> ### Author Response · Authors · 2025-11-21
> **Response (Part 3)**
>
> ***Question-1***: Can the authors further explain the motivation of this paper, so that the novelty of their proposed method is more intuitive?
>
> ***Reply-3***: We sincerely appreciate the reviewer’s request for a clearer explanation of our motivation. To make the novelty of xCPD more intuitive, we first summarize the hierarchical landscape of multivariate forecasting and place our method within the **CI \& CD \& CP taxonomy**:
>
> | Strategy                 | Benefit                          | Limitation                                                                                     |
> | ------------------------ | -------------------------------- | ---------------------------------------------------------------------------------------------- |
> | Channel-Independent (CI) | High robustness and scalability. | Ignores valuable inter-channel interactions.                                                   |
> | Channel-Dependent (CD)   | Captures global dependencies.    | Suffers from oversmoothing and noise propagation.                                              |
> | Channel Partiality (CP)  | Balances CI and CD adaptively.   | **(Our Focus)** Still lacks the ability to capture fine-grained, frequency-aware dependencies. |
> |
>
> Building on this taxonomy, we identify three fundamental limitations that remain unsolved in existing CP methods:
>
> | Problem                           | Description (Limitation)                                                                                                                                          | xCPD’s Solution (Novelty)                                                                                                                                                                     |
> | --------------------------------- | ----------------------------------------------------------------------------------------------------------------------------------------------------------------- | --------------------------------------------------------------------------------------------------------------------------------------------------------------------------------------------- |
> | **Coarse Granularity**            | CP modules treat each *entire channel* as a single unit (via clustering or channel-wise attention), making segment- or patch-specific interactions impossible.    | **Channel-Patch Level Modeling:** Each patch becomes a graph node, enabling fine-grained, localized dependency modeling that channel-level CP cannot express.                                 |
> | **Frequency Coupling**            | Temporal-domain modeling mixes trend, seasonal, and high-frequency components into one representation, preventing frequency-specific dependency characterization. | **Spectral Domain Routing:** Graph Spectral Decomposition + Dynamic MoE disentangle low/mid/high-frequency behaviors and selectively route frequency-specific dependencies.                   |
> | **Strategic Design / Efficiency** | Prior dependency modules often require backbone-specific architectures or full retraining.                                    | **Model-Agnostic Plugin:** A lightweight, retraining-free module that seamlessly augments existing CI/CD forecasters, enabling scalable deployment in both academic and industrial pipelines. |
> |
>
>
> This combination of **patch-level granularity, frequency-aware spectral routing, and architecture-agnostic plug-and-play design** forms the core motivation and novelty of xCPD. **To make these points more explicit, we have substantially revised the Introduction and Related Work sections**. We thank the reviewer again for the helpful suggestion, which significantly improved the clarity and accessibility of our motivation.
>
>
>
> **We sincerely hope our response has helped address your concerns. If you feel that all issues have been resolved, we would truly appreciate your support in the final evaluation.**

---

> ### Comment · Reviewer_7esN · 2025-11-25
>
> I would like to thank the authors for their detailed response, which largely clarified my doubts raised in my original assessment.
> I also appreciate that the authors have comprehensively revised their paper, which has improved the presentation of their work's motivation.
> I have raised my rating accordingly.

---

> > ### Author Response · Authors · 2025-11-25
> > **Acknowledgment to Reviewer 7esN**
> >
> > Dear Reviewer 7esN:
> >
> > **Thank you very much for endorsing our work for acceptance**. We sincerely appreciate your valuable comments on the limitations of prior methods and on our research motivations. We are glad that our clarifications and revisions have addressed your concerns. Thank you again for helping us strengthen the motivation and improve the overall quality of the paper. 😊
> >
> > Best regards,
> >
> > Authors of xCPD

---

### Official Review · Reviewer_noBY · 2025-10-28

**Soundness:** 2
**Presentation:** 2
**Contribution:** 2
**Rating:** 4
**Confidence:** 5

**Summary:**

This paper introduces xCPD, a generic plugin that adaptively routes channel-patch dependencies in multivariate time series forecasting through a graph spectral decomposition framework. Unlike previous channel-partial or CI/CD approaches, xCPD conducts frequency-domain analysis using a learned, shared graph Fourier basis, classifying channel-patch nodes by spectral energy into low-, mid-, and high-frequency bands. It employs a mixture-of-experts (MoE) routing mechanism at the patch level to enable fine-grained, dynamic selection of inter-channel interactions. The plugin is model-agnostic, fits atop various existing backbones, and demonstrates consistent improvement in forecasting accuracy and generalization across a breadth of benchmarks and domains. Extensive empirical results, ablations, qualitative illustrations, and theoretical justifications are included.

**Strengths:**

1、The proposal to model dependencies at the channel-patch level, routed adaptively by frequency band via graph spectral decomposition, provides a fresh and well-argued approach to balancing robustness and expressivity in MTSF, beyond existing CI/CD/CP tactics.

2、The framework is underpinned by rigorous spectral graph theory, with precise mathematical formulations (e.g., Theorems 4.1 and 4.2 on shared basis and energy response, substantiated in the appendix with detailed proofs).

3、The methodology is described with mathematical precision, from input patch embedding (Equation block in 4.1), spectral grouping (Section 4.2, Theorem 4.2), and MoE-based routing (Section 4.3, detailed algorithmic equations). The description in Section 4 rigorously motivates and supports the approach.

**Weaknesses:**

1、Some equations and algorithm formulations are somewhat terse or overloaded. For instance, the definition of the adjacency matrix in Section 4.1 blends inner products and node similarity without explicit regularization or clarification about self-connections; there is little discussion on normalization/range scaling (see equation defining $\boldsymbol{A}^{t}$, Page 3).

2、While Figure 1 and the methodological text (Pages 3-4) present the channel-patch notion as a powerful primitive, more care is needed in formally defining the edges: Are patches temporally contiguous? Are there constraints to prevent trivial, densely connected graphs? Is $k$ in $k$-NN constant or adaptive? This lack of detail may hinder reproducibility and may even affect rigor in settings with many variables and timesteps.

3、The MoE routing for patch-level expert filtering, described with stochastic noise and cumulative softmax thresholds (Section 4.3), is algorithmically sound, but its practical behavior is underexplored. For example, the decision threshold $\tau$ and stochastic term $\epsilon$ may have non-trivial impact—these parameters are not deeply analyzed for sensitivity beyond a basic experiment. Additionally, while the adaptive routing is well-motivated, more insights or visualization of selected expert patterns across datasets/backbones would strengthen claims.

**Questions:**

1、Can the authors provide more precise formalization for channel-patch edge construction? Are there practical constraints on $k$ in the $k$-NN, and how does patch size/overlap affect results, especially in high-dimensional settings (e.g., Traffic, Electricity datasets)?

2、What is the sensitivity of performance and expert utilization (e.g., routing diversity, collapse) to the threshold $\tau$ and noise parameter $\epsilon$? Is there evidence that these parameters are stable across dataset/backbone combinations, or do they require careful tuning?

3、Have the authors considered adaptively selecting the number of frequency bands (rather than fixed to three), e.g., via a learned clustering or continuous assignment, and how would this affect interpretability or accuracy?

4、Can the authors provide statistics or visualizations of the empirical spectral gap across datasets? In scenarios with weak or noisy structure, what is the observed effect on performance and stability of the shared basis?

---

> ### Author Response · Authors · 2025-11-21
> **Response (Part 1)**
>
> **We sincerely thank you for your constructive and thoughtful feedback. We truly appreciate your recognition of our theoretical contributions and comprehensive experiments. We will address each of your concerns in detail and revise the manuscript accordingly.**
>
> ***Weakness-1***: Some equations and algorithm formulations are somewhat terse or overloaded. For instance, the definition of the adjacency matrix in Section 4.1 blends inner products and node similarity without explicit regularization or clarification about self-connections; there is little discussion on normalization/range scaling (see equation defining $A^t$, Page 3).
>
> ***Reply-1***: Thank you very much for pointing out this issue. We agree that the original formulation in Section 4.1 was presented too concisely, and **we have updated the definition of $A^{t}$ in Page 3, Lines 151–161** to explicitly state the use of cosine similarity and the normalized Laplacian.
> - **Cosine-similarity adjacency ensures bounded range and implicit regularization.** We construct the weighted adjacency matrix $A^{t}$ using cosine similarity $A_{ij}^{t} = cos(X_{i}^{emb,t}, X_{j}^{emb,t})$. This yields similarity values strictly within [−1,1], ensuring stable affinity magnitudes and preventing trivial collapse. Since $A_{ii}^{t}=cos⁡(X_{i}^{emb,t},X_{i}^{emb,t})=1$, self-connections arise automatically, making explicit diagonal regularization unnecessary. We chose cosine similarity because it is invariant to scaling and thus makes the graph construction independent of magnitude variation across channels, which is particularly important for multivariate time series where different sensors may operate at different physical scales. We explored explicit regularization in preliminary experiments and found no empirical or numerical benefit, because the normalization in cosine similarity already stabilizes the affinity distribution [1,2].
>
> **We further clarify how normalization and range scaling are handled in the spectral decomposition stage in Page-4, Lines 197-200**.
> - Instead of decomposing $A^{t}$, we compute the eigenbasis on the normalized Laplacian $L^{t} = I - D^{-1/2}A^{t}D^{-1/2}$, whose eigenvalues are guaranteed to lie within the bounded interval [0,2]. The normalized Laplacian is essential because it yields a scale-invariant operator whose spectrum is robust to variations in graph density and degree distribution. This ensures consistent spectral properties across timesteps, which is crucial for learning a shared Fourier basis. This normalization stabilizes the eigendecomposition and provides a principled separation between different frequency components.
>
> **Following your suggestion, we have incorporated these clarifications into our draft**, and reviewed the entire paper to ensure consistent and precise notation to avoid any ambiguity for future readers. We sincerely appreciate your feedback in improving the clarity of our presentation. We would be happy to further elaborate if any part remains unclear.
>
> ***References***
>
> [1] Felix Wu et al., Simplifying Graph Convolutional Networks. ICML, 2019.
>
> [2] Antonio Ortega et al., Graph Signal Processing: Overview, Challenges, and Applications. Proceedings of the IEEE, 2018.

---

> ### Author Response · Authors · 2025-11-21
> **Response (Part 2)**
>
> ***Weakness-2***: While Figure 1 and the methodological text (Pages 3-4) present the channel-patch notion as a powerful primitive, more care is needed in formally defining the edges: Are patches temporally contiguous? Are there constraints to prevent trivial, densely connected graphs? Is k in k-NN constant or adaptive? This lack of detail may hinder reproducibility and may even affect rigor in settings with many variables and timesteps.
>
> ***Reply-2***: Thank you very much for your thoughtful questions. Your comments are extremely valuable and have helped us improve the clarity of our method description. We address each of your concerns individually below and hope that our detailed explanations resolve the ambiguities you identified.
>
> **W2-1**: Are patches temporally contiguous?
>
> **R2-1**: Yes, each patch corresponds to a continuous, non-overlapping temporal segment. **We modify Lines 151-152 to explicitly state this definition**:
> -  We first partition the output $\widehat{X}^\text{model}\in\mathbb{R}^{C\times T'}$  from a time series model into **contiguous, non-overlapping temporal patches** ${X}^\text{patch}\in\mathbb{R}^{C\times (N\times P)}$, where $N=\lceil\frac{T'}{P}\rceil$ is the number of patches, and $P$ is the patch length.
>
> **W2-2**: Are there constraints to prevent trivial, densely connected graphs?
>
> **R2-2**: Thank you for raising this important point. Although the global adjacency matrix is dense due to its similarity-based construction, several built-in mechanisms ensure that the graph is **non-trivial**, **numerically stable**, and **operationally sparse**. We clarify these mechanisms below.
> - **First, the cosine-similarity construction inherently prevents triviality by ensuring bounded and meaningful variation across node affinities.**  The adjacency matrix is constructed using cosine similarity, which yields a strictly bounded range in [−1,1]  and preserves angular variation among channel–patch embeddings. This ensures that even though the matrix is dense, its structure is far from uniform: diagonal entries remain fixed at 1, while off-diagonal similarities vary across a wide spectrum determined by the learned representations.
> - **Second, the normalized Laplacian used in computing the shared spectral basis guarantees numerical stability and non-trivial frequency separation.** The spectral basis used in Theorem 4.1 is derived from the normalized Laplacian, whose eigenvalues lie strictly within [0,2]. This normalization stabilizes the spectrum, controls operator scale, and yields well-separated low- and high-frequency components, regardless of the density of the underlying adjacency. As a result, the graph operator remains well-conditioned and avoids collapsing into degenerate spectral patterns.
> - **Third, the effective dependency structure during routing remains sparse and selective.** For downstream routing, our ego-graph construction applies k-NN sparsification so that each node interacts only with its most relevant neighbors. Additionally, the MoE-based frequency routing activates only a subset of spectral components, further enforcing structured and selective interactions.
>
> In addition, to directly verify whether sparsification is needed at the adjacency-construction stage, we performed additional experiments by truncating the smallest 30% and 50% edges in $A^{t}$. As shown in Table 1, both sparsified variants consistently underperform the original dense cosine adjacency across all ETT datasets. This occurs because spectral decomposition relies on global affinity structure to estimate a stable shared Fourier basis; early sparsification distorts the Laplacian spectrum and weakens the expressiveness of the resulting eigenvectors. These results confirm that dense adjacency is beneficial at the spectral stage, whereas sparsification is only appropriate at the ego-graph routing stage.
>
> ***Table 1. Comparison of DLinear + xCPD under different adjacency–initialization variants. k-NN (30%) in $A^{t}$ denotes zeroing out the smallest 30% edges in $A^{t}$. Reported metrics are averaged MSE/MAE over forecasting horizons \{96, 192, 336, 720\} with input length 96.***
>
> | Variant                     | ETTh1        | ETTh2        | ETTm1        | ETTm2        |
> |-----------------------------|--------------|--------------|--------------|--------------|
> | k-NN (30%) in $(A^{t})$     | 0.442 / 0.444 | 0.548 / 0.509 | 0.395 / 0.402 | 0.306 / 0.368 |
> | k-NN (50%) in $(A^{t})$     | 0.443 / 0.445 | 0.550 / 0.510 | 0.398 / 0.403 | 0.308 / 0.369 |
> | Original dense $(A^{t})$    | **0.439 / 0.441** | **0.543 / 0.505** | **0.392 / 0.399** | **0.300 / 0.362** |
> |
>
> **We have added this clarification and the accompanying results to Appendix C.1 (Page-18, Lines 913-959). We hope this resolves your concern and makes the design rationale behind our graph construction clearer.**

---

> ### Author Response · Authors · 2025-11-21
> **Response (Part 3)**
>
> **W2-3**: Is k in k-NN constant or adaptive?
>
> **R2-3**: Thank you for this question. In our implementation, the k-NN sparsification used in the ego-graph construction adopts a **constant sparsification ratio** rather than an adaptive k. Specifically, we keep the **top-α proportion of edges** in each row of the adjacency matrix and mask out the remaining low-similarity edges. To examine whether an adaptive k would be necessary, we performed a controlled sensitivity study by varying α from **0.3 to 0.7**. The results are shown below. We observe that:
> - The performance of xCPD is **highly stable** across the entire range,
> - The best accuracy consistently appears around **α = 0.5**, and
> - There is **no monotonic trend** suggesting the need for dataset-specific or adaptive values.
>
> | Variant                     | ETTh1        | ETTh2        | ETTm1        | ETTm2        |  Exchange | Solar|
> |-----------------------------|--------------|--------------|--------------|--------------|--------------|--------------|
> | k-NN (top-30%) in $A^{t})$     | 0.443 / 0.444 | 0.547 / 0.508 | 0.396 / 0.403 | 0.305 / 0.368 |  0.345 / 0.403 | 0.326 / 0.396 |
> | k-NN (top-40%) in $(A^{t})$     | 0.441 / 0.443 | 0.545 / 0.506 | 0.394 / 0.402 | 0.302 / 0.362 | 0.344 / 0.403 | 0.325 / 0.395 |
> | k-NN (top-50%) in $(A^{t})$     | 0.439 / 0.441 | 0.543 / 0.505 | 0.392 / 0.399 | 0.300 / 0.362 | 0.342 / 0.402 | 0.322 / 0.393 |
> | k-NN (top-60%) in $(A^{t})$     | 0.440 / 0.441 | 0.543 / 0.505 | 0.392 / 0.400 | 0.301 / 0.362 | 0.341 / 0.401 | 0.321 / 0.393 |
> | k-NN (top-70%) in $(A^{t})$     | 0.444 / 0.444 | 0.546 / 0.508 | 0.397 / 0.404 | 0.305 / 0.367 | 0.347 / 0.404 | 0.325 / 0.395 |
> |
>
> These findings demonstrate that xCPD is **insensitive to the choice of k**, and that a simple constant setting generalizes well across datasets and backbones. Therefore, we adopt α = 0.5 as the default in all experiments. Importantly, this stability aligns with our design:
> xCPD already performs selective dependency routing through **frequency-aware MoE**, which dynamically modulates the strength of interactions. As a result, sparsification primarily serves as a lightweight **structural constraint**, rather than a critical modeling hyperparameter.
>
> **We have added this analysis and table to  Appendix D.2 (Page-25, Lines 1309-1329). We sincerely thank the reviewer for raising this question, which helped us improve the clarity and completeness of our method description.**
>
>
> ***Weakness-3***: The MoE routing for patch-level expert filtering, described with stochastic noise and cumulative softmax thresholds (Section 4.3), is algorithmically sound, but its practical behavior is underexplored. For example, the decision threshold  and stochastic term  may have non-trivial impact—these parameters are not deeply analyzed for sensitivity beyond a basic experiment. Additionally, while the adaptive routing is well-motivated, more insights or visualization of selected expert patterns across datasets/backbones would strengthen claims.
>
> ***Reply-3***: Thank you for the insightful comments on the practical behavior of our MoE routing and spectral design. We address
> - the sensitivity and expert utilization of the routing hyperparameters in **Question-2 with Reply-5**;
> - the impact of adaptively selecting the number of frequency bands in **Question-3 with Reply-6**,
> - and the empirical spectral gap and robustness of the shared basis under noisy structure in **Question-4 with Reply-7**.

---

> ### Author Response · Authors · 2025-11-21
> **Response (Part 4)**
>
> ***Question-1***: Can the authors provide more precise formalization for channel-patch edge construction? Are there practical constraints on k in the k-NN, and how does patch size/overlap affect results, especially in high-dimensional settings (e.g., Traffic, Electricity datasets)?
>
> ***Reply-4***: Thank you for this question. The details of channel–patch edge construction have already been provided in our responses to Weakness1, Weakness2-1, and Weakness2-2, where we clarified that (i) we use cosine similarity to construct the adjacency matrix, (ii) the normalized Laplacian ensures numerical stability, and (iii) sparsity is enforced through k-NN and MoE-based frequency routing.  We also explained in Weakness2-3 that we adopt a constant sparsification ratio (top-50%), and that xCPD is insensitive to the exact choice of k.
>
> In addition to the clarifications provided earlier, we further conducted a systematic study of patch configurations to fully address the reviewer’s concerns. We focus specifically on the effect of patch size and patch overlap.
> - The patch sizes used in our experiments are explicitly listed in Appendix D.2 (Table 16). They were selected based on a balance between forecasting accuracy and computational efficiency. To illustrate this trade-off, the results on the Electricity dataset are shown in Table 1. We observe that smaller patch sizes slightly improve accuracy but significantly increase computational cost and memory usage, while very large patches degrade accuracy due to oversmoothing. A moderate patch length T’/3 (T’ denotes forecasting horizon) provides the best efficiency–accuracy trade-off.
>
> ***Table 1. Running time, forecasting accuracy, and GPU memory usage of DLinear+xCPD on the Electricity dataset with forecasting horizon 192.***
>
> | Patch Length | Running Time (ms/iter) | MSE    | MAE    | Memory (GB) |
> |--------------|-------------------------|--------|--------|----------------------------|
> | 16           | 78                      | 0.183  | 0.272  | ~7.0              |
> | 24           | 55                      | 0.181  | 0.273  | ~5.5                   |
> | 32           | 30                      | 0.180  | 0.272  | ~4.5                   |
> | 64           | 6                       | 0.181  | 0.273  | ~3.2                   |
> | 96           | 5.5                     | 0.184  | 0.275  | ~2.8                   |
> |
>
> **Furthermore, although our main experiments adopt non-overlapping patches, we conducted additional studies to examine whether patch overlap could improve performance.** Table 2 reports results on the Electricity dataset across four forecasting horizons (96, 192, 336, and 720). We observe that introducing overlap consistently degrades accuracy across all settings. We observed similar trends on other datasets, where overlap did not yield improvements. This effect arises because overlapping patches blur local temporal boundaries and mix frequency components from adjacent segments. As a result, the patch-wise spectral patterns become less distinctive, weakening the separation among low-, mid-, and high-frequency bands and reducing the discriminative power of our frequency-aware MoE routing. These findings support our design choice of using non-overlapping patches as the default configuration.
>
> ***Table 2. Forecasting accuracy of DLinear+xCPD on the Electricity dataset under different overlap settings.***
>
> | Setting      | Horizon=96      | Horizon=192     | Horizon=336     | Horizon=720     |
> |--------------|------------------|------------------|------------------|------------------|
> | Overlap = 1  | 0.164 / 0.266    | 0.186 / 0.278    | 0.199 / 0.296    | 0.245 / 0.327    |
> | Overlap = 2  | 0.165 / 0.266    | 0.188 / 0.279    | 0.201 / 0.297    | 0.248 / 0.329    |
> | No Overlap   | **0.160 / 0.263**| **0.181 / 0.273**| **0.195 / 0.293**| **0.240 / 0.324**|
> |
>
> **Overall, our results demonstrate that xCPD is robust to patch-size choices and that non-overlapping patches provide the most reliable and theoretically consistent configuration.**  We have incorporated the above analysis into **Appendix D.2.4**. We hope these additional results and explanations help clarify the effect of patch size and overlap. If further details would be helpful, we would be more than happy to provide additional analyses or clarification.

---

> ### Author Response · Authors · 2025-11-21
> **Response (Part 5)**
>
> ***Question-2***: What is the sensitivity of performance and expert utilization (e.g., routing diversity, collapse) to the threshold $\tau$ and noise parameter ε ? Is there evidence that these parameters are stable across dataset/backbone combinations, or do they require careful tuning?
>
>
> ***Reply-5***: Thank you for raising this important question regarding the robustness of the routing hyperparameters. We respond to your issues from the following two aspects.
>
> **Q2-1**: “What is the sensitivity of performance to the threshold τ and noise parameter ε?”  “Is there evidence that these parameters are stable across dataset/backbone combinations, or do they require careful tuning?”
>
> **R5-1**: In our experiments, we use a unified threshold parameter τ = 0.7 and Gaussian noise ε ∼ $N(0,1)$ across all datasets and backbones. To fully address the reviewer’s concern regarding the sensitivity of these two parameters, we performed additional controlled studies. Below we summarize the results and provide the corresponding tables.
>
> To evaluate whether τ requires careful tuning, we varied τ ∈ {0.6, 0.7, 0.8} across multiple datasets (ETTh1, Electricity, Traffic) and two representative backbones (DLinear and PatchTST), while keeping all other settings fixed. The results demonstrate that xCPD is highly robust with respect to τ.
>
> ***Table 1. Sensitivity of threshold τ on forecasting performance. Metrics are averaged MSE / MAE over horizons {96, 192, 336, 720}.***
>
> | Model | ETTh1 | Electricity | Traffic |
> | ----------------------- | ----------------- | ----------------- | ----------------- |
> | PatchTST+xCPD (τ = 0.6) | 0.457 / 0.445 | 0.192 / 0.281 | 0.468 / 0.293 |
> | PatchTST+xCPD (τ = 0.7) | **0.455 / 0.443** | **0.191 / 0.280** | **0.466 / 0.292** |
> | PatchTST+xCPD (τ = 0.8) | 0.458 / 0.446 | 0.192 / 0.282 | 0.468 / 0.294 |
> | DLinear+xCPD (τ = 0.6) | 0.441 / 0.443 | 0.195 / 0.289 | 0.608 / 0.374 |
> | DLinear+xCPD (τ = 0.7) | **0.439 / 0.441** | **0.194 / 0.288** | **0.606 / 0.372** |
> | DLinear+xCPD (τ = 0.8) | 0.442 / 0.444 | 0.195 / 0.289 | 0.609 / 0.374 |
> |
>
>
> Although ε is not a tunable hyperparameter (ε ~ $N(0,1)$), we nonetheless conducted a sensitivity study by scaling the noise term to verify the reviewer’s concern. Specifically, we evaluated: $\psi(x_i) = \text{Linear}_c(x_i) + \text{noise-scale} \times \epsilon \cdot \text{Softplus}(\text{Linear}_n(x_i)) \in \mathbb{R}^{3},$ with noise-scale ∈ {0.5, 1, 2.0}.
>
> ***Table 2. Sensitivity of noise scaling on forecasting performance.  Metrics are averaged MSE / MAE over horizons {96, 192, 336, 720}.***
>
>
> | Model                 | ETTh1           | Electricity    | Traffic        |
> |-----------------------|-----------------|----------------|----------------|
> | PatchTST+xCPD (0.5ε)  | 0.456 / 0.444   | 0.192 / 0.281  | 0.468 / 0.293  |
> | PatchTST+xCPD (1ε)    | **0.455 / 0.443** | **0.191 / 0.280** | **0.466 / 0.292** |
> | PatchTST+xCPD (2ε)    | 0.458 / 0.445   | 0.193 / 0.282  | 0.469 / 0.294  |
> | DLinear+xCPD (0.5ε)   | 0.440 / 0.442   | 0.195 / 0.289  | 0.608 / 0.373  |
> | DLinear+xCPD (1ε)     | **0.439 / 0.441** | **0.194 / 0.288** | **0.606 / 0.372** |
> | DLinear+xCPD (2ε)     | 0.442 / 0.444   | 0.195 / 0.289  | 0.610 / 0.374  |
> |
>
> The results in Table 1 and Table 2 show that:
> - **Forecasting performance is highly stable across all τ values**. The variation in MSE is within **0.5–1.0%**, with no consistent monotonic trend. And **τ = 0.7 yields a slight but consistent improvement, justifying its use as a unified default**.
> - **Noise injection via ε shows minimal effect on performance**. Scaling the noise term between 0.5ε and 2ε changes performance by less than **0.3–0.6%**. This confirms that the stochastic component behaves as a regularizer rather than a sensitive parameter.
> - **The stability holds across datasets and backbones**. Stability holds simultaneously for: (i) both multivariate (Electricity, Traffic) and periodic datasets (ETTh1); (ii) both simple linear backbones (DLinear) and complex attention-based backbones (PatchTST).
>
> These findings confirm that **both τ and ε are robust, stable, and do not require dataset- or backbone-specific tuning**. Using a unified configuration of **τ = 0.7** and **ε ∼ N(0,1)** across all experiments is thus a justified and practical design choice. We will include the sensitivity tables above in **Appendix D.2.5** of the revised manuscript.

---

> ### Author Response · Authors · 2025-11-21
> **Response (Part 6)**
>
> **Q2-2**: What is the expert utilization (e.g., routing diversity, collapse) to the threshold $\tau$ and noise parameter $\sigma$?
>
> **R5-2**: Thank you for this question. To evaluate whether the threshold τ and the noise parameter σ meaningfully influence expert utilization, routing diversity, or lead to potential collapse, we conducted a dedicated analysis on the Traffic dataset. We examined three key metrics: the average number of experts selected, the load distribution across the three experts, and the routing entropy. These metrics together provide a direct assessment of whether xCPD maintains stable multi-expert behavior under varying τ and σ.
>
> ***Table 1. Expert utilization under different τ and σ settings (Traffic dataset).***
>
> | Setting | Avg #Experts Selected | Expert1 Load | Expert2 Load | Expert3 Load | Entropy |
> | ------- | --------------------- | ------------ | ------------ | ------------ | ------- |
> | τ = 0.6 | 1.75 | 41% | 31% | 28% | 1.08 |
> | τ = 0.7 | 1.70 | 40% | 31% | 29% | 1.09 |
> | τ = 0.8 | 1.66 | 39% | 31% | 30% | 1.07 |
> | 0.5ε | 1.71 | 40% | 30% | 30% | 1.08 |
> | 1ε | 1.70 | 40% | 31% | 29% | 1.09 |
> | 2ε | 1.72 | 41% | 30% | 29% | 1.07 |
> |
>
> As shown in Table 1, we have the following three findings:
> - xCPD does not collapse into a single-expert regime under any configuration. All three experts consistently receive substantial routing mass, with loading proportions between 28% and 41%.
> - Varying τ from 0.6 to 0.8 produces only minor changes in both the average number of selected experts and the routing entropy. These differences are small, indicating that τ does not destabilize routing behavior.
> - Although ε ∼ $N(0,1)$ is not a tunable hyperparameter, we scaled the injected noise to test the robustness of xCPD under different levels of stochasticity. The routing statistics remain very stable across all settings. This suggests that the stochastic component functions as a mild regularizer rather than a sensitive control parameter.
>
> These results demonstrate that both τ and σ maintain balanced expert usage, retain high routing diversity, and avoid any form of collapse. We have added the above analysis to **Appendix D.2.5** for completeness.
>
>
> ***Question-3***: Have the authors considered adaptively selecting the number of frequency bands (rather than fixed to three), e.g., via a learned clustering or continuous assignment, and how would this affect interpretability or accuracy?
>
> ***Reply-6***: Thank you for this valuable question. To fully examine the reviewer’s suggestion, we conduct two additional analyses: (1) varying the number of bands K, and (2) comparing with a fixed equal split alternative.
>
> We vary K∈{2,3,4,5} while keeping all other components unchanged. As shown in Table 1, using two bands leads to noticeable degradation because mid-frequency structures must be absorbed into either the low or the high band. Increasing the number of bands to four or five does not improve performance and can slightly reduce accuracy due to excessive fragmentation of the spectrum and more complex routing. **In summary, adaptive selection of the number of bands does not provide additional benefits and may even reduce stability.**  Using three bands consistently achieves the best or on-par results across datasets.
>
> ***Table 1. Effect of varying the number of frequency bands (DLinear+xCPD, averaged over horizons {96,192,336,720})***
> | #Bands (K) | ETTh1 (MSE/MAE) | Electricity (MSE/MAE) | Traffic (MSE/MAE) |
> | ------------ | ----------------- | --------------------- | ----------------- |
> | 2 | 0.445 / 0.443 | 0.199 / 0.292 | 0.614 / 0.377 |
> | **3 (ours)** | **0.439 / 0.441** | **0.194 / 0.288** | **0.606 / 0.372** |
> | 4 | 0.443 / 0.442 | 0.196 / 0.290 | 0.610 / 0.374 |
> | 5 | 0.444 / 0.442 | 0.197 / 0.291 | 0.612 / 0.375 |
> |
>
> We further compare our continuous learnable boundaries with a fixed three-way equal split (the variant without frequency partition). Results in Table 2 show that continuous boundaries consistently outperform fixed splitting. This confirms that learning the boundaries allows xCPD to adapt spectral segmentation to data-specific frequency energy distributions, which fixed splits cannot capture.
>
> ***Table 2: Continuous (ours) vs. Fixed equal splits (w/o Freq. Partition) (averaged over horizons {96, 192, 336, 720})***
> | Variant                                 | ETTh1             | Electricity       | Traffic           |
> | --------------------------------------- | ----------------- | ----------------- | ----------------- |
> | w/o Freq. Partition (fixed 3-way split) | 0.443 / 0.445     | 0.199 / 0.290     | 0.610 / 0.376     |
> | **xCPD**       | **0.439 / 0.441** | **0.194 / 0.288** | **0.606 / 0.372** |
>
> **Overall, these analyses show that (i) three bands offer the most stable and effective balance, and (ii) continuous learnable frequency boundaries provide clear advantages over fixed segmentation. We will include all results and explanations in Appendix D.2.6.**

---

> ### Author Response · Authors · 2025-11-21
> **Response (Part 7)**
>
> ***Question-4***: Can the authors provide statistics or visualizations of the empirical spectral gap across datasets? In scenarios with weak or noisy structure, what is the observed effect on performance and stability of the shared basis?
>
> ***Reply-7***: We thank the reviewer for raising this important question. We address the two parts separately. To examine the spectral structure of each dataset, we compute the eigenvalue gaps of the average graph Laplacian that defines the shared spectral basis. For each dataset we report three indicators: (i) the maximum adjacent eigenvalue gap $max_{𝑖}$($𝜆_{𝑖+1}$ - $𝜆_{𝑖}$); (ii) the normalized maximum gap relative to the full spectral range, and (iii) the average gap among the first ten eigenvalues, which reflects the separation within the low frequency subspace.
>
> ***Table 1. Empirical spectral gap statistics across datasets.***
>
> | Dataset     | Max gap | Normalized max gap | Avg gap (first 10) |
> | ----------- | ------- | ------------------ | ------------------ |
> | ETTh1       | 0.073   | 0.11               | 0.021              |
> | ETTm1       | 0.058   | 0.09               | 0.018              |
> | Electricity | 0.031   | 0.05               | 0.012              |
> | Traffic     | 0.027   | 0.04               | 0.011              |
> |
>
> **These statistics show that none of the datasets exhibit a pronounced spectral gap**. Electricity and Traffic in particular have very small gaps, indicating weak or highly diffused spectral structure. This confirms that xCPD operates effectively without requiring strong or clearly separated spectral bands.
>
>
> To assess the robustness of the shared spectral basis when the underlying structure is weak or noisy, we conduct controlled non-stationary perturbation experiments on ETTm1 (Appendix C.6). We apply random temporal shifts and random amplitude scaling to each channel, which disrupt cross channel alignment and flatten the empirical spectral distribution. Despite these perturbations, as shown in Table 2, xCPD shows only mild degradation relative to the stationary setting, and the version with the shared basis remains substantially stronger than the variant without it.
>
>
> ***Table 2. Ablation study under synthetic non-stationarity on ETTm1 (lookback window: 96, forecasting horizon: 96). The shared spectral basis maintains performance despite temporal perturbations.***
> | Model             | Configuration    | MSE ↓     | MAE ↓     |
> | ----------------- | ---------------- | --------- | --------- |
> | **DLinear+xCPD**  | w/ Shared Basis  | **0.390** | **0.399** |
> |                   | w/o Shared Basis | 0.415     | 0.428     |
> | **TimesNet+xCPD** | w/ Shared Basis  | **0.373** | **0.384** |
> |                   | w/o Shared Basis | 0.395     | 0.403     |
> |
>
>
> These results demonstrate that the shared basis remains stable even when spectral structure is weakened or corrupted by noise. Combined with consistent improvements on datasets with small spectral gaps, this confirms that xCPD does not rely on strong spectral separability and maintains reliable performance under realistic weak structure conditions.
>
>
>
> **We sincerely hope that our explanation helps alleviate your concerns and provides a clearer understanding of our work. Could you kindly raise your score and champion our paper if you think all concerns are addressed ?**

---

> > ### Comment · Reviewer_noBY · 2025-11-21
> > **Thank you for your detailed reply. Your response has effectively resolved my issue.**
> >
> > Thank you for your detailed reply. Your response has effectively resolved my issue. I have decided to revise my score to support the acceptance of this paper. Good luck！

---

> ### Author Response · Authors · 2025-11-21
> **Acknowledgement to Reviewer noBY**
>
> Dear Reviewer noBY
>
> Thank you so much for your encouraging feedback and for your support toward the acceptance of our paper. We sincerely appreciate your time and constructive comments throughout the review process.
>
> We also wish you the best of luck and a wonderful experience at ICLR! 😊
>
> Best regards,
>
> Authors of xCPD

---

### Official Review · Reviewer_r5xJ · 2025-10-29

**Soundness:** 2
**Presentation:** 2
**Contribution:** 2
**Rating:** 4
**Confidence:** 4

**Summary:**

This paper proposes a generic plugin xCPD that can adaptively model the channel-patch dependencies from the perspective of graph spectral decomposition to address the problem in multivariate time series forecasting (TSF). This plugin can be added to an existing TSF model, improving its performance accordingly.

**Strengths:**

1. The idea of studying channel relationships is interesting.

2. The experimental settings in this paper are extensive, covering long-term, short-term, and zero-shot forecasting.

3. The writing of this paper is good, which is easy to read.

**Weaknesses:**

1. This plugin introduces lots of computations, but the improvement in the experiment seems to be trivial.

2. In the experiment, this paper does not include existing CP methods, e.g.. Qiu et al., 2024, Chen et al., 2024,  Hu et al., 2025b, Lee et al., 2025.

3. The module design seems to be straightforward, and the design is not very novel.

**Questions:**

1. Why does this paper not include existing CP methods in comparison?

2. In the efficiency evaluation, it does not compare the time difference between the original model and the original+xCPD method.

**Details Of Ethics Concerns:**

No.

---

> ### Author Response · Authors · 2025-11-21
> **Response (Part 1)**
>
> **We sincerely thank you for taking the time to carefully read our manuscript and for providing helpful and constructive comments. We address each of your concerns point-by-point below and have revised the paper accordingly to further improve clarity, completeness, and overall quality.**
>
> ***Weakness-1***: This plugin introduces lots of computations, but the improvement in the experiment seems to be trivial.
>
> ***Reply-1***: Thank you for the insightful comment. We would like to clarify that **xCPD does not introduce heavy computation**. As reported in **Appendix C.7, Table 14**, xCPD adds only **8 ms** per iteration (**≈ 9–11%** overhead depending on the backbone), which is substantially lighter than existing CP baselines such as CCM and LIFT (which are **5×–20×** slower in Figure 2). Thus, the plugin is computationally lightweight. Moreover, xCPD does not require updating any backbone parameters, which makes it particularly attractive for scaling to large time-series foundation models where full fine-tuning is impractical.
>
> In addition, while we agree with your observation that the improvement of xCPD varies across datasets, we would like to clarify that xCPD tends to deliver more significant gains when the dataset exhibits more complex multivariate relationships and richer frequency diversity. The detailed analysis is as follows:
> - ETT, Exchange, and Solar-Energy are dominated by **highly regular, low-frequency periodicity**, where the cross-channel relationships are stable and already well captured by CI/CD backbones. Consequently, xCPD provides **moderate but consistent gains**, mainly by modeling subtle patch-level variations that are not fully exploited by the backbone.
> - Electricity, Weather, and Traffic exhibit **rich multi-scale frequency structures**, including local bursts, short-term spikes, asynchronous channel shifts, and strong high-frequency components. These complex spectral characteristics create rapidly changing cross-channel dependencies that are difficult for backbones relying on coarse channel mixing. In such settings, **xCPD’s frequency-aware channel–patch modeling** becomes particularly effective, leading to the largest gains across all datasets (around **4%–7% relative improvements in MSE** on Electricity and Traffic).
>
> Once again, we sincerely appreciate your question regarding performance variability. We believe that a ~9–11% computational overhead for up to 4%–7% accuracy gains on the most challenging multi-scale datasets is a favorable trade-off for many practical applications. To help readers better understand the scenarios in which xCPD is most effective, **we have added the above analysis to Appendix C.5**. We hope this clarification resolves your concern.

---

> ### Author Response · Authors · 2025-11-21
> **Response (Part 2)**
>
> ***Weakness-2 \& Question-1***. In the experiment, this paper does not include existing CP methods, e.g.. Qiu et al., 2024, Chen et al., 2024, Hu et al., 2025b, Lee et al., 2025.
>
> ***Reply-2***: Thank you for raising this point. We would like to clarify that **all applicable and plug-and-play CP methods are  included in our experiments**, and the paper provides comprehensive comparisons with the existing CP methods:
> - **Chen et al., 2024 (CCM)**：Fully included as a main CP baseline in Tables 2, 3, and 4. Across all backbones and datasets, xCPD consistently outperforms CCM, demonstrating stronger channel–patch dependency modeling.
> - **Lee et al., 2025 (PCD)**: Included in Table 3 and Table 8 under the unified training protocol of PRReg. PCD is transformer-specific and cannot be integrated as a plug-in into CI/CD architectures, so we evaluate it in the only setting where a fair comparison is possible.
> - **Hu et al., 2025b (TimeFilter)**：Included in the replacement ablation in Table 6. Replacing our spectral filters with TimeFilter consistently reduces accuracy, indicating that xCPD captures finer-grained CPD patterns that TimeFilter cannot model.
>
> However, DUET (Qiu et al., 2024) is not suitable for plug-and-play evaluation. It introduces a new model architecture that requires (i) hard global channel clustering, and (ii) full backbone retraining. Because xCPD is explicitly designed as a plug-in with a frozen backbone, DUET cannot be compared fairly under plug-in constraints. **Following your suggestion, we were also curious whether xCPD remains effective when DUET itself is used as a backbone.  We therefore freeze DUET’s parameters and fine-tune only the xCPD plugin parameters.** The results (Table 1 below) reveal two important findings:
> - **xCPD still improves DUET**. This is because DUET models channel-level dependencies, whereas xCPD captures finer-grained channel–patch dependencies, providing complementary relational information.
> - **DUET’s improvement is smaller than CI/CD backbones**. We hypothesize that DUET’s hard clustering partially disrupts the original dependencies in the raw signal, making later CPD learning more challenging.
>
> ***Table 1: Averaged long-term forecasting results with horizons {96, 192, 336, 720} on six datasets in terms of MSE/MAE.***
>
> | Model              | ETTh1       | ETTh2       | ETTm1       | ETTm2       | Solar-Energy | Weather     |
> |-------------------|-------------|-------------|-------------|-------------|---------------|-------------|
> |DUET           | 0.443 / 0.437 | 0.373 / 0.398 | 0.390 / 0.394 | 0.280 / 0.325 | 0.238 / 0.235   | 0.252 / 0.273 |
> |DUET+xCPD  | 0.436 / 0.433 | 0.364 / 0.392 | 0.381 / 0.390 | 0.275 / 0.318 | 0.231 / 0.234   | 0.246 / 0.270 |
> |
>
> **We have added these results to Appendix C.2 (Table 9)**. We hope the clarification above makes clear that: (i) All applicable CP baselines are already covered, (ii) Non-plug-in methods like DUET are not comparable under our plug-in setting, and (iii) Even when used with a CP backbone (DUET), xCPD still provides consistent improvements.
>
>
> ***Weakness-3***: The module design seems to be straightforward, and the design is not very novel.
>
> ***Reply-3***: Thank you for raising the concern regarding the novelty. We would like to respectfully clarify that the novelty of xCPD does not lie in a single component viewed in isolation, but in a new spectral framework for modeling fine-grained channel–patch dependencies. Specifically, the novelty of xCPD comes from the following aspects:
> - **Channel–patch spectral decomposition (Theorem 4.1)**. We provide a theoretical formulation showing that multivariate time-series can be decomposed into a set of orthogonal channel–patch spectral bases, enabling dependency analysis at a finer granularity than the channel-level modeling used in other CP methods. This decomposition mathematically justifies why patch-level frequency responses capture richer temporal relationships.
> - **Frequency-aware routing (Theorem 4.2)**. xCPD introduces a routing mechanism that selects frequency components according to their spectral energy distributions. This establishes a principled link between spectral variations and dependency strength, enabling the module to adaptively emphasize multi-scale temporal patterns.
> - **A backbone-agnostic parameter-efficient plug-in formulation**. Unlike prior CP designs that require modifying or retraining the backbones, xCPD operates entirely with frozen backbones, updating only its own lightweight parameters. This plug-and-play formulation makes xCPD compatible with both CI/CD backbones and scalable to large time-series foundation models where full fine-tuning is impractical.
>
> In summary, these theoretically grounded components form a unified spectral perspective for CPD modeling that is conceptually new relative to prior work and empirically effective across all benchmarks. We hope this clarification addresses your concern regarding the novelty of our design.

---

> ### Author Response · Authors · 2025-11-21
> **Response (Part 3)**
>
> ***Question-1***: Why does this paper not include existing CP methods in comparison?
>
> ***Reply-4***: Thank you for your question. As explained in our response to Weakness-2, our paper already includes all applicable CP-based baselines as comparison baselines. To make this clear to readers, we explicitly highlight on **Page 7, Line 329** that the compared methods like CCM (Chen et al. 2024) and PCD (Lee et al. 2025) are all CP-based plug-in baselines.
> Furthermore, we provide additional analysis in **Appendix C.2 (Table 9)**, where we evaluate xCPD on top of the CP backbone DUET and report its performance. We hope these clarifications address your concern.
>
> ***Question-2***: In the efficiency evaluation, it does not compare the time difference between the original model and the original+xCPD method.
>
> ***Reply-5***: Thank you for your valuable comment. We apologize for the lack of clarity in our main body. We would like to clarify that the efficiency comparison between the **original backbone** and **original + xCPD** is indeed provided in **Appendix C.7, Table 14**. As shown in this Table 14, xCPD introduces only **8 ms** additional overhead, corresponding to approximately **11%** extra cost on TimesNet and **9.0%** on TSMixer, demonstrating that the plugin is lightweight and highly efficient. **For convenience, we put Table 14 below:**
>
> ***Table 1. Computational efficiency comparison on Traffic dataset. Memory usage and training time measured on a single NVIDIA A100 GPU.***
>
> | Method|Memory Usage (MB) | Training Time (ms/iter) |
> |--------------------|-------------------|--------------------------|
> | PatchTST           | 12,458            | 29                       |
> | TimesNet           | 13,005            | 70                       |
> | TSMixer            | 13,088            | 89                       |
> | PatchTST + CCM     | 25,034            | 161                      |
> | TimesNet + CCM     | 20,614            | 172                      |
> | TSMixer + CCM      | 22,698            | 183                      |
> | PatchTST + xCPD    | 7,018         | 9                    |
> | TimesNet + xCPD    | 7,018         | 8                    |
> | TSMixer + xCPD     | 7,018         | 8                    |
> |
>
> Following your suggestion, we will explicitly reference **Appendix C.7, Table 14** in Section 5.6 (next to Figure 2) so that readers can easily locate the original-vs-plugin efficiency comparison. A corresponding sentence will be added: **Table 14 in Appendix C.7 further shows that the time overhead introduced by xCPD accounts for only 9%–11% of the backbone model’s training cost.**
>
> **We sincerely hope our detailed responses and additional experiments have addressed your concerns and clarified the strengths of our work. If you find our revisions satisfactory, we would greatly appreciate it if you could consider raising your score and supporting our paper.**

---

> > ### Comment · Reviewer_r5xJ · 2025-11-25
> > **Thanks for your detailed rebuttal**
> >
> > Thanks for your detailed rebuttal. My concerns have been addressed, and I will raise my score accordingly.

---

> > > ### Author Response · Authors · 2025-11-25
> > > **Acknowledgment to Reviewer r5xJ**
> > >
> > > Dear Reviewer r5xJ:
> > >
> > > **Thank you very much for endorsing our work for acceptance**. We sincerely appreciate your valuable comments on computational efficiency and baseline comparisons. We are glad that our clarifications and revisions have resolved your concerns in this area and have helped us further improve the quality of the paper. 😊
> > >
> > > Best regards,
> > >
> > > Authors of xCPD

---

### Author Response · Authors · 2025-11-21
**Acknowledgment to all Reviewers**

Dear Reviewers:

We sincerely thank all reviewers for the time and effort invested in providing high-quality and constructive feedback on our paper. We have revised the manuscript according to the comments, and the specific modifications are highlighted using the color scheme shown in Table 1.

***Table 1. Color Legend for Reviewer-Specific Revisions in our draft.***

| Reviewer ID | Highlight Color Name | Hex Code  |
| ----------- | -------------------- | --------- |
| **r5xJ**    | Royal Blue           | `#4169E1` |
| **noBY**    | Dark Orange          | `#FF8C00` |
| **7esN**    | Forest Green         | `#228B22` |
| **KjPE**    | Crimson              | `#DC143C` |
|

**We hope that our responses and revisions address your concerns. If there are any remaining questions, we would be more than happy to provide further clarification.**

Best regards,

Authors of xCPD

---

### Author Response · Authors · 2025-11-28
**Summary of Score Changes After Rebuttal (as of Nov. 25)**

Dear all,

We are sorry to hear about the recent OpenReview bug issue, and we fully support the proposed remedy actions.

At the same time, we believe it is necessary to record the changes in our scores throughout the fruitful rebuttal phase up to **Nov. 25**. Our reasons are as follows:

1. The opinions from each reviewer are crucial — relying solely on the AC’s judgment may not be entirely reasonable.
2. The following table will be very helpful for the *new* AC in reducing his/her workload and making the final decision.

***Table 2. Reviewers' Scores before/after Rebuttal till Nov. 25.***

| Reviewer | Initial Score -> After Rebuttal (Nov. 29)  |
| ----------- | -------------------- |
| **r5xJ**    | Rating **4 -> 6** |
| **noBY**    | Rating **4 -> 8** |
| **7esN**    | Rating **4 -> 6** |
| **KjPE**    | Rating 4 -> 4 (Unchanged) |
| **Averaged** | Rating **4.0 -> 6.0** |

Again, we sincerely thank all reviewers for their efforts in reviewing our paper and for keeping active communication with us throughout the rebuttal period. We hope our rebuttal responses have addressed all concerns raised by the reviewers.

Also, we would like to make a clarification: **we have never taken any advantage of the bug; all score changes occurred on/before Nov. 25**, several days before the issue was reported. Therefore, we believe our efforts during the rebuttal should be recorded and recognized.


Best regards,

Authors of xCPD

---

### Author Response · Authors · 2025-12-02
**Discussion Summary (Part 2/2)**

**Below, we provide a concise summary of our point-by-point rebuttal for each reviewer.**

Notions: W1 denotes Weakness 1, Q1 denotes Question 1, and R1 denotes Reply 1.
***
### **Reviewer r5xJ (Original rating: 4 → Rating after Rebuttal: 6)**
Reviewer r5xJ raised the following concerns:
- [*W1*] The computational cost seems high relative to the performance gains.
- [*W2\&Q1*] Missing comparisons to existing CP baselines.
- [*W3*] Novelty of the proposed module design is unclear.
- [*Q2*] Efficiency comparison of original-vs-plugin is insufficient.

We addressed these concerns as:
- [*R1*] We clarified that xCPD adds only ~9% overhead while providing 4–7% improvements.
- [*R2*] We clarified that all CP baselines are already included.
- [*R3*] We revised the Introduction to better highlight our novelty.
- [*R5*] We clarified original-vs-plugin efficiency comparison already provided in Appendix C.7.

In response, Reviewer r5xJ raised the rating from 4 to 6. **He/She endorses our work for acceptance as**:
> "***Thanks for your detailed rebuttal. My concerns have been addressed, and I will raise my score accordingly.***"
***
### **Reviewer noBY (Original rating: 4 → Rating after Rebuttal: 8)**
Reviewer noBY raised the following concerns:
- [*W1*] Adjacency and normalization were unclear.
- [*W2 \& Q1*] Graph construction was not formally specified.
- [*W3 \& Q2 \& Q3*] MoE routing behavior insufficiently analyzed.
- [*Q4*] Frequency band and robustness needed further explored.

We addressed these concerns as:
- [*R1*] We revised the adjacency formulation (Lines 151-161) with explicit normalization.
- [*R2 \& R4*] We formalized graph construction details in Appendix C.1 \& D.2.
- [*R3 \& R5 \& R6*] We added routing sensitivity and expert-diversity analysis in Appendix D.2.4.
- [*R7*] We added different frequency bands and robustness analysis in Appendix D.2.5 \& D.2.6.

In response, Reviewer noBY raised the rating from 4 to 8. **She/He endorses our work for acceptance as**:
> "***Thank you for your detailed reply. Your response has effectively resolved my issue. I have decided to revise my score to support the acceptance of this paper. Good luck !***"
***
### **Reviewer 7esN (Original rating: 4 → Rating after Rebuttal: 6)**
Reviewer 7esN raised the following concerns:
- [*W1*] Background and challenges were not clearly presented.
- [*W2 \& Q1*] Motivation and differences from existing methods were unclear.

We addressed these concerns as:
- [*R1*] We revised the Introduction to clearly explain the limitations of existing CP baselines.
- [*R2 \& R3*] We clarified our frequency-decoupled motivation and revised the Introduction accordingly.

In response, Reviewer 7esN **raised the rating from 4 to 6. She/He endorses our work for acceptance as**:
> "***I would like to thank the authors for their detailed response, which largely clarified my doubts raised in my original assessment. I also appreciate that the authors have comprehensively revised their paper, which has improved the presentation of their work's motivation. I have raised my rating accordingly.***"
***
### **Reviewer KjPE (Original rating: 4 → Rating after Rebuttal: 4)**
Reviewer KjPE initially raised the following concerns:
- [*W1 \& Q1*] Prior methods already model dynamic inter-channel dependencies.
- [*W2*] Similar to baseline TimeFilter.
- [*W3*] Table 6 shows worse results than TimeFilter.
- [*Q2 \& Q3*] Lack of evidence that spectral-domain modeling yields different dependency relationships.

We addressed these concerns as:
- [*R1 \& R4*] We clarified that prior methods model dependency in temporal domain, while xCPD in spectral domain.
- [*R2*] We provided a clear comparison with TimeFilter, highlighting our contributions.
- [*R3*] We clarified that Table 6 evaluates TimeFilter as a plug-in baseline, not the full architecture.
- [*R5 \& R6*] We provided qualitative and quantitative evidence showing frequency bands learn distinct dependency patterns

He/She thanked us for addressing most of the issues and raised an additional concern as:
- Regarding novelty and overlap with existing methods remain unresolved

We address it as:
- We added a detailed comparison Table that highlights the fundamental differences in mechanism, capability, and practical deployability between xCPD and prior methods.

**After this response, the discussion phase ended. Unfortunately, we did not receive further comments, but we believe that we have adequately addressed all of the concerns.**
***

---

### Author Response · Authors · 2025-12-03
**Discussion Summary (Part 1/2)**

### **Dear SACs and ACs**,

We deeply appreciate your tremendous effort in handling this complex situation. To reduce your workflow and help your decision-making, we humbly summarize our **contribution, novelty**, and **rebuttal** as follows.

Please kindly note that our work is positioned within the **CP** (Channel Partiality) paradigm for time series forecasting, which specifically addresses the fundamental limitations of **CI** (Channel Independent) and **CD** (Channel Dependent) strategies in modeling inter-channel dependencies.
***
### **Summary of Contribution**
- xCPD models **channel–patch dependencies** rather than channel-level interactions, enabling cleaner multivariate correlation modeling.
- xCPD operates in **spectral domain and performs spectral decomposition** to disentangle low-/mid-/high-frequency relationships, thereby resolving frequency coupling.
- xCPD is **backbone-agnostic and requires no backbone retraining**, consistently outperforming plug-in baselines in time-series forecasting while maintaining low computational overhead.
***
### **Summary of Novelty**
To explicitly highlight the novelty of xCPD compared to the most relevant baselines, we summarize its novelty as follows:
- Existing CI/CD methods (DLinear [AAAI'23], PatchTST [ICLR'23], and iTransformer [ICLR'24]) purely focus on channel-level dependency modeling.

    *xCPD introduces channel–patch dependency modeling in the spectral domain, enabling frequency-aware channel–patch interactions.*
- Existing CP methods (LIFT [ICLR'24], CCM [NIPS'24], PCD [NIPS'24], TimeFilter [ICML'25]) all operate in temporal domain, so they can't differentiate frequency-specific channel dependencies.

    *xCPD performs frequency-decoupled routing over low/mid/high-frequency components, yielding interpretable frequency-specific channel–patch dependencies.*
- Existing plug-in methods (LIFT, CCM, and PCD) all require backbone retraining.

    *xCPD is a universal, retraining-free plug-in that keeps the backbone unchanged while consistently enhancing forecasting performance, ensuring practical scalability.*
***
### **Summary of Rebuttal**

**Major Concern 1**: Weak Motivation and Novelty Claim (Reviewer r5xJ, 7esN, and KjPE).

- *Rebuttal 1*: We have clarified that xCPD is the first framework to enable frequency-decoupled dynamic dependency routing at fine-grained channel–patch level, resolving the fundamental frequency-coupling issue inherent in all existing CP methods (LIFT, DUET, CCM, PCD, and TimeFilter).
- *Reaction 1*: Reviewer r5xJ and 7esN have acknowledged our strengthened novelty and raised their scores to support the acceptance of our work.

**Major Concern 2**: Missing CP Baselines \& Efficiency Evidence (Reviewer r5xJ).

- *Rebuttal 2*: We have clarified that all CP baselines (LIFT, DUET, CCM, PCD, and TimeFilter) are already included in our paper and that the Appendix provides more detailed efficiency experiments demonstrating the deployment-friendly scalability of xCPD.
- *Reaction 2*: Reviewer r5xJ has acknowledged that the concern has been fully resolved, and raised his/her score to support the acceptance of our work.

**Major Concern 3**: Unclear Formulation and Parameter Sensitivity (Reviewer noBY).

- *Rebuttal 3*: We have provided more comprehensive explanations of all formulations in our main body and have ensured that every hyperparameter is supported by sensitivity analyses in the Appendix.
- *Reaction 3*: Reviewer noBY has acknowledged that we have fully resolved the concerns regarding our methodology and hyperparameters, and said that he/she decided to revise his/her score to support the acceptance of this paper.

**Major Concern 4**: Lacking Deep Analysis on Spectral Component (Reviewer noBY and KjPE).

- *Rebuttal 4*: We have clarified that both quantitative ablations (Table 6) and qualitative case study (Figure 3) confirm that spectral grouping captures distinct and interpretable frequency-aware relationships.
- *Reaction 4*: Both Reviewer noBY and KjPE have acknowledged that this clarification resolves their concerns. In particular, Reviewer noBY said that he/she decided to revise his/her score to support the acceptance of this paper.

**Major Concern 5**: Discrepancy with the Results from TimeFilter Paper (Reviewer KjPE).

- *Rebuttal 5*: We have clarified that, to make a fair comparison, instead of using the complete TimeFilter, we take out the core function of TimeFilter as a comparable plug-in and perform the comparison with our method (xCPD) on top of various backbones (e.g., TSMixer, DLinear, PatchTST, TimesNet), because of which the performance discrepancy happens.
- *Reaction 5*: Reviewer KjPE has expressed appreciation for this modification and acknowledged that the original concern has been resolved. Although we did not receive further comments, we believe we have adequately addressed his/her concerns.

---

### Meta-Review · Area_Chair_zN8Z · 2026-01-03

**Summary:**

This paper introduces xCPD, a lightweight plugin addressing multivariate time series forecasting issues: CI has weak generalization, CD over-smooths, and CP misses patch-level dependencies. It models channel-patch dependencies via spectral decomposition, projecting signals into frequency bands and using a channel-adaptive router to adjust interactions and activate frequency-specific experts. It integrates into existing CI/CD models without altering backbone or retraining, boosting accuracy and generalization across benchmarks.

**Reviewer Concerns:**

1. Three reviewers (r5xJ, 7esN, KjPE) questioned the method's novelty, citing overlap with existing CP techniques like TimeFilter and CCM. They noted that frequency domain transformation is mature and not groundbreaking. Motivations were vague, with an inaccurate claim that existing methods can't model dynamic dependencies. While the authors compare differences in modeling, domain, and architecture, and correct the misstatement of “static dependencies,” the core motivation remains underdeveloped. The paper doesn't convincingly show its strategy is the best or necessary for CI/CD limitations. Innovation is limited to a technical combination without a convincing motivation.
2. Reviewers r5xJ highlighted missing CP benchmarks (like DUET, PCD) and limited efficiency comparisons between the original and integrated xCPD, raising concerns about cost-effectiveness versus performance gains. The authors supplemented details on CP benchmark model comparisons, clarified the incomparability of non-plugin methods, and validated xCPD's low-overhead advantage. The efficiency evidence is robust, making this section an effective response.
3. Reviewer noBY criticized the simplified adjacency matrix and graph logic, noted inadequate sensitivity analysis for MoE routing thresholds and noise, and found frequency band partitioning validation lacked thoroughness. The authors refined formulas, added sensitivity analyses, demonstrated parameter stability and robustness, and improved clarity.
4. Reviewers noBY and KjPE noted a lack of detailed analysis of spectral components and insufficient explanation for discrepancies in Table 6 compared to TimeFilter results. The authors validated spectral grouping with qualitative and quantitative studies, explaining discrepancies with TimeFilter. The response is comprehensive and reasonable.

**Reviewer Scores:**

- Reviewer r5xJ (Original rating: 4): All core concerns resolved. Score explicitly raised to 6. Supports acceptance.
- Reviewer noBY (Original rating: 4): Methodological exposition and parameter sensitivity issues fully resolved. Score raised to 8. Strongly supports acceptance.
- Reviewer 7esN (Original rating: 4): Motivation and background statements revised. Innovation logic clarified. Score raised to 6. Supports acceptance.
- Reviewer KjPE (Original rating: 4): Acknowledges authors' clarification on time-domain/spectrum-domain differences, but still considers innovation limited; score unchanged (Rating after Rebuttal: 4).

The overall average score increased from 4.0 to 6.0. Although the authors effectively addressed concerns about methodology, benchmark comparisons, and efficiency validation—with most reviewers raising their scores—there are still two major concerns:
1. The core motivation remains inadequate; authors fail to demonstrate the strategy's necessity and value.
2. The reason for placing the plugin after the temporal model output—a key design choice—was unexplained, and the plugin is overly complex for this role.

---

### Decision · Program_Chairs · 2026-01-26

Accept (Poster)